



# Peering into the heart of thunderstorm clouds: Insights from cloud radar and spectral polarimetry

Ho Yi Lydia Mak and Christine Unal

Department of Geoscience and Remote Sensing, Delft University of Technology, Delft, The Netherlands

**Correspondence:** Ho Yi Lydia Mak (hylmak@connect.ust.hk), Christine Unal (c.m.h.unal@tudelft.nl)

**Abstract.** Lightning is a natural phenomenon that can be dangerous to humans. It is however challenging to study thunderstorm clouds using direct observations since it can be dangerous to fly into thunderstorm clouds. In this study, cloud radar at 35 GHz is used to study the properties and dynamics of thunderstorm clouds. It is based on a case of thunderstorm on 18 June 2021 from 16:10 to 17:45 UTC near Cabauw, the Netherlands. The technique of spectral polarimetric analysis at millimeter wavelength, which has not been used in previous studies about thunderstorm clouds so far, is used to help understand the behaviours of different types of particles within a radar resolution volume. Spectral polarimetric radar variables are used to investigate possible hydrometeors in the clouds and look for vertical alignment of ice crystals that is expected due to electric torque. Due to challenges posed by Mie scattering, scattering simulations are carried out to aid the interpretation of spectral polarimetric variables. It is shown that the start of the Mie regime in the Doppler spectrum can be identified by the use of the spectral differential phase. From the results, there is a high chance that supercooled liquid water and conical graupel are present in thunderstorm clouds. There is also a possibility of ice crystals arranged in chains at the cloud top. Ice crystals become vertically aligned a few seconds before lightning and return to their usual horizontal alignment afterwards. However, this phenomenon has been witnessed in only a few cases, specifically when the lightning strike is in close proximity to the radar's line of sight or when the lightning is exceptionally strong. Doppler analyses show that updrafts are found near the core of the thunderstorm cloud, while downdrafts are observed at the edges. Strong turbulence is also observed as reflected by the large Doppler spectrum width.

## 1 Introduction

Lightning is a natural phenomena that is dangerous to humans. It is the electric discharge caused by an electrical breakdown of charges built up in a cloud. Scientists began investigating atmospheric electrification and lightning several hundred years ago. Many studies have shown that the charge distribution in most thunderclouds follow a tripole structure, with positive charges in the upper and lower levels and negative charges in the middle level (Wang, 2013). The positive charge center near the cloud base is relatively small, thus is sometimes ignored. Typically, a breakdown can occur when the environmental electric field established by the charges is around 100-300 V m$^{-1}$, though the critical field at the point of breakdown is likely much higher (Wang, 2013). During a thunderstorm, the electric field builds up and breaks down continuously. The time needed to accumulate large enough electric fields for lightning to occur ranges from less than a minute to several minutes (Gunn, 1954;





Marshall and Winn, 1982). For active thunderstorm clouds with tens of kV m$^{-1}$ in the interior, the magnitude of the electric field decreases to 3 kV m$^{-1}$ within 5 km away from the cloud edge on average (Merceret et al., 2008).

Over the years, numerous charging mechanisms were proposed to account for charge separation in thunderstorm clouds. These can be divided into three major categories: convective charging, inductive charging and non-inductive charging. Accord-
ing to the convective charging mechanism proposed by Vonnegut (1955), updrafts carry fair-weather positive charges into the cloud to form a positive charge center. Negative charges are then attracted to the top and edges of the cloud, which are subsequently brought to the lower level by downdrafts. However, numerous investigators such as Chiu and Klett (1976) have found inconsistencies between this mechanism and observations. Inductive charging includes different charge separation mechanisms that involve charges induced by the external fair-weather electric field, such as charging by selective ion capture (Wilson et al.,
1929), drop breakup charging and particle rebound charging. However, many studies have shown that these mechanisms are quantitatively unrealistic or ineffective (Pruppacher and Klett, 1980; Wang, 2013). For non-inductive charging, charge separation occurs without the presence of an external electric field. Under this category, the most widely accepted mechanism is charging due to the collision of ice crystals with riming graupel pellet, which was first studied in the laboratory by Reynolds et al. (1957). It was found that graupel pellets that are growing by the accretion of supercooled droplets acquires negative
charges as they collide with ice crystals. Takahashi (1978) further investigated this phenomenon and found that the magnitude and sign of the electrification depend largely on temperature and cloud water content. The optimal cloud water content for graupel to become highly charged is 1 to 2 g m$^{-3}$. Graupel will become positively charged if the temperature is above the charge reversal temperature $T_R$, which ranges from $-20$ °C to $-10$ °C, and negatively charged otherwise (Takahashi, 1978). Within the updraft column in a thundercloud where temperature is low, negatively charged graupel and positively charged ice
crystals will be formed. The negatively charged graupel will fall at the periphery of the column where the updraft is weak, while the positively charged ice crystals will be thrown upwards. As the graupel reach a region warmer than $T_R$, they become positively charged, forming the tripole structure of most thunderclouds. Although non-inductive charging due to the collision of graupel and ice crystals best explains tripolar cloud structure, it should be noted that all charging mechanisms above could contribute to certain extent at some time to cloud charging (Pruppacher and Klett, 1980).

To know what could be observed in thunderstorm clouds, it is important to first identify the ingredients of thunderstorms. A wide variety of ice particles can be found in thunderclouds. Ice crystals of different shapes and sizes can be formed at different temperatures and ice supersaturation (Bailey and Hallett, 2009). These crystals can grow within clouds through three major processes (Pruppacher and Klett, 1980; Lamb and Verlinde, 2011): riming, water vapor diffusional growth and aggregation. Riming occurs when supercooled water droplets collide with ice crystals and freeze on them, forming large, dense and near
spherical particles. Conical graupel can be formed if riming occurs while particles fall through strong updrafts containing water droplets. Since the bottom windward side of the particle grows faster than the top leeward side, the particle develops a conical shape (Tang et al., 2017). Scattering simulations carried out by Oue et al. (2015) and Lu et al. (2016b) showed that conical graupel can produce negative differential reflectivity ($Z_{DR}$) values at X-, Ka- and W-band. Diffusional growth takes place when water vapor diffuses towards ice crystals from gas phase. During this process, crystals keep their characteristic
shape (Lamb and Verlinde, 2011). Aggregation occurs when ice crystals collide with each other and form larger crystals that



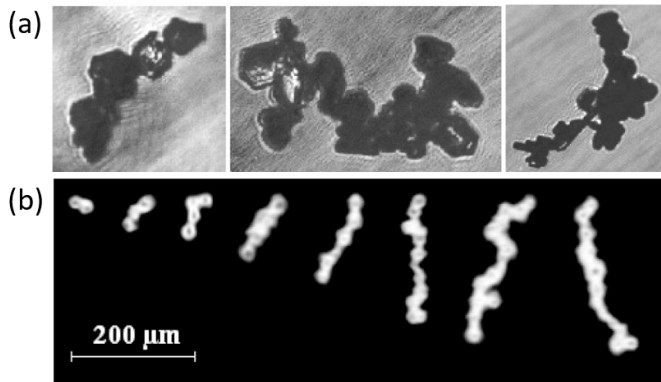

**Figure 1.** Examples of (a) plate crystals arranged into chains in anvil clouds taken by Connolly et al. (2005) (chain lengths from left to right are 381, 632 and 721 $\mu$m respectively) and (b) frozen drops arranged into chains near the top of an overshooting convective cloud taken by Gayet et al. (2012).

are more spherical in shape. When an electric field of more than 50 kV m$^{-1}$ is present, aggregation of ice crystals may be enhanced due to attractive electrical forces induced between neighbouring conducting crystals (Connolly et al., 2005), forming elongated chains rather than spherical clusters. Figure 1(a) shows some examples of plate crystals arranged into chains in anvil clouds, i.e. the region of convective cloud detraining from the main cell of the thunderstorm cloud, captured by a cloud particle imager taken by Connolly et al. (2005) at an altitude of around 12 km where the temperature is below $-40$ °C. Chain-like aggregates can also be formed from frozen droplets, such as those observed by Gayet et al. (2012) near the top of an overshooting convective cloud at 11080 m where the temperature is $-58$ °C as shown in Fig. 1(b). The enhancement of aggregation starts to decrease when the electric field exceeds 150 kV m$^{-1}$ since the strong electric field would fragment the ice particle (Connolly et al., 2005). Meanwhile, laboratory experiments have found that electric field enhanced aggregation does not occur when ice particle number concentration is below 2 cm$^{-3}$ (Wahab, 1974). High concentration of ice particles could be present in convective clouds if strong updrafts carry supercooled droplets to a level of $-37$ °C where they freeze rapidly by the process of homogeneous nucleation (Gayet et al., 2012).

Evidences of the presence of graupel and ice crystals in thunderstorm clouds were found using polarimetric and Doppler measurements. Mattos et al. (2016) used X-band radar to compare storms with and without lightning activities and analysed the vertical distribution of hydrometeors within the clouds. They found that in the lower layer of thunderclouds (from 0 to $-15$ °C), there is an enhanced positive specific differential phase shift ($K_{DP}$) probably associated with supercooled oblate raindrops lofted by updraft; in the middle layer (from $-15$ to $-40$ °C), there is negative $Z_{DR}$ and $K_{DP}$ and moderate horizontal reflectivity, which are possibly associated with the presence of conical graupel. With Ka-band cloud radar, Sokol et al. (2020) identified a mixture of hydrometeors at an elevation of 4–7 km with a predominance of ice and snow particles and graupel based on the terminal velocities of different hydrometeors. The coexistence of different types of hydrometeors is supported by the high Doppler spectrum width, which also implies the existence of collisions of hydrometeors.





In addition to the existence of a variety of hydrometeors in thunderstorm clouds, it was first suggested by Vonnegut (1965) based on changes in cloud brightness observed during lightning that ice crystals would align under strong electric field. Weinheimer and Few (1987) studied the magnitude of electric field needed to align particles of different sizes and shapes. They

compared the magnitudes of electrical torques that try to align particles' long axis with the electric field, and aerodynamic torques that attempt to align particles with their long axes perpendicular to their direction of motion. They estimated that for an electric field of $100 \, \text{kV m}^{-1}$, plates with a major dimension of less than 0.6 mm can be aligned, while the threshold is 1 mm for dendrites and 0.2 mm for thick plates. Columns of all sizes can be aligned by such a field. Meanwhile, only particles smaller than 0.05 mm can be aligned by an electric field of $10 \, \text{kV m}^{-1}$. Such alignment of ice crystals is observed in various

thunderstorm cases using polarimetric radar measurements. For example, Lund et al. (2009) observed negative $Z_{DR}$ in or near clusters of lightning initiations using S-band radar, while Mattos et al. (2016), using X-band radar, found that in the upper layer (above $-40 \, °\text{C}$) of thunderclouds, $K_{DP}$ becomes more negative with increasing lightning density. These are likely due to ice particles being aligned vertically by a large vertical electric field. Meanwhile, only one study that used cloud radar to study the alignment of ice crystals during thunderstorms is found. Using a Ka-band radar, Sokol et al. (2020) observed high

linear depolarisation ratio ($L_{DR}$) in clouds that produce lightning in the vicinity, which is likely caused by the alignment of ice crystals in an electric field.

Another important ingredient for lightning is strong updraft. According to Zipser and Lutz (1994), lightning is highly unlikely if the mean updraft speed is less than around 6 to 7 m s$^{-1}$, or the peak updraft speed is less than around 10 to 12 m s$^{-1}$. Deierling and Petersen (2008) found that time series of updraft volume in the charging zone where the temperature is below

$-5 \, °\text{C}$ with vertical velocities exceeding 10 m s$^{-1}$ is highly correlated to total lightning activity. In general, it is common to find updrafts of more than 10 m s$^{-1}$ and up to 30 m s$^{-1}$ in thunderstorms Stith et al. (2016); Marshall et al. (1995).

Up to this date, most research about thunderstorms made use of S-band (2-4 GHz), C-band (4-8 GHz) and X-band (9-12 GHz) radar, while limited studies were conducted using cloud radar with millimeter wavelength. Radars at lower frequencies are common choices for investigating thunderstorms as they have larger ranges and suffer from less attenuation, but high

frequency cloud radars could bring new insights on thunderstorm clouds before precipitation starts given their higher spatial resolution. Moreover, existing studies have only analysed integrated radar variables that include the contribution of all particles within each radar resolution volume. There have been no attempts to utilise the polarimetric Doppler spectra to disentangle the contributions of different types of particles. Nonetheless, variations in the Doppler spectra not only can indicate another type of particles, but also the presence of Mie scattering regime when the particles grow. This study explores new ways to study

thunderstorm events by using cloud radar observations and polarimetric Doppler spectra. The goal is to establish links between radar observations and physical processes in thunderstorms to enhance our understanding about lightning.

## 2 Instruments and data

The cloud radar used in this study is a dual-frequency scanning polarimetric frequency-modulated continuous-wave (FMCW) radar produced by Radiometer Physics GmbH located at Cabauw, the Netherlands (51.968° N 4.929° E). It operates at 35



**Table 1.** Configuration parameters of cloud radar at 35 GHz at 45° elevation for each chirp sequence

| Attributes | Chirp sequence | | |
| --- | --- | --- | --- |
| | 1 | 2 | 3 |
| Integration time (s) | 1.20 | 0.96 | 0.82 |
| Range interval (m) | 119.2-1192.5 | 1222.3-4889.1 | 4953.3-14969.9 |
| Range resolution (m) | 29.8 | 29.8 | 55.0 |
| Nyquist velocity ($\pm$ m s$^{-1}$) | 19.7 | 16.1 | 10.7 |
| Doppler velocity resolution (m s$^{-1}$) | 0.15 | 0.13 | 0.17 |

GHz (Ka-band) and 94 GHz (W-band) in Simultaneous Transmission Simultaneous Reception (STSR) mode and measures at a constant elevation of 45° and constant azimuth of 282° at some selected periods. Its half power beam width at 35 GHz is 0.84° and temporal sampling is 3.59 s. In this study of thunderstorm clouds, only the 35 GHz data is used since there are numerous issues associated with the 94 GHz data including significant attenuation, Doppler aliasing and complications due to Mie scattering. The configuration parameters for each chirp sequence is shown in Table 1. The transmission power is

continuously monitored, and the radar receiver (including the receiving antenna) undergoes calibration every six months using clear sky calibration. Short-term calibration is provided through periodic Dicke switching. Prior to the semiannual calibration procedure, the hydrophobic antenna radomes are replaced.

The cloud radar provides two types of output data. The Level 0 dataset contains the raw data, which includes the Doppler spectrum at horizontal and vertical polarizations ($sZ_{hh}$ and $sZ_{vv}$), as well as the real and imaginary parts of the covariance

spectrum between horizontal and vertical polarizations ($sC_{hh,vv}$). The Level 1 dataset contains processed data, including the equivalent radar reflectivity factor ($Z_e$, or $Z_{hh}$), mean Doppler velocity, Doppler spectrum width, differential reflectivity ($Z_{DR}$), copolar correlation coefficient ($\rho_{hv}$), specific differential phase shift ($K_{DP}$), and slanted linear depolarization ratio ($SL_{DR}$). $SL_{DR}$ is a proxy for $L_{DR}$, which can only be computed when the radar transmits horizontally and vertically polarized electromagnetic waves alternatively. Since the radar used in this study transmits them simultaneously, only $SL_{DR}$ is available.

The radar also has a passive broad band channel operated at a centre frequency of 31.4 GHz that provides information about the integrated liquid water path (LWP). A weather station is also attached, which provides rain rate, surface wind speed and wind direction, but does not provide the wind profile. Wind profile is obtained instead from European Centre for Medium-Range Weather Forecasting (ECMWF) Integrated Forecast System output over Cabauw (O'Connor, 2022) available at https://cloudnet.fmi.fi/. This model provides hourly forecast of zonal (eastward) and meridional (northward) wind up to 80000 m

with a horizontal resolution of 9 km. The vertical resolution of the first 10000 m ranges from around 20 m near the surface to around 300 m at the top. A microwave radiometer beside the radar provides temperature and relative humidity profiles along the zenith. Lightning data is obtained from the online lightning map at meteologix.com provided by Siemens BLIDS. Clicking on the lightning stroke gives its location, time, type, charge (positive or negative) and power.



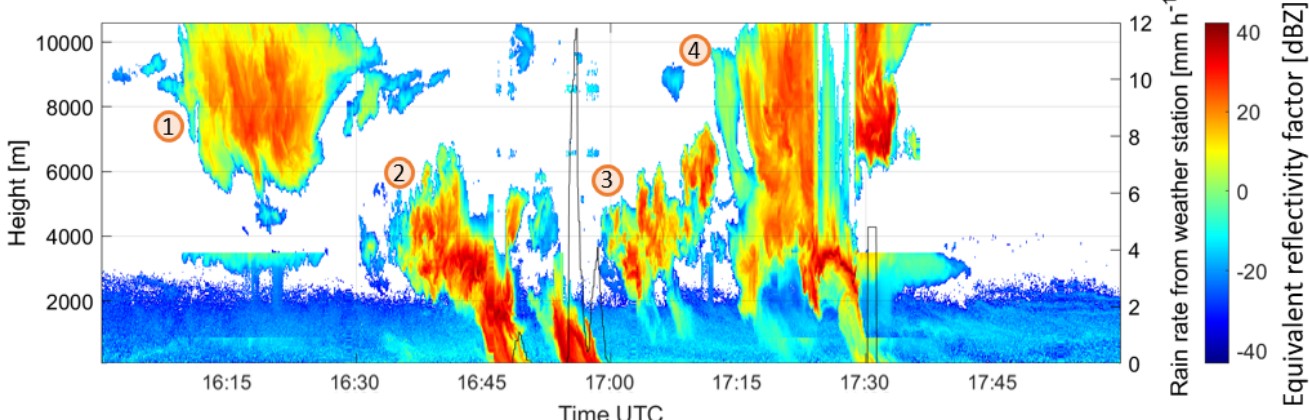

**Figure 2.** Equivalent reflectivity factor on 18 June 2021 16:00-17:59 UTC. Black line shows the rain rate.

The thunderstorm case being studied took place on 18 June 2021 from 16:15 to 17:45 UTC near Cabauw. Four major
thunderstorm clouds (numbered in Fig. 2, Fig. A1, Fig. A2 and Fig. A3) crossed the line of sight of the radar from southwest to
northeast. The equivalent reflectivity factor, $Z_e$, and rain rate from 16:00 to 17:59 UTC are shown in Fig. 2, while $Z_{DR}$, $K_{DP}$,
$SL_{DR}$ and $\rho_{hv}$ are shown in Fig. 3 and Fig. 4. Note that $Z_e$, $SL_{DR}$ and $\rho_{hv}$ are taken directly from the Level 1 files, while
$Z_{DR}$ and $K_{DP}$ are re-calculated from Level 0 files and calibrated.

It is evident from Fig. 2 that due to significant attenuation, the top part of the second and fourth clouds which produced
precipitation that reached the ground are missing. Some artefacts are observed, such as the noise from ground level to 2500
m over the entire period, and the 'ghost' signals between 2500 m and 3500 m from 16:10 to 16:25 UTC and from 17:30 to
17:40 UTC, which are likely due to signals from the top of the cloud being folded into the second chirp. These artefacts are
also present in other variables, thus the data in the second chirp might not be reliable. From Fig. 2, no melting layer with high
$Z_e$ is visible even though the temperature was about 0 °C at around 4000 m, which is likely due to convective mixing.

From Fig. 3(a) and Fig. 4(a), negative $Z_{DR}$ and high $SL_{DR}$ values are observed from 16:42 to 16:48 UTC and from 17:24
to 17:30 UTC, which could be associated to the alignment of particles near lightning. From Fig. 4(b), lower $\rho_{hv}$ values of
0.9 are found also from 16:42 to 16:48 UTC and from 17:24 to 17:30 UTC, which could suggest that there may be a mixture
of hydrometeors in the cloud. However, at those times and locations, the decreasing $\rho_{hv}$ and increasing $SL_{DR}$ values could
be due to a lower signal-to-noise ratio (SNR) because of the attenuated equivalent reflectivity factor, thus caution is required
when interpreting these values. Also the differential reflectivity may be impacted by rain differential attenuation. Therefore,
these times/locations will not be discussed further. Comparing Fig. 3(a) and (b), $Z_{DR}$ and $K_{DP}$ show different patterns in
some areas, such as in the first high cloud and in the top part of the cloud from 17:20 to 17:25 UTC. These will be further
investigated.



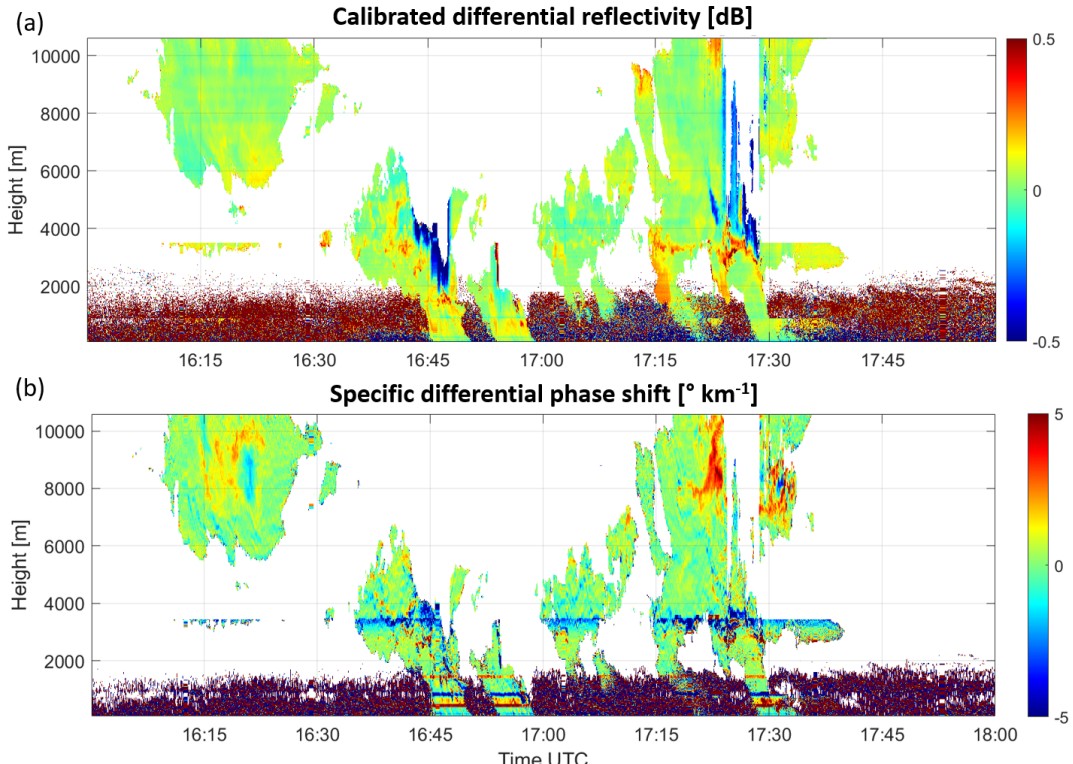

**Figure 3.** (a) $Z_{DR}$ and (b) $K_{DP}$ on 18 June 2021 16:00-17:59 UTC.

Fig. 5 shows the mean vertical velocity, vertical air velocity and Doppler spectrum width during the thunderstorm. The
mean vertical velocity in Fig. 5(a) eliminates the contribution of horizontal wind obtained from ECMWF model data from the
measured mean Doppler velocity. For such a complex system as thunderstorm, this leads to a first approximation of the mean
vertical velocity of hydrometeors. In the first cloud from 16:10 to 16:30 UTC, particles are mainly falling, while in the other
clouds, there are alternate regions where particles are falling and rising. The vertical air velocity is obtained from the smallest
Doppler velocity bin. From Fig. 5(b), vertical air velocity varies a lot within the clouds. There are regions with upward velocity
exceeding 20 m s$^{-1}$, which shows there may be strong updrafts in the thunderstorm clouds. There are also adjacent regions
with upward and downward motion, such as near 16:22 and 17:20 UTC. These may represent convective motion in the clouds.
Figure 5(c) shows that some regions in the clouds have high Doppler spectrum width, such as within the first cloud and near
the top of the fourth cloud. This could mean that there is a wide variety of particles within the radar resolution volume or the
Doppler spectrum is broadened by turbulence.
For a better understanding of the cloud radar data, weather radar images from 16:15 to 17:40 UTC are shown in Fig. A1,
Fig. A2 and Fig. A3 (GmbH). Lightning strikes within the 5 minutes prior to the labelled time are marked by yellow asterisks.
The red triangle shows the cloud radar location and the red ruler shows the line of sight of the cloud radar with each mark equal
to 1 km. Lightning occurred in all four major clouds labelled in Fig. 2. For the first cloud, lightning occurred at least 10 km



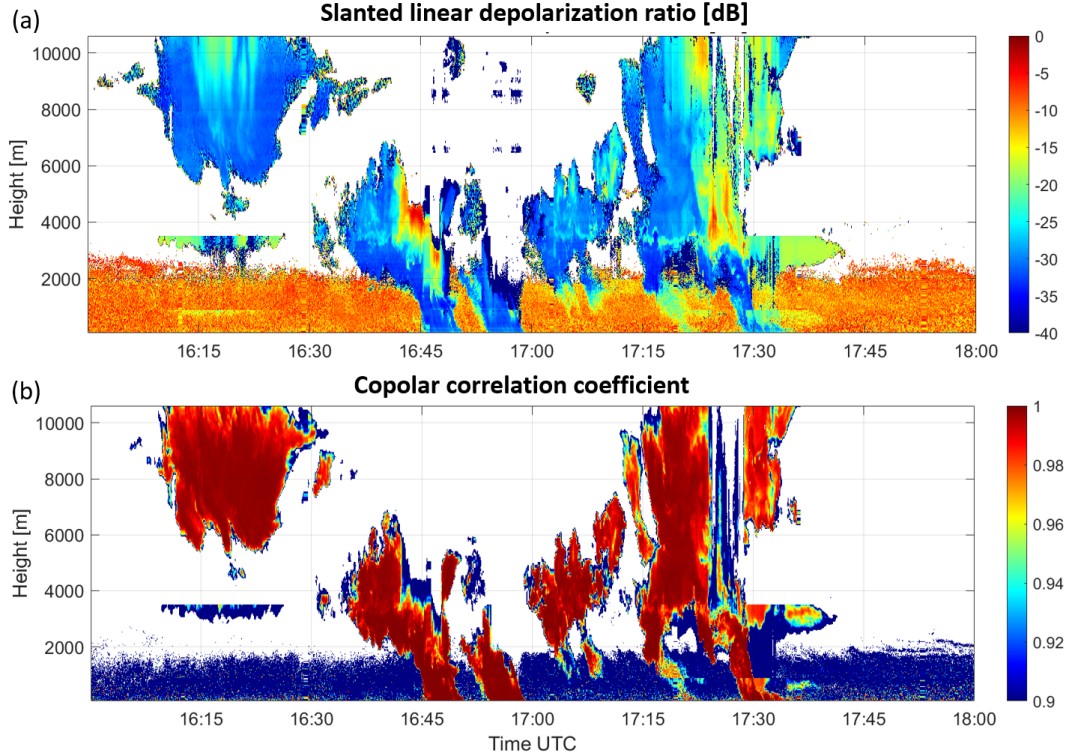

**Figure 4.** (a) $SL_{DR}$ and (b) $\rho_{hv}$ on 18 June 2021 16:00-17:59 UTC.

away from the radar. For the second cloud, lightning occurred at a perpendicular distance of around 3 to 8 km from the radar.
The third cloud only produced two lightning strikes after passing through the line of sight of the radar. The strikes were at a perpendicular distance of around 6 to 7 km from the radar. The fourth cloud produced a large number of lightning strikes from less than 1 km to more than 15 km away from the radar. Lightning was most active from 17:15 to 17:25 UTC, and became less active as the cloud passed through the line of sight of the radar and moved away.

## 3 Methodology

This section explains the steps required to analyse radar data to investigate thunderstorm events. First, the way to compute polarimetric and Doppler variables from raw data is explained in Sect. 3.1. Then, Sect. 3.2 explains how integrated variables and Doppler spectra were used to investigate properties of the thunderstorm clouds. Finally, Sect. 3.3 explains the motivation and method of performing scattering simulations.



**Figure 5.** (a) Mean vertical velocity, (b) vertical air velocity and (c) Doppler spectrum width on 18 June 2021 16:00 - 17:59 UTC from 35 GHz radar with 45° elevation.

## 3.1 Radar variables

### 3.1.1 Polarimetric variables calculation

Spectral polarimetric radar variables which are estimated from the raw data (Level 0) were used in this research. Consequently, most of the integrated radar variables are also computed from Level 0 data.





The integrated $Z_{DR}$ and $\Psi_{DP}$ can be computed by:

$$Z_{DR}(r,t) = 10\log_{10}\left(\frac{\sum_v sZ_{hh}(r,v,t)}{\sum_v sZ_{vv}(r,v,t)}\right) \tag{1}$$

$$\Psi_{DP}(r,t) = \arctan\left(\frac{\sum_v \Im(-sC_{hh,vv}(r,v,t))}{\sum_v \Re(sC_{hh,vv}(r,v,t))}\right) \tag{2}$$

The covariance spectrum $sC_{hh,vv}$ corresponds to the Level 0 array CHVSpec. The minus sign in Eq. (2) is added in order to obtain the right trend for $K_{DP}$ in rain, namely positive at 35 GHz and negative at 94 GHz. Here, $r$ is the range, $v$ is the Doppler velocity and $t$ the time. Only data with signal-to-noise ratio above 10 dB were included in the summations to be consistent with the analysed spectral data.

The spectral differential reflectivity ($sZ_{DR}$) and spectral differential phase shift ($s\Psi_{DP}$) can be computed by:

$$sZ_{DR}(r,v,t) = 10\log_{10}\left(\frac{sZ_{hh}(r,v,t)}{sZ_{vv}(r,v,t)}\right) \tag{3}$$

$$s\Psi_{DP}(r,v,t) = \arctan\left(\frac{-\Im(sC_{hh,vv}(r,v,t))}{\Re(sC_{hh,vv}(r,v,t))}\right) \tag{4}$$

Only the part of the spectra with signal-to-noise ratio above 10 dB were used to exclude the noisy edges of the spectra where values often fluctuate significantly (Yu et al., 2012). In addition, the spectra were smoothed using a 5-point moving average in Doppler bin to reduce noise. For this study, an extra polarimetric calibration was carried out using vertical profiles of precipitation involving high precipitating clouds. This procedure resulted in reducing the expected error associated with $Z_{DR}$ and $\Psi_{DP}$ to 0.05 dB and 0.6° respectively.

The $SL_{DR}$ and $\rho_{hv}$ values were taken from the Level 1 dataset.

The specific differential phase shift ($K_{DP}$) was approximated from the calibrated $\Psi_{DP}$ in degrees in two steps. First, $\Psi_{DP}$ was smoothed using a 5-point moving average in range to reduce noise. Then, $K_{DP}$ was computed by

$$K_{DP}(r,t) = \frac{\Delta\Psi_{DP}(r,t)}{2\Delta r} \ [°\ \text{km}^{-1}], \tag{5}$$

where $\Delta r$ is distance between adjacent range bins in km. Note that this quick estimation of the specific differential phase shift is meant for detecting areas of interest in thunderstorm cloud profiles. For quantitative values of $K_{DP}$, this processing may be too simple when large sized ice particles are present in the thunderstorm cloud and Mie scattering occurs.

### 3.1.2 Doppler variables calculation

The measured Doppler velocity $v$ of a particle, defined negative as the particle approaches the radar, is given by

$$v = (w - V_t)\sin\theta + v_H\cos\theta\cos(D - \pi - \phi), \tag{6}$$

where $w$ is the vertical air velocity, $v_H$ is the horizontal wind speed, $V_t$ is the terminal fall velocity of the particle (positively defined), $\theta$ is the elevation angle of the radar, $D$ is the wind direction relative to North and $\phi$ is the azimuth angle of the radar beam relative to North. The mean Doppler velocity can reflect the average motion of particles in a radar resolution volume





along the line of sight of the radar. To extract it from Level 0 data, the first step is to unfold and dealiase each Doppler spectrum. Then, the mean Doppler velocity ($\overline{v_D}$) can be computed by

$$\overline{v_D}(r,t) = \frac{1}{Z_{hh}(r,t)} \sum_{v_{\text{SNR}>10\,\text{dB}}} v \times sZ_{hh}(r,v,t). \tag{7}$$

The Doppler spectrum width ($\sigma_{v_D}$) can also be computed by

$$\sigma_{v_D}(r,t) = \sqrt{\frac{1}{Z_{hh}(r,t)} \sum_{v_{\text{SNR}>10\,\text{dB}}} (v - \overline{v_D}(r,t))^2 \times sZ_{hh}(r,v,t).} \tag{8}$$

The mean vertical velocity ($\overline{w - V_t}$) can give information about the vertical motion of hydrometeors in thunderstorm clouds. It can be estimated by solving Eq. (6) using the mean Doppler velocity ($\overline{v_D}$) together with $v_H$ and $D$ estimated from the ECMWF model data.

It is also useful to extract the vertical air velocity, which can give information about the updraft and downdraft pattern in thunderstorm clouds. It can be estimated by assuming that the smallest particles in the Doppler spectra are so light that their fall velocity is very close to zero, thus their vertical velocity is equal to the vertical air velocity. Therefore, the first step is to identify the Doppler velocity of the rightmost valid bin of the Doppler spectra with a 10 dB SNR threshold. Then, the vertical air velocity $w$ can be estimated by solving Eq. (6) with $V_t = 0$ and $v_H$ and $D$ estimated from the ECMWF model data.

### 3.2 Analysing radar variables

#### 3.2.1 Analysing integrated variables

Integrated variables were used in this study to identify time instants and ranges where signals related to lightning activities are present. During lightning, the electric field in clouds would align ice crystals vertically, causing $Z_{DR}$ and $K_{DP}$ to become negative. When negative $Z_{DR}$ or $K_{DP}$ is observed in the integrated profile, more in depth analyses were carried out by investigating $sZ_{DR}$ and $s\Psi_{DP}$ at those time instances to understand the causes of those negative values.

Other useful variables may be the linear depolarisation ratio ($L_{DR}$) and the copolar correlation coefficient ($\rho_{hv}$). High $L_{DR}$ values may indicate canting of ice crystals in a specific direction due to cloud electrification (Sokol et al., 2020). Regions with low $\rho_{hv}$ could be regions where graupel and ice crystals co-exist, and they may collide with each other to produce an electric field. However, when SNR is low, $SL_{DR}$ and $\rho_{hv}$ values may become large and low, respectively, regardless of the characteristics of the particles. Therefore, analysis was made at sufficient SNR, which is above 10 dB.

#### 3.2.2 Analysing Doppler spectrum

While integrated variables contain information about all particles within a radar resolution volume, Doppler spectra separate the contributions of particles with different Doppler velocities, hence different sizes or densities. With spectral $Z_{DR}$, it would be possible to identify whether negative $Z_{DR}$ is contributed by small particles that would appear on the right part of the Doppler spectrum, or by large particles that would appear on the left part of the Doppler spectrum. If negative $Z_{DR}$ is observed for small





particles, it is likely that an electric field is present that aligns the small particles. On the other hand, negative $Z_{DR}$ for large particles only may indicate the presence of conical graupel (Lu et al., 2016a). However, the possible transition from Rayleigh to Mie scattering regime may complicate these interpretations of spectral $Z_{DR}$.

The vertical gradient of the differential phase shift ($\Psi_{DP}$) is related to $K_{DP}$. A positive gradient indicates positive $K_{DP}$ and vice versa. With the use of $s\Psi_{DP}$ the Mie scattering regime can be identified. As mentioned before, fluctuations in $sZ_{DR}$

values in the Mie scattering regime makes it difficult to interpret those values. It is therefore crucial to identify when the Mie scattering regime begins. To do this, a useful fact is that the differential phase shift ($\Psi_{DP}$) is the sum of the two-way differential propagation phase ($\Phi_{DP}$) and the differential backscatter phase ($\delta_{co}$). In the Rayleigh scattering regime, spectral differential phase shift should be constant since the electromagnetic wave that scatters from particles at a particular range has propagated through the same set of particles in all previous ranges, and $\delta_{co}$ is zero. This part of the spectrum is often

referred to as the Rayleigh plateau. In the Mie scattering regime, $\delta_{co}$ is non-zero and depends on the particle properties, thus the differential phase shift spectrum is no longer flat. Therefore, the Mie scattering regime begins when the left part of the differential phase shift spectrum starts to increase or decrease. The effect of noise may sometimes affect the identification of the Mie scattering regime. It is useful to know that the maximum or minimum of spectral $\Psi_{DP}$ are often aligned with the maximum or minimum of spectral $Z_{DR}$. Thus, if the maxima or minima of $s\Psi_{DP}$ and $sZ_{DR}$ are aligned, one can be more

confident that the fluctuations observed are due to Mie scattering instead of noise.

The left column of Fig. 6 shows an example where the Mie scattering regime can be clearly identified using $s\Psi_{DP}$. The Rayleigh plateau is found from $-1$ to $3$ m s$^{-1}$, while Mie scattering occurs at Doppler velocity smaller than $-1$ m s$^{-1}$ since $s\delta_{co}$ becomes non-zero. $sZ_{DR}$ follows a similar trend, which strengthens the proof that Mie scattering occurs. However, some cases can be more tricky, such as the one shown in the right column of Fig. 6. Here, the Rayleigh plateau ends at about $-0.5$

m s$^{-1}$, while $sZ_{DR}$ only begins to decrease at about $-4$ m s$^{-1}$. To understand this better, scattering simulations are needed, which is discussed next.

### 3.3   Scattering simulations

Studying the Doppler spectrum of $Z_{DR}$ is challenging when Mie scattering is involved. This is because $sZ_{DR}$ values fluctuate in the Mie scattering regime, so it will become difficult to determine whether the fluctuations in the observed $Z_{DR}$ spectrum

are due to changes in shape/density of hydrometeors or Mie scattering. Therefore, scattering simulations were carried out to understand how Mie scattering affects the $Z_{DR}$ spectrum using the python code pyTmatrix (Waterman, 1965; Leinonen, 2014). The code is based on the T-matrix method (Waterman, 1965), which is a numerical model of electromagnetic and light scattering by non-spherical particles with sizes comparable to the wavelength of the incident radiation. The code supports simulations of spheroids or cylinders. The scattering matrix of a scatterer depends on several parameters, including the axis

ratio, ice fraction and canting angle. The axis ratio is defined as the length along the scatterer's rotational axis to its width perpendicular to this axis. It is smaller than one for oblate particles and larger than one for prolate particles. Ice fraction ($f_i$) characterizes how much ice and air a scatterer is composed of, which affects the density of the particle. A value of 1 means pure ice, while a value of 0 means pure air. Ice fraction affects the complex effective relative permittivity of the scatterer ($\varepsilon_{eff}$).



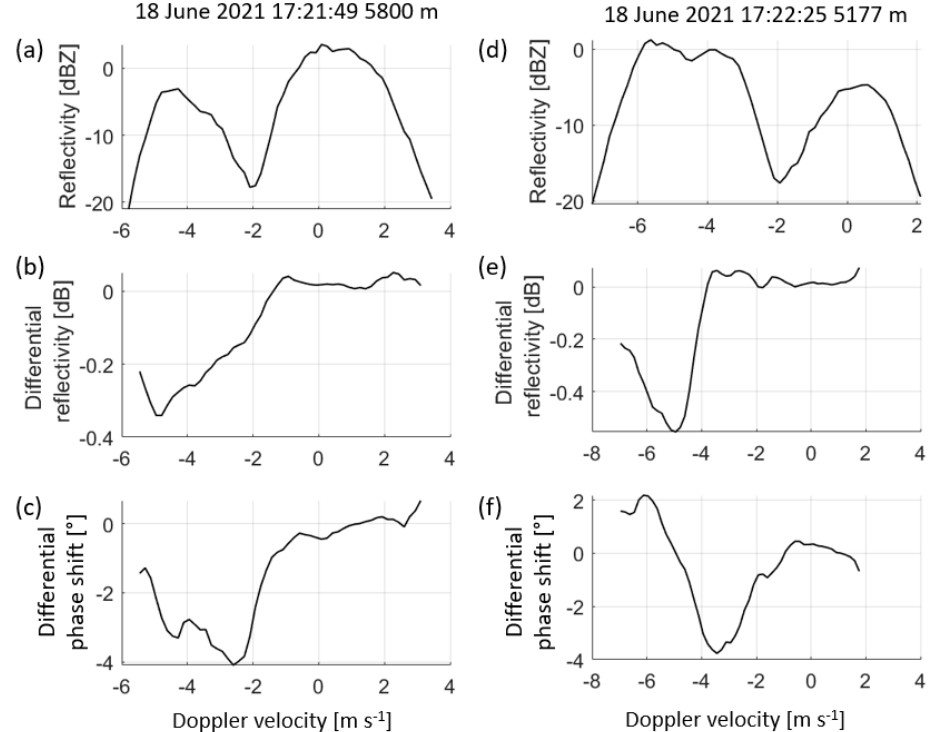

**Figure 6.** Two examples of Doppler spectra of $Z_e$, $Z_{DR}$ and $\Psi_{DP}$ at 35 GHz showing Mie scattering.

One approximation is given by the Maxwell-Garnett formula:

$$\frac{\varepsilon_{eff} - 1}{\varepsilon_{eff} + 2} = f_i \cdot \frac{\varepsilon_i - 1}{\varepsilon_i + 2}, \tag{9}$$

where $\varepsilon_i$ is the complex relative permittivity of ice. The value of $\varepsilon_i$ is $3.19015 + 0.00285i$ at 35 GHz at 266 K, and the temperature dependence is small for the part of the spectrum from ultraviolet (175 nm) to the microwave (1 cm) (Warren and Brandt, 2008). The complex effective refractive index of the scatterer ($m_{eff}$), which is a parameter that can be specified in the simulation code, can then be determined using

$$\varepsilon_{eff} = m_{eff}^2. \tag{10}$$

The canting angle refers to the Euler angle $\beta$ of the scatterer defined in Fig. 7.

In the simulation, a scatterer object in the shape of a spheroid was defined, and the backscatter radar reflectivity ($Z_e$), differential reflectivity ($Z_{DR}$) and differential backscatter phase ($\delta_{co}$) at 35 GHz, with 45° looking angle were retrieved. In the first experiment, the axis ratio of scatterers with zero mean canting angle was varied from 0.1 to 1.2, which covers the axis ratio range of plates, dendrites, aggregates and graupel. The ice fraction was fixed at 0.6, which is the average ice fraction of different types of particles. In the second experiment, the ice fraction of scatterers with zero mean canting was varied from 0.2



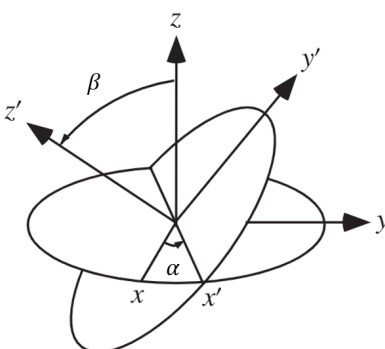

**Figure 7.** Definition of Euler angles $\alpha$ and $\beta$.

to 1, which covers the ice fraction range of plates, dendrites, aggregates and graupel. Simulations for both oblate and prolate particles were carried out, with an axis ratio of 0.8 or 1.2. In the third experiment, the canting angle was varied from 0° to 90°. Three sets of simulations were carried out to simulate different types of particles, including plates (axis ratio = 0.1, ice fraction

= 0.98), aggregates (axis ratio = 0.8, ice fraction = 0.3) and graupel (axis ratio = 1.2, ice fraction = 0.6). For all simulations, the canting angles of the scatterers follow a Gaussian distribution with a standard deviation of 0.1°. The Euler angle $\alpha$ of the scatterers (see Fig. 7) follows a uniform distribution from 0 to 360°.

Note that the T-matrix method (Leinonen, 2014) offers flexibility for simulating the radar spectral variables by varying different input parameters (axis ratio, ice fraction, Euler angles) for a first examination of trends at 35 GHz. Nonetheless, this

method has limitations as it assumes that ice particles are spheroidal and have a fixed ice fraction or density. It ignores the non-homogeneity of ice particles, especially aggregates, which may result in a bias in the spectral polarimetric variables when frequency increases. This is another reason to carry out this study of thunderstorm clouds at 35 GHz but not at 94 GHz.

## 4 Scattering simulation results

This section gives an overview of the dependencies of spectral polarimetric radar variables of particles, $sZ_{hh}$, $sZ_{DR}$ and $s\delta_{co}$,

on axis ratio, ice fraction and canting angle in the Rayleigh and Mie scattering regimes based on scattering simulations.

### 4.1 Axis ratio

Figure 8 shows the simulation results for horizontally aligned scatterers with ice fraction 0.6 with different axis ratios at 35 GHz. The radius refers to the maximum radius of the spheroid, i.e. half the length of its long axis. From Fig. 8(a), the first Mie minimum occurs at a maximum radius of around 2 mm for axis ratio 1.2, 2.6 mm for axis ratio 0.8, and 3.2 for axis ratio 0.4.

Therefore, for oblate spheroidal particles, the position of the first Mie minimum goes towards larger radius when axis ratio decreases.




From Fig. 8(b), in the Rayleigh scattering regime, $Z_{DR}$ decreases with increasing axis ratio, with positive values for oblate spheroids (axis ratio < 1) and negative values for prolate spheroids (axis ratio > 1). When entering the Mie scattering regime, $Z_{DR}$ of oblate particles increases slightly, while that of prolate particles decreases. At the first Mie minimum, particles with axis ratio 0.1, 0.4 and 1.2 give a trough in $Z_{DR}$, but those with axis ratio 0.8 give a peak. In addition, the lines for different axis ratios cross over each other in the graph of $Z_{DR}$, meaning that the trend between $Z_{DR}$ and axis ratio depends on particle size. From Fig. 8(c), $\delta_{co}$ of oblate particles increases when entering the Mie scattering regime and gives a peak at the first Mie minimum, while that of prolate particles decreases and gives a trough.

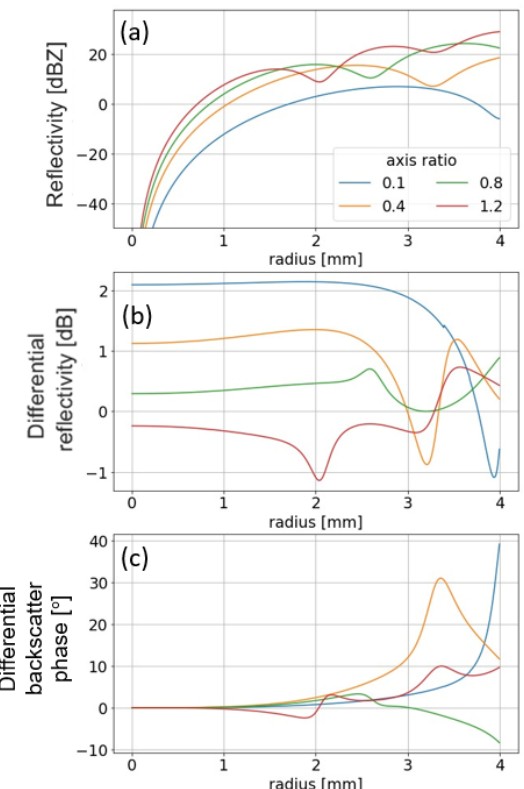

**Figure 8.** Simulated (a) radar reflectivity, (b) differential reflectivity and (c) differential backscatter phase for horizontally aligned scatterers with different axis ratios as a function of maximum radius at 35 GHz with 45° looking angle. All scatterers have ice fraction of 0.6.

## 4.2 Ice fraction

Figure 9 shows two sets of simulations for horizontally aligned scatterers with different ice fractions. In the Rayleigh scattering regime, the magnitude of $Z_{DR}$ increases with increasing ice fraction. The first extremum of $Z_{DR}$ is attained at a smaller size for scatterers with higher ice fraction. For low ice fraction (0.2, 0.4 and 0.6), the sign of $Z_{DR}$ does not change after entering the Mie scattering regime (except for radius larger than 3.2 mm for scatterers with axis ratio 1.2 and ice fraction 0.6). When ice




fraction is large (0.8 and 1), the sign of $Z_{DR}$ flips soon after reaching the first extremum, and the trend is rather unpredictable.
For a radius of larger than 2.5 mm representing large aggregates such as graupel, significant negative (positive) values could
be obtained, which increases the interpretation challenge. The differential backscatter phase initially increases (decreases) for
oblate (prolate) particles when entering the Mie scattering regime. The sign reverses afterwards, and the trend becomes less
predictable especially if ice fraction is high.

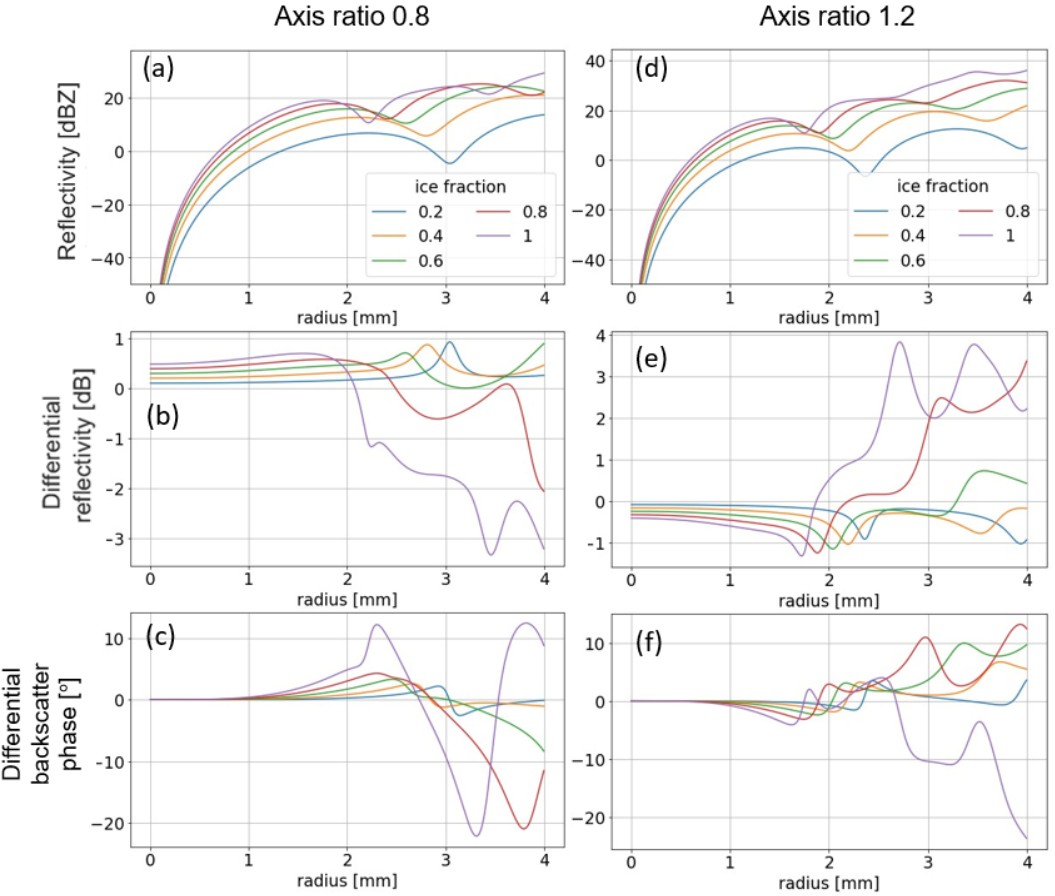

**Figure 9.** Simulated radar variables for horizontally aligned scatterers with different ice fractions as a function of maximum radius at 35
GHz with 45° looking angle. (a-c) show the radar reflectivity, differential reflectivity and differential backscatter phase for scatterers with
fixed axis ratio of 0.8. (d-f) show the same for scatterers with fixed axis ratio of 1.2.

### 4.3    Canting angle

Figure 10 shows three sets of simulations for scatterers with different canting angles. For oblate particles (left and middle
columns), $Z_{DR}$ in the Rayleigh scattering regime is negative when the canting angle becomes larger than 45°. One can un-
derstand this as the effective axis ratio of an oblate scatterer getting larger than one when it becomes vertically aligned. The





opposite is true for prolate particles. However, in the Mie scattering regime, the relationship between the sign of $Z_{DR}$ and

the canting angle is not trivial. For scatterers similar to plates with axis ratio 0.1 and ice fraction 0.98, the first extremum of

$Z_{DR}$ is positive for $\beta = 90°$ but negative for $\beta = 0°$. There is no sharp extremum for $\beta = 30°$ or $60°$. For scatterers similar

to conical graupel with axis ratio 1.2 and ice fraction 0.6, the sign of $Z_{DR}$ also changes when particle size becomes larger.

The differential backscatter phase does not have a trend that can be easily summarised for different canting angles for all three

cases.

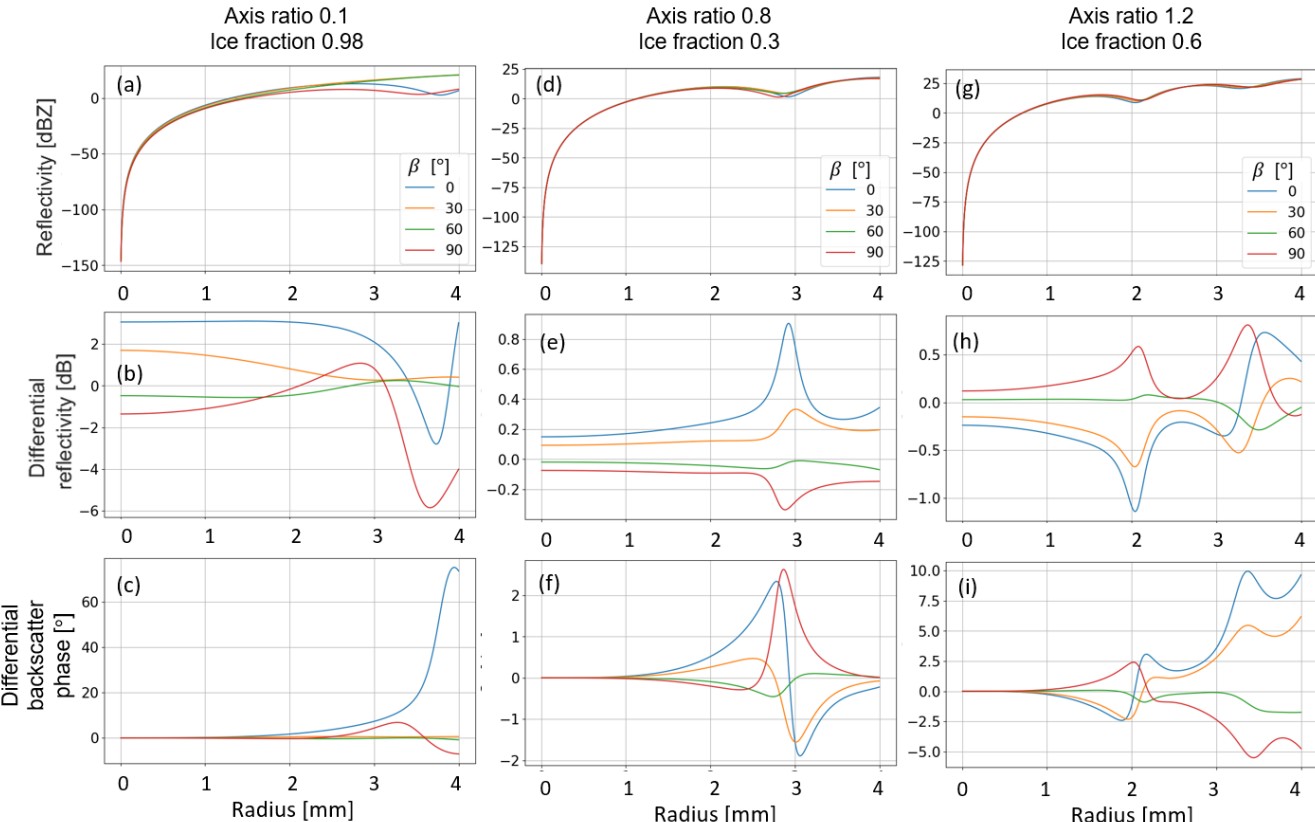

**Figure 10.** Simulated radar variables for horizontally aligned scatterers with different canting angles as a function of maximum radius at 35 GHz with 45° looking angle. (a-c) show the radar reflectivity, differential reflectivity and differential backscatter phase for scatterers similar to plates with fixed axis ratio of 0.1 and ice fraction of 0.98. (d-f) show the same for scatterers similar to aggregates with fixed axis ratio of 0.8 and ice fraction of 0.3. (g-i) show the same for scatterers similar to conical graupel with fixed axis ratio of 1.2 and ice fraction of 0.6.

## 4.4   Summary

In this section, the effects of axis ratio, ice fraction and canting angle of scatterers on $Z_{DR}$ and $\delta_{co}$ are investigated. Table 2

summarises the key trends of $Z_{DR}$ in the Rayleigh scattering regime and the trend of $\delta_{co}$ before the first Mie minimum for

horizontally aligned scatterers with different axis ratios and ice fractions. Changing the canting angle has similar effect as




altering the axis ratio of the scatterers in terms of the initial trend of $Z_{DR}$. In general, the sign of $Z_{DR}$ is the same as the sign of $\delta_{co}$ before the first Mie minimum. However in some cases, $\delta_{co}$ shows a sign inversion at the first Mie minimum. The

fluctuations of $Z_{DR}$ and $\delta_{co}$ after the first Mie minimum are difficult to predict and often involve sign changes. The most unpredictable behaviours are found when ice fraction is high.

**Table 2.** $Z_{DR}$ characteristics in Rayleigh scattering regime and trend of $\delta_{co}$ before first Mie minimum

| | $Z_{DR}$ in Rayleigh scattering regime | $\delta_{co}$ trend before first Mie minimum for horizontally aligned scatterers |
|---|---|---|
| Axis ratio < 1 | positive, increase with decreasing axis ratio | increase |
| Axis ratio > 1 | negative, more negative with increasing axis ratio | decrease |
| Ice fraction | magnitude increases with increasing ice fraction | same trend as $Z_{DR}$ except for large ice fraction |

From this first analysis, our investigation of spectral polarimetric variables in thunderstorm clouds will start by identifying the Rayleigh scattering part of the spectrum using the measurement of the spectral differential phase. In the Rayleigh scattering regime, the spectral differential backscatter phase is zero and the spectral differential phase equals the spectral differential

propagation phase. This will prevent the misinterpretation of variations in spectral differential reflectivity caused by Mie scattering. Next, focus will be given to the $sZ_{DR}$ signature in the Rayleigh scattering regime. Afterwards, analysis can be carried out considering both $sZ_{DR}$ and $s\delta_{co}$ in the Mie scattering regime at least until the first Mie minimum. Concerning second extrema, they are expected to be challenging to interpret and to measure, the latter in terms of signal-to-noise ratio at high altitudes.

**5 Case analysis**

This section discusses interesting observations in the thunderstorm event on 18 June 2021 from 16:15 to 17:45 UTC near Cabauw. Focus has been given to the first and the fourth cloud that passed through the line of sight of the radar. The second cloud was not investigated as the radar suffered from significant attenuation due to the precipitation, while the third cloud was not studied as it only had two lightning strokes after it passed through the line of sight of the radar.

**5.1 First cloud**

The first cloud came within the sight of the radar from 16:10 to 16:30 UTC. The centre of the cloud that contained lightning activities was more than 10 km away from the radar, thus the radar could only see the edge of the cloud.





**Figure 11.** (a) Differential reflectivity, (b) specific differential phase shift and (c) vertical air velocity of the first thunderstorm cloud on 18 June 2021 from 16:09 to 16:30 UTC.



### 5.1.1 Alignment of particles

From Fig. 11(a) and (b), interesting polarimetric signatures can be found in the cloud. As shown in Fig. 11(a), the $Z_{DR}$ values
in the cloud is close to zero and do not show much variations. However, from Fig. 11(b), there is a cluster of negative $K_{DP}$
values between 7600 m and 9300 m, which may indicate the alignment of non-spherical small ice particles but not large ones.
If the small ice particles have sufficient number concentration, $K_{DP}$ would become negative. However, since $Z_{DR}$ carries
reflectivity weighting, large ice particles that do not align with the electric field influence $Z_{DR}$ strongly. This could explain
why $Z_{DR}$ does not show significant negative values.

From Fig. 11(c), the first cloud mainly shows downdrafts from 16:15 to 16:18 UTC and after 16:22 UTC. Referring to Fig.
A1, in these periods, the radar was looking at the edge of the thunderstorm cloud. Therefore, the radar did not see regions with
strong updrafts which is normally found in the core of thunderstorm clouds, but observed downdrafts outside the core instead.
From 16:18 to 16:22 UTC, updrafts of up to 12 m s$^{-1}$ are observed, which could be because the core of the thunderstorm cloud
is closer to the line of sight of the radar. The estimated vertical air velocity is not uniform within the cloud, which suggests that
there might be a lot of turbulence.

Figure 12 shows the spectral $Z_{DR}$ across the period when negative $K_{DP}$ is observed. At 16:18:59 UTC, the right part of the
spectrum, which corresponds to small ice particles, has positive $sZ_{DR}$, suggesting that the particles are horizontally aligned.
However, at 16:21:05 UTC, the right part of the spectrum becomes slightly negative, suggesting that small ice particles are
vertically aligned. At 16:22:34 UTC, $sZ_{DR}$ of the right part of the spectrum becomes positive again, which suggests that the
particles return to being horizontally aligned. Figure 13 shows the mean $sZ_{DR}$ of the lightest 10% of the particles in each
radar resolution volume at the three time instants. It is clear that from 7000 m to 9000 m, $sZ_{DR}$ of the lightest 10% particles
are positive at 16:18:59 UTC and 16:22:34 UTC, and is negative at 16:21:05 UTC. The question is: are these negative $sZ_{DR}$
values associated with cloud electrification before lightning?

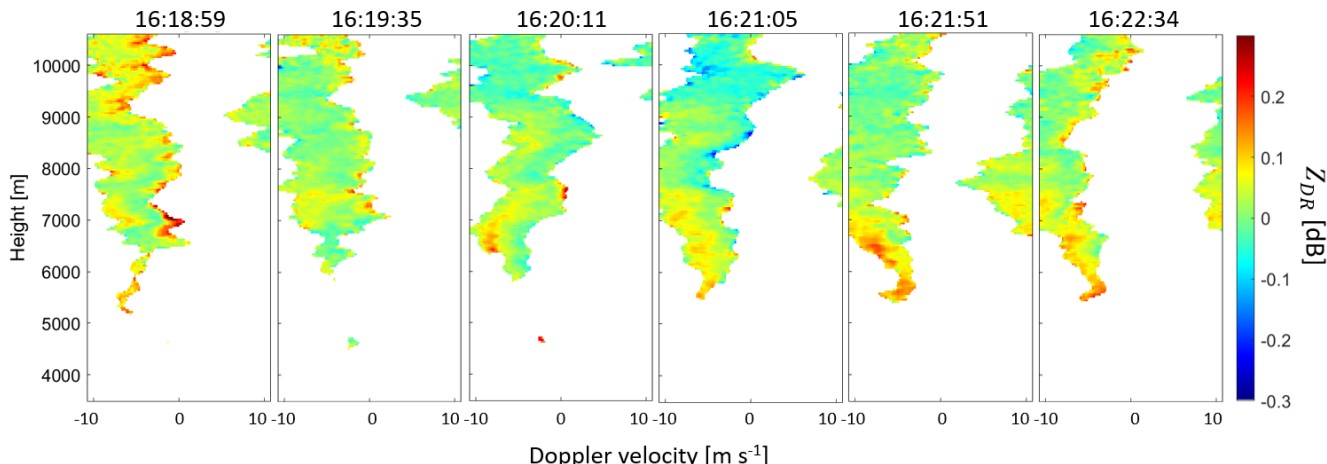

**Figure 12.** Spectral $Z_{DR}$ on 18 June 2021 from 16:18:59 to 16:22:34 UTC.





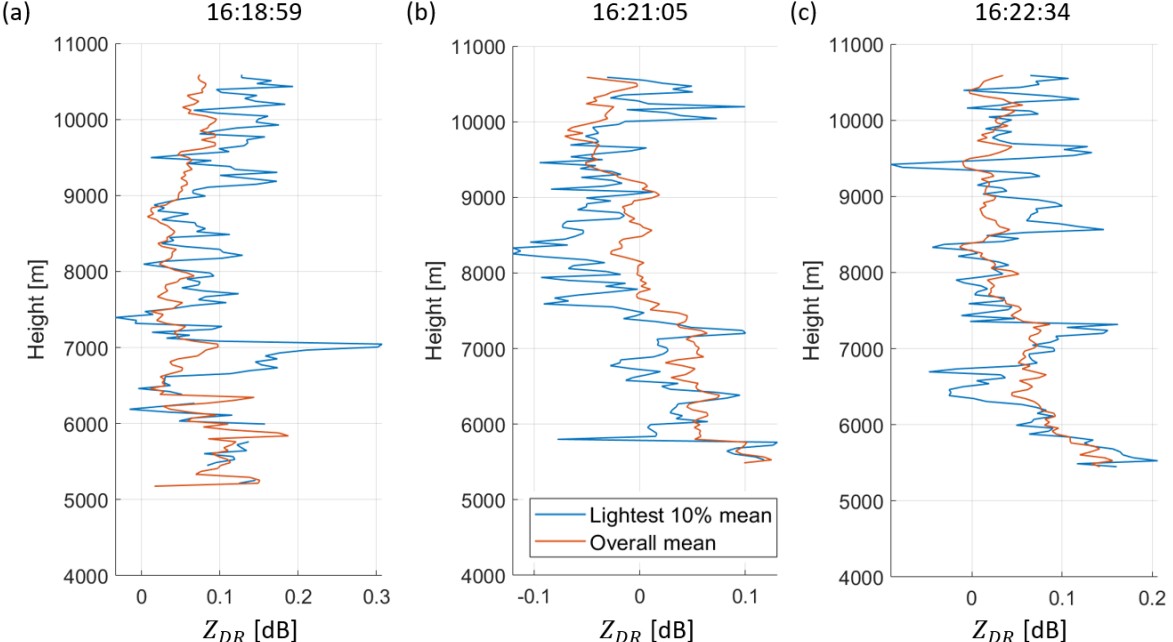

**Figure 13.** Mean $sZ_{DR}$ of all particles and the lightest 10% of the particles in a radar resolution volume at (a) 18 June 2021 16:18:59 UTC, (b) 16:21:05 UTC and (c) 16:22:34 UTC.

Our expectation is that particles align vertically before a lightning stroke, and return to horizontal alignment afterwards. However, the lightning strokes closest to the line of sight of the radar occurred at 16:20:17, 16:21:50 and 16:22:20 UTC (strokes number 9, 11, 14-17 in Fig. B2), but negative $K_{DP}$ is observed continuously from 16:20:11 to 16:21:37 UTC. Negative $K_{DP}$ appears at a distance of 7600 m to 9300 m away from the radar, but the lightning strokes occurred at least 13 km away from the radar. If the electric field that caused these lightning strokes is responsible for the alignment of particles observed, one would expect to observe negative $K_{DP}$ also for ranges beyond 9000 m. Making a closer inspection with spectral $Z_{DR}$, negative $sZ_{DR}$ smaller than $-0.1$ dB are found beyond 9000 m from 16:20:21 to 16:21:15 UTC (Fig. 16(b)), though more negative $sZ_{DR}$ are found on the left side of the spectrum that corresponds to large particles instead of the right side as expected (e.g. 16:21:01 UTC in Fig. 16(b)).

The first inquiry is whether this could be caused by wind shear flipping the Doppler spectrum such that lighter particles appear on the left side. The horizontal and vertical velocities ($V_h$ and $V_v$) of particles are given by (Wang et al., 2019):

$$V_h = v_H + \frac{sV_t^2}{g}, \tag{11}$$

$$V_v = -V_t + w \tag{12}$$

where $v_H$ is the horizontal wind speed, $w$ is the vertical wind, $s = \frac{dv_H}{dz}$ is the constant vertical wind shear, $g$ is gravitational acceleration and $V_t$ is the terminal velocity of the particle which depends on its size. For a radar looking at an elevation angle



$\theta$ and azimuth $\phi$, the Doppler velocity is $V_v \sin\theta + V_h \cos\theta \cos(D - \pi - \phi)$. Without vertical wind shear, the Doppler spectrum

will be shifted as a whole by $v_H$ and $w$, with light particles remaining on the right side of the spectrum. If $s$ is negative, since $V_t$ increases with particle size, the left side of the spectrum would shift to the left more than the left shift of the right side of the spectrum, thus the spectrum widens (Fig. 14(b)). On the other hand, if $s$ is positive, the left side of the spectrum would shift to the right more than the right side of the spectrum (Fig. 14(c)). If the right shift of the left side of the spectrum due to the term $\frac{sV_t^2}{g}$ is larger than the original spectrum width, the spectrum could be flipped (Fig. 14(d)). Assuming a spectrum width of 10 m

s$^{-1}$ and taking the upper bound of the terminal velocity of plates, i.e. $V_t = 2$ m s$^{-1}$ (Spek et al., 2008), for a radar looking at 45°, the vertical wind shear required to flip the Doppler spectrum is approximately 35 m s$^{-1}$ m$^{-1}$. However, according to the ECMWF Integrated Forecast System output over Cabauw (O'Connor, 2022) shown in Fig. 15(c), this is much larger than the strongest vertical wind shear of 4 m s$^{-1}$ km$^{-1}$ from 7500 m to 10000 m. Therefore, it is unlikely that the negative $sZ_{DR}$ on the left side of the spectrum is due to flipping of the spectrum caused by wind shear.

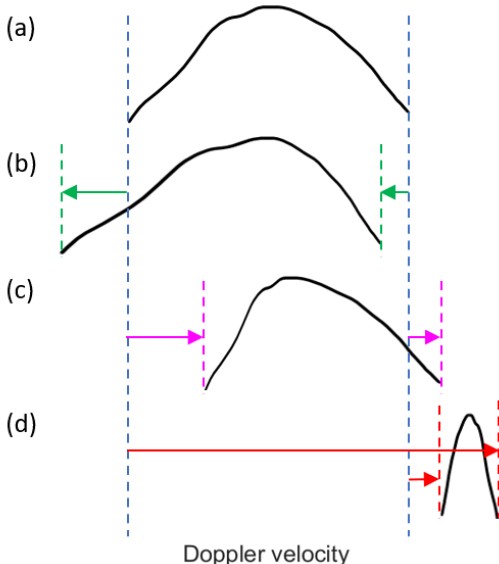

**Figure 14.** A figure to illustrate effects of the sign of vertical wind shear $s$ on the Doppler spectrum: (a) Doppler spectrum when there is no shear (b) Doppler spectrum widens when $s$ is negative (c) Doppler spectrum may become narrow when $s$ is positive (d) Doppler spectrum may flip when $s$ is positive and $\frac{sV_t^2}{g}$ is larger than the original spectrum width.

If the negative $sZ_{DR}$ values are caused by vertical alignment of particles by the electric fields, one possible explanation for this is that the axis ratios of small particles are close to one, thus their $sZ_{DR}$ is close to zero. Meanwhile, large ice chains may be formed under an electric field (Connolly et al., 2005). $sZ_{DR}$ could become negative if these chains are vertically aligned. Nonetheless, the most negative $sZ_{DR}$ is found at 16:21:01 UTC, which is not exactly the moment before any lightning strokes. This could be because the region with chains and sufficiently strong electric field had moved away from the line of sight of the

radar when lightning occurred, or that the electric field was not strong enough to trigger lightning.



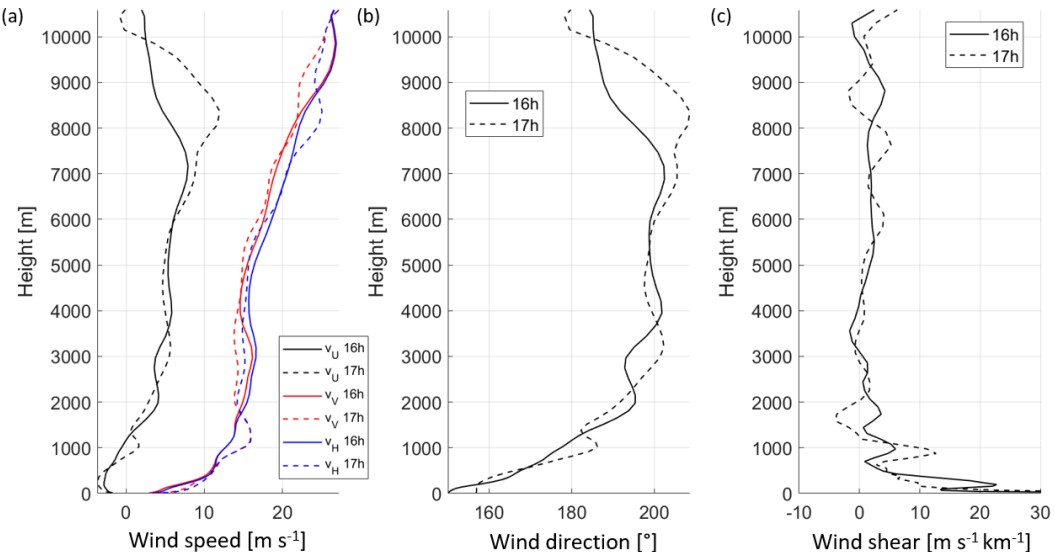

**Figure 15.** (a) Mean horizontal wind, (b) horizontal wind direction relative to North and (c) vertical wind shear at 18 June 2021 16:00 and 17:00 UTC.

The negative $sZ_{DR}$ values on the right edge of the spectra at around 7500 m to 9000 m also do not occur right before any lightning strokes. Although there is a lightning at a perpendicular distance of 13 km away from the radar at 16:21:09 UTC (stroke number 5 in Fig. B2), from the literature, electric field in thunderstorm clouds reduces significantly over 5 km (Merceret et al., 2008), thus it is unlikely that the alignment of particles observed is caused by the build up of electric field that

generated the lightning stroke. There is another lightning at a perpendicular distance of around 11 km away from the radar at 16:21:50 UTC (stroke number 7 in Fig. B2). It is possible that the electric field associated with this stroke caused the vertical alignment of particles observed, but the relationship is hard to prove since the spatial and temporal variation of the electric field is unknown. The next lightning at 16:29:08 UTC at a perpendicular distance of around 11.5 km (stroke number 8 in Fig. B3) occurred 8 minutes after negative $sZ_{DR}$ is observed. This is a rather long period of time compared to common durations

of charging cycles (Gunn, 1954; Marshall and Winn, 1982), thus it is unlikely that it is associated with the observed negative $sZ_{DR}$.

Meanwhile, one should not rule out other possible causes of vertical alignment of particles. As suggested by Brussaard (1976), canting of particles may occur when there is vertical wind shear. He derived the canting angle of particles due to vertical wind shear, i.e. difference in horizontal wind speed in vertical direction, by assuming that the mean orientation of their

rotational symmetric axes are always parallel to the direction of the airflow around them. If there is no updraft and the particle is falling at its terminal velocity, for a linear wind profile, the canting angle ($\gamma$) of the particle is given by

$$\tan\gamma = -\frac{sV_t}{g} \tag{13}$$





To estimate the largest possible canting angle in the current cloud, terminal velocity $V_t$ is taken to be 2 m s$^{-1}$ (Spek et al., 2008) and $s = \frac{dv_H}{dz} = 4$ m s$^{-1}$ according to the ECMWF Integrated Forecast System output (O'Connor, 2022) shown in Fig.

15(c). Using Eq. (13), the corresponding canting angle is 0.05°, which is insignificant. Therefore, wind shear is not likely the cause of negative $sZ_{DR}$ in this case. On the other hand, it is found by Cho et al. (1981) that turbulence is unable to destroy the preferred orientation of falling ice crystals in cumulonimbus clouds.

In conclusion, the vertical alignment of particles observed in the first cloud could be due to electric field, though the electric field may not be strong enough to trigger lightning, or there are lightnings that are not measured by the lightning sensor.

**5.1.2 Supercooled liquid water**

Another interesting feature observed in this cloud is the possible presence of supercooled liquid water. From 16:20:21 to 16:21:15 UTC, spectograms of reflectivity show a separate mode of particles on the right side of the spectrum at around 6000 m (see Fig. 16(a)). From Fig. 16(b), $sZ_{DR}$ of this mode of particles is close to zero. Figure 17 shows the time series of spectral reflectivity and spectral $Z_{DR}$ at 5916 m. A small peak at Doppler velocity of around $-4$ to $-3$ m s$^{-1}$ is consistently present.

The $sZ_{DR}$ of this mode of particles is lower than the left part of the spectrum, with values of around $-0.1$ to 0 dB. By manually identifying the part of the Doppler spectrum that may contain supercooled liquid water for 139 range bins over 16 time steps, it was found that the average $sZ_{DR}$ is $-0.0370$ dB. Since the error of $Z_{DR}$ after calibration is 0.05 dB and small supercooled liquid water droplets are nearly spherical and have a differential reflectivity of 0 dB, there is a high chance that supercooled liquid water is indeed present in the cloud. This is further supported by the liquid water path measured by the cloud radar with a passive channel that has the same looking direction as the radar. From 16:20:21 to 16:21:15 UTC (marked by the red lines in Fig. 18), there is indeed a peak in liquid water path, which agrees with the hypothesis that supercooled liquid water may be present in the cloud. Supercooled liquid water plays a role in the non-inductive charging mechanism as it is needed for riming to occur, which in turn forms graupel that collide with ice crystals to produce charges. Nonetheless, the radar was not able to look at the lower part of the cloud, thus it is unknown whether graupel is formed in this case.



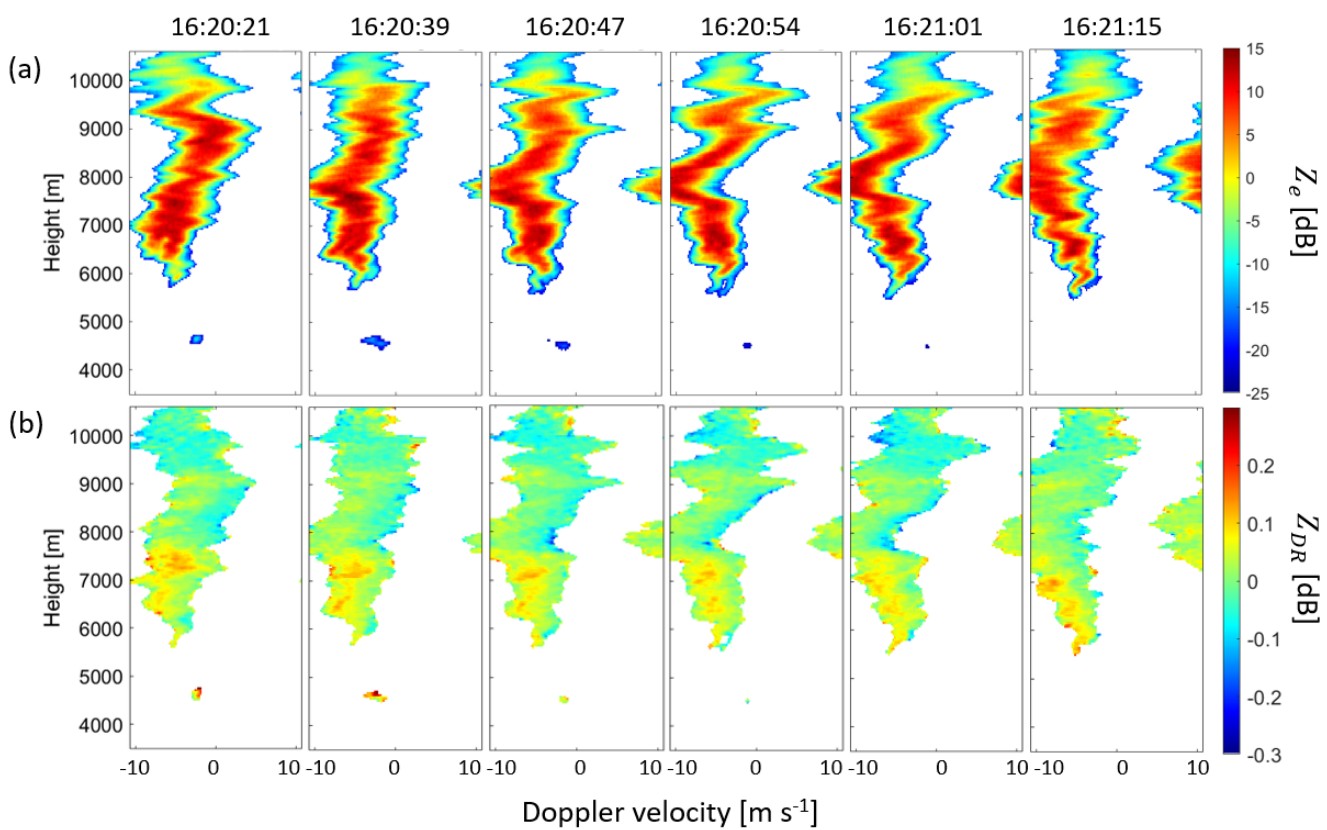

**Figure 16.** (a) Spectral reflectivity and (b) spectral $Z_{DR}$ on 18 June 2021 from 16:20:21 to 16:21:15 UTC showing presence of supercooled liquid water near 6000 m.

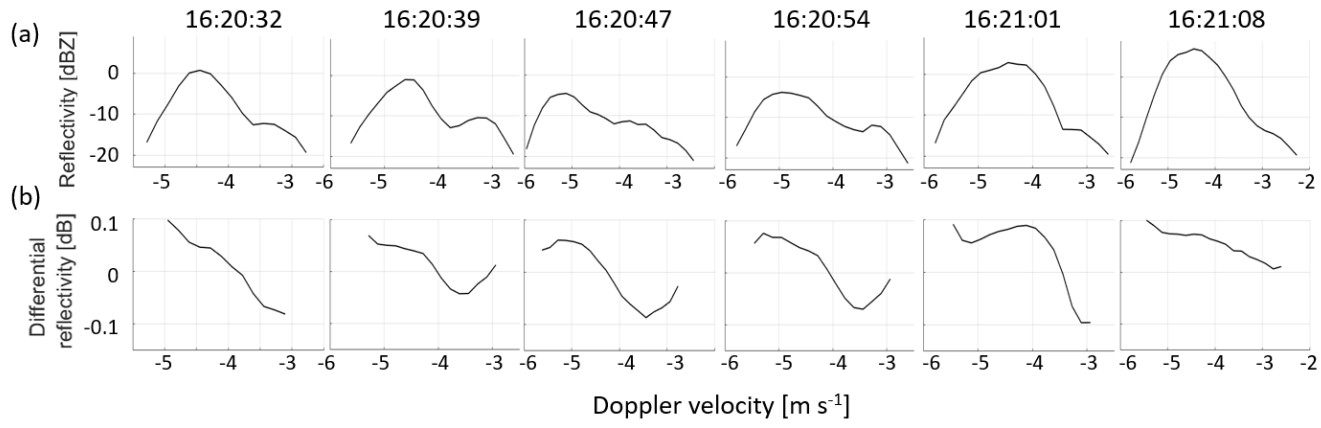

**Figure 17.** (a) Spectral reflectivity and (b) spectral $Z_{DR}$ on 18 June 2021 from 16:20:32 to 16:21:08 UTC at 5916 m.





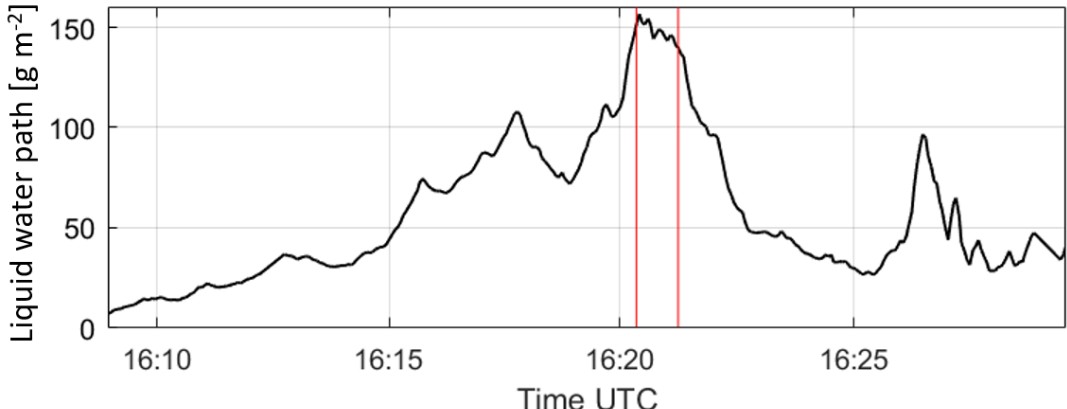

**Figure 18.** Liquid water path of the first cloud measured at 94 GHz. Time interval from 16:20:21 to 16:21:15 UTC is marked by red lines.

## 5.2 Fourth cloud

The fourth cloud came within the sight of the radar from 17:15 to 17:40 UTC. The part of the cloud that passed through the line of sight of the radar from 17:15 to 17:20 UTC did not contain active lightning activities. From 17:20 to 17:35 UTC, the part of the cloud with the most active lightning activities passed through the line of sight of the radar. Afterwards, lightning activities ceased and the cloud moved away from the line of sight of the radar. For an overview of the cloud including the radar

images showing its motion, see Appendix A.





**Figure 19.** (a) Differential reflectivity, (b) specific differential phase shift, (c) slanted linear depolarisation ratio and (d) copolar correlation coefficient of the fourth thunderstorm cloud on 18 June 2021 from 17:14 to 17:26 UTC. Vertical black lines indicate time instants 17:20:26, 17:21:31, 17:22:25, 17:22:57 and 17:23:47 UTC.



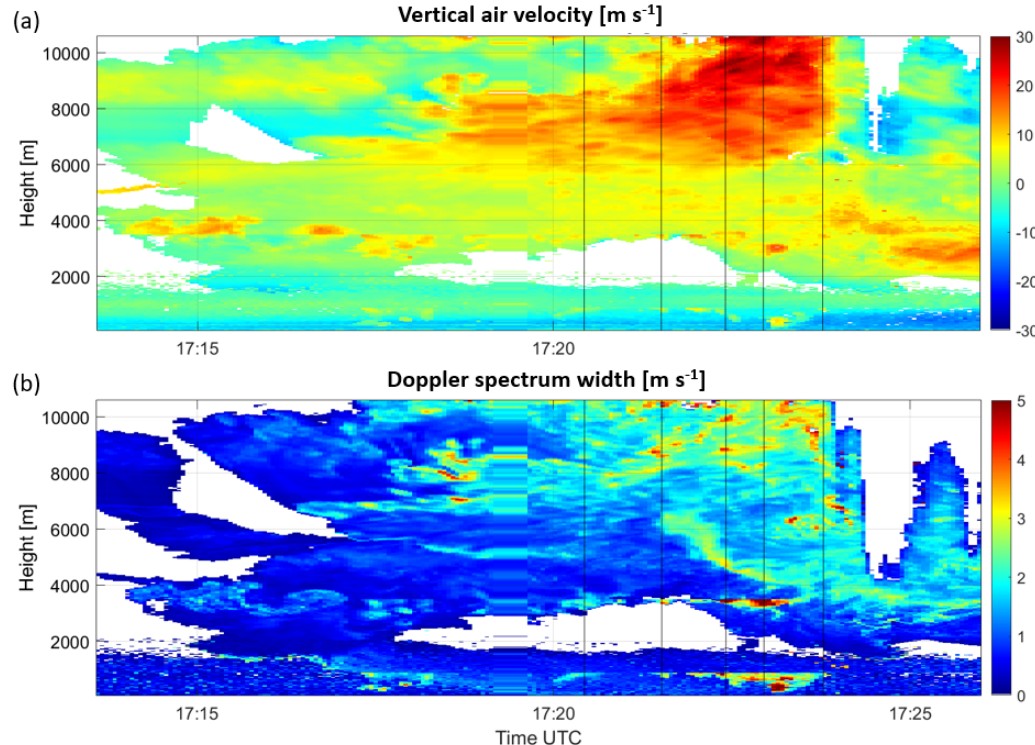

**Figure 20.** (a) Vertical air velocity and (b) Doppler spectrum width of the fourth thunderstorm cloud on 18 June 2021 from 17:14 to 17:26 UTC. Vertical black lines indicate time instants 17:20:26, 17:21:31, 17:22:25, 17:22:57 and 17:23:47 UTC.

### 5.2.1 Evidence of conical graupel

According to Fig. 19(a), from around 17:22 UTC, a region with large negative differential reflectivity appears at around 4000 m to 6000 m. From Fig. 19(c) and (d), this region has enhanced slanted linear depolarisation ratio and reduced copolar correlation coefficient. Inspecting the spectograms during this period, it is found that from 17:21:24 UTC, a separate particle mode with

negative $sZ_{DR}$ is present on the left side of the Doppler spectrum at around 6000 m as shown in Fig. 23(a). The reflectivity of this mode grew with time and it descended to around 4300 m near 17:24 UTC. The spectral reflectivity and $sZ_{DR}$ of one instant where this mode is present is shown in Fig. 21(a) and (b). When negative $sZ_{DR}$ appears on the left part of the spectrum, $sZ_{DR}$ of the right part of the spectrum is close to zero. Since small particles are more easily aligned by an electric field and they are not aligned in this case, the negative $sZ_{DR}$ in the left part of the spectrum might be due to the presence of conical

graupel (Lu et al., 2016a).





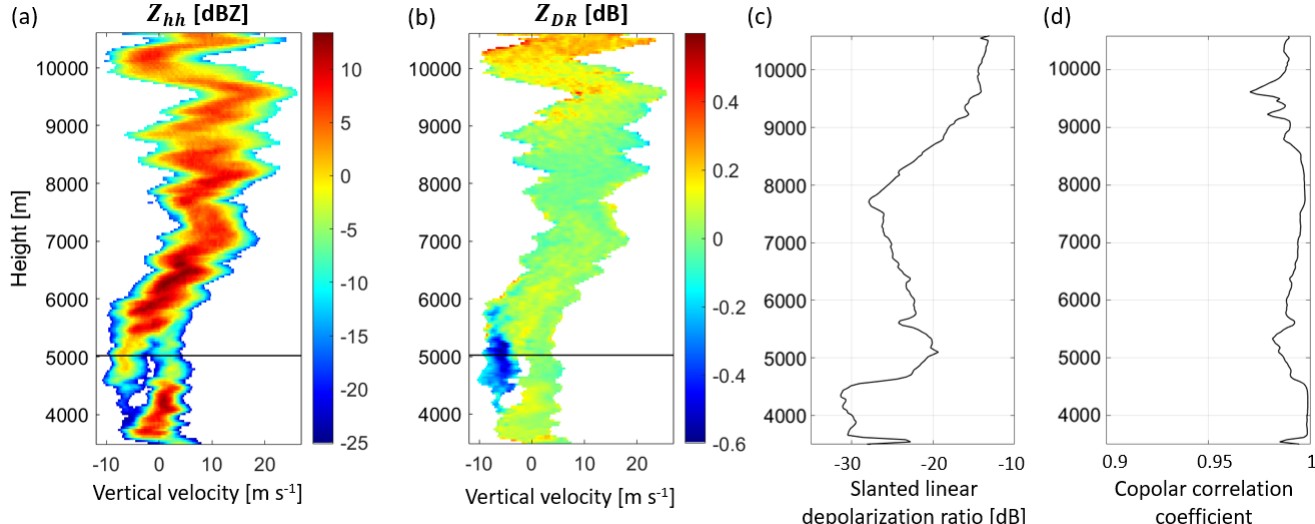

**Figure 21.** Spectograms of (a) equivalent reflectivity, (b) differential reflectivity, and profiles of (c) slanted linear depolarisation ratio, (d) copolar correlation coefficient at 17:22:25 UTC. Spectra at 5021 m indicated by black horizontal line in (a) and (b) are shown in Fig. 22(d-f).

The Doppler spectra of reflectivity, $Z_{DR}$ and $\Psi_{DP}$ at 5021 m at the instant shown in Fig. 21 are shown in Fig. 22(d-f). The spectral differential phase shift deviates from the Rayleigh plateau at about $-2$ m s$^{-1}$, signifying Mie scattering. To ensure correct interpretation of $sZ_{DR}$, scattering simulation is carried out using typical parameters of conical graupel. From the literature, the theoretical axis ratio of conical graupel is 1.05 and the density is 0.55 g cm$^{-3}$ (Spek et al., 2008), which

is equivalent to an ice fraction of 0.6. The diameter of conical graupel is typically 2 to 8 mm (Pruppacher and Klett, 1980). The canting angle follows a Gaussian distribution with a zero mean and a standard deviation of 30° (Bringi and Chandrasekar, 2001). Conical shape is not supported by the simulation code used, thus the shape is assumed to be spheroidal. The simulated $Z_{DR}$ and $\delta_{co}$ are shown in Fig. 22(b-c). In Rayleigh scattering regime, the differential reflectivity of conical graupel is negative. Both $Z_{DR}$ and $\delta_{co}$ decrease when the Mie scattering regime is reached. $\delta_{co}$ reaches a minimum earlier than $Z_{DR}$. As particle

size increases further, $\delta_{co}$ increases sharply and becomes positive, during which $Z_{DR}$ reaches its minimum. Afterwards, $\delta_{co}$ reaches a local maximum and decreases slightly before increasing again. $Z_{DR}$ increases and continues to oscillate. Similar behaviours are observed in the Doppler spectra at 5021 m at 17:22:25 UTC (Fig. 22(e-f)). $s\Psi_{DP}$ reaches a minimum at $-3.9$ m s$^{-1}$ and increases sharply as particle size further increases. $sZ_{DR}$ reaches a minimum at $-5.0$ m s$^{-1}$ while $s\Psi_{DP}$ is still increasing. Afterwards, $s\Psi_{DP}$ reaches a maximum and decreases slightly, while $sZ_{DR}$ continues to increase. The minimum of

reflectivity in Fig. 22(d) is not located at the Mie minimum according to the simulation (Fig. 22(a)). This suggests that the two peaks in spectral reflectivity represent two particle populations, the left peak corresponding to conical graupel and the right peak relating to spherical smaller ice particles. Comparing the simulation results and the observed spectra, the conical graupel present is about 3 to 5.2 mm in diameter.





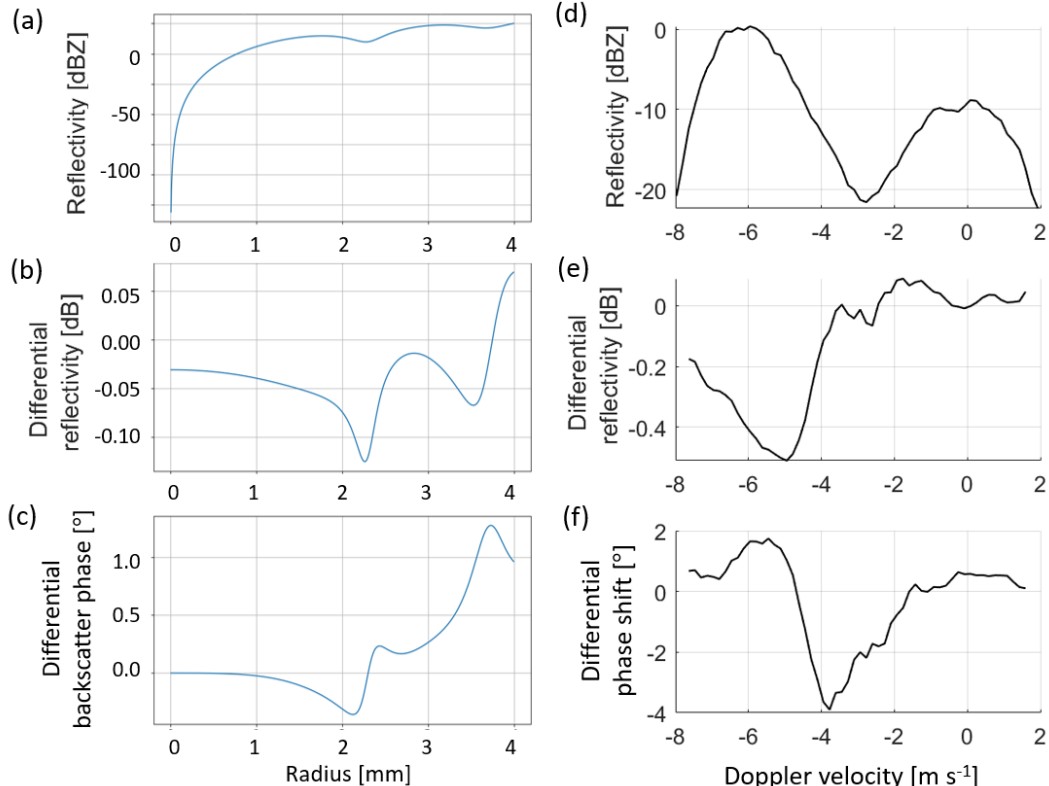

**Figure 22.** (a-c) Simulated reflectivity, differential reflectivity and differential backscatter phase of conical graupel. (d-f) Spectral reflectivity, differential reflectivity and differential phase shift at 5021 m at 17:22:25 UTC. Note that the Doppler velocity decreases towards more negative values when the radius increases.

From 4400 m to 5600 m where the negative $sZ_{DR}$ signature of graupel is the most prominent, $SL_{DR}$ increases and $\rho_{hv}$ decreases as shown in Fig. 21(c) and (d). This is likely because the radar resolution volume contains a variety of hydrometeors, including conical graupel and other small ice particles. Unfortunately, it is challenging to look for supercooled liquid water in this case since there is liquid water at the bottom of the cloud below the 0 °C level at around 4000 m, which means it is impossible to identify supercooled liquid water using liquid water path. It is worth noting from Fig. 21(a) and (b) that the population of graupel ends at around 4000 m. Since the radar is looking at an elevation angle of 45°, this suggests that graupel is not present closer than 4000 m from the radar, which means the region with graupel is localised in the thunderstorm cloud.

In Fig. 21(a) and (b), the spectograms are plotted with vertical velocity instead of Doppler velocity as in other spectograms in this article. The vertical velocity is estimated by assuming uniform horizontal wind predicted by the ECMWF model. By plotting with vertical velocity, it is clear that the graupel are falling, while smaller ice particles on the right with positive vertical velocities are brought upwards by updrafts. As the falling graupel collide with the rising ice particles, charges can be produced. According to Takahashi (1978), if temperature is below $-10$ °C, graupel will become negatively charged and vice versa. From the temperature profile measured by the microwave radiometer coupled to the cloud radar, the temperature is $-10$ °C at around





5550 m. This means that above 5550 m, falling graupel that collides with rising ice particles becomes negatively charged, forming a negative charge region in the cloud. Meanwhile, small ice particles that gained positive charges due to collisions are brought upwards by updrafts, so the upper part of the cloud is positively charged. Below 5550 m where temperature is above

$-10\,°C$, falling graupel acquires positive charge, causing the cloud base to become positively charged. This could result in the typical tripolar structure of thunderstorm clouds. Nonetheless, the temperature profile inside the thunderstorm cloud may be different from the temperature profile measured by the microwave radiometer looking towards the zenith, so the actual charge distribution in the cloud may be different.

### 5.2.2  Alignment of particles

At 17:21:32 UTC, a lightning of 5 kA occurred around 8.5 km away on the line of sight of the radar (stroke number 7 in Fig. B4). This is a cloud-to-cloud lightning with medium strength. One second before that, negative $sZ_{DR}$ values are observed for large and small particles from 8000 m to 9000 m as shown in Fig. 23(a). The minimum value is around $-0.40$ dB on the left side of the spectrum and $-0.36$ dB on the right side of the spectrum. Negative $sZ_{DR}$ values disappeared at 17:21:38 UTC. Note that the timestamps of the cloud radar correspond to the end of the measurement after all chirp sequences have

been transmitted, therefore the spectrum at 17:21:34 UTC may contain backscattered signals before the lightning, which could explain why negative $sZ_{DR}$ is still observed. Since the location and time of negative $sZ_{DR}$ agree well with that of the lightning stroke and there are no other strokes close to this one in time and space, what is observed here is likely the vertical alignment and relaxation of particles right before and after a lightning stroke.

The $SL_{DR}$ across this lightning stroke also shows interesting signature. As shown in Fig. 23(b), at 17:21:31 UTC, $SL_{DR}$

from 8000 m to 9000 m suddenly decreases significantly to $-100$ dB and only recovered at 17:21:38 UTC. During this period, $\rho_{hv}$ does not change significantly and is high (Fig. 23(c)), which means that the decrease in $SL_{DR}$ is not due to low SNR. One possible cause is that almost all crystals are vertically aligned right before the lightning close to the location of lightning, which leads to low canting variance. As a result, there is a sudden decrease in $SL_{DR}$.

At 17:20:27 UTC, a strong cloud-to-cloud lightning of $-18$ kA occurred at a perpendicular distance of around 3 km away

from the radar (stroke number 92 in Fig. B5), but it is about 5.5 km away from the line of sight of the radar. Despite being quite distant from the line of sight of the radar, negative $sZ_{DR}$ values are observed for small particles from 5200 m to 5700 m before the lightning as shown in Fig. 24, which is probably due to the large magnitude of the electric field that generated the strong lightning. The minimum $sZ_{DR}$ observed is around $-0.36$ dB, which is similar to that observed in the previous case. Also similar to the previous case is that $sZ_{DR}$ values returned to the level before the lightning about 4 seconds after the lightning

from 17:20:31 UTC onward. However, unlike the previous case, negative $sZ_{DR}$ is only found for small particles, which may be because electric field strength reduces with distance from the lightning, thus it is not strong enough to align larger and heavier particles vertically. It is difficult to pinpoint when negative $sZ_{DR}$ first emerged due to this particular lightning stroke. Slightly negative $sZ_{DR}$ of about $-0.16$ dB can be found for light particles as early as 17:19:39 UTC, which could be due to other lightning in the same cloud. Although the same part of the spectrum shows positive $sZ_{DR}$ instead one timestamp before

at 17:19:09 UTC, a closer inspection reveals that it may be due to a different population of particles.





**Figure 23.** (a) Spectral differential reflectivity (b) slanted linear depolarisation ratio and (c) copolar correlation coefficient before and after lightning stroke (5 kA) at 17:21:32 UTC (stroke number 7 in Fig. B4).



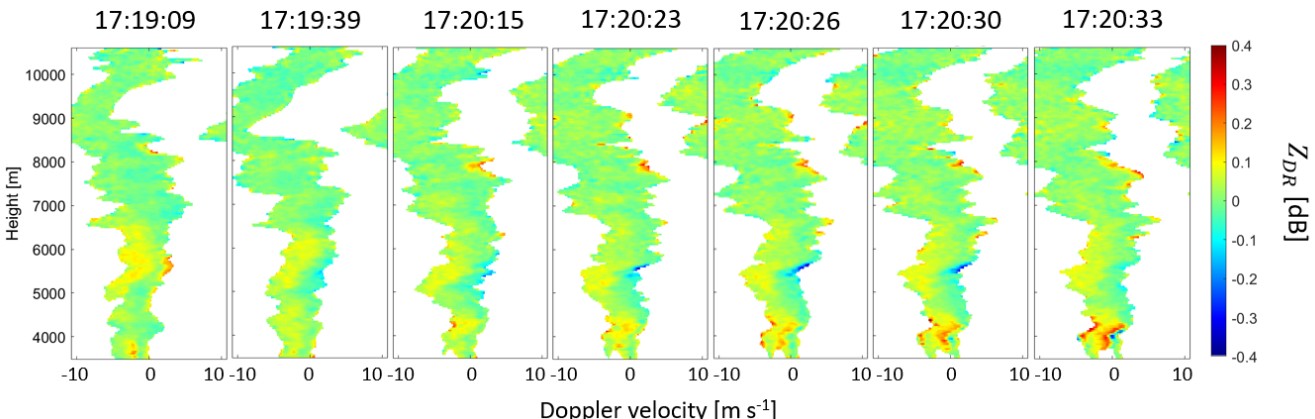

**Figure 24.** Spectral differential reflectivity before and after a strong lightning stroke ($-18$ kA) at 17:20:27 UTC (stroke number 92 in Fig. B5).

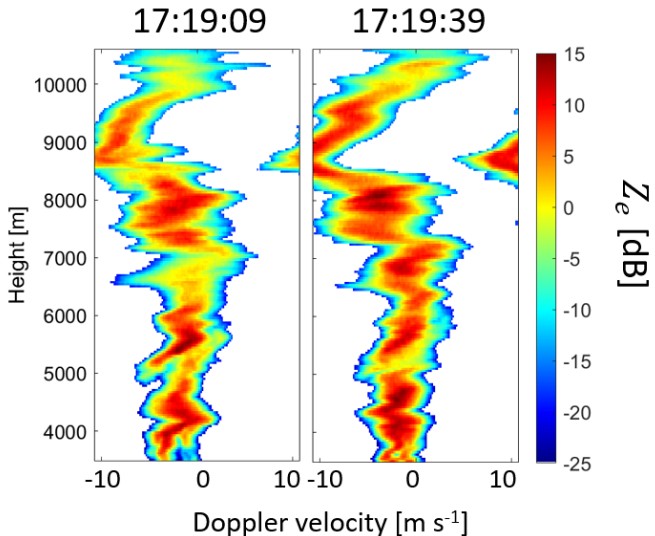

**Figure 25.** Spectral reflectivity at 17:19:09 and 17:39:39 UTC.

Comparing the spectral reflectivity at 17:19:09 and 17:19:39 UTC in Fig. 25 with the spectral differential reflectivity in Fig. 24, negative $sZ_{DR}$ from 5000 m to 6000 m at 17:19:39 UTC likely belongs to particles associated with the upper part of the spectrum, while positive $sZ_{DR}$ at the same height at 17:19:09 UTC belongs to particles associated with the lower part of the spectrum. Since the two populations of particles may overlap, small negative $sZ_{DR}$ values may be masked by positive values.

Therefore, it is not possible to conclude that no negative $sZ_{DR}$ values or no vertical alignment of particles are found before 17:19:39 UTC.




Also, unlike the previous case, right before the lightning at 17:20:27 UTC, $SL_{DR}$ does not show a sudden decrease. This could be because the lightning occurred some distance away from the line of sight of the radar. Therefore, not all particles are aligned, thus $SL_{DR}$ did not decrease significantly.

From 17:23:40 UTC, $sZ_{DR}$ becomes negative for the entire Doppler spectrum above 7000 m, such as the spectrum at 17:23:47 UTC shown in Fig. 27(b). This could be due to vertical alignment of all particles by strong cloud electric field. However, from Fig. 2, most of the thunderstorm cloud above 4000 m from 17:24 to 17:29 UTC was not visible to the radar due to large attenuation. There is also significant amount of liquid water below the cloud, leading to differential attenuation of horizontal and vertical polarizations, which may cause $Z_{DR}$ values to be negatively biased. An evidence of differential

attenuation is that $Z_{DR}$ values become more negative as the thickness of the layer that contains liquid water increases. Also, many lightnings occurred close to each other in time during this period, so it is impossible to isolate each lightning stroke and analyse the changes before and after each stroke. These limit the investigation on the period with the most intense lightning activities.

### 5.2.3   Strong updraft and turbulence

As shown in Fig. 20(a), from 17:18 to 17:24 UTC, vertical air velocity is large and positive (15-30 m s$^{-1}$) above 7000 m, indicating strong updraft in the cloud. From Fig. 20(b), the top of the cloud above 6000 m has large Doppler spectrum width of 3 to 4 m s$^{-1}$. In stratiform rain, the cloud top usually has low spectrum width since small and light particles have a small range of fall velocities. The large spectrum width observed here might be due to strong turbulence in the thunderstorm cloud. Slanted linear depolarisation ratio is high and copolar correlation coefficient is low in this region, which could be the result of

large canting variance of particles due to strong turbulence.

From 17:22:30 to 17:24:00 UTC from 5000 m to 7000 m, there are three co-locating peaks of $SL_{DR}$ and troughs of $\rho_{hv}$. The lowest peak at around 5000 m is located just above the graupel layer, such as in the example shown in Fig. 26 where the peak is found at 5060 m. From Fig. 26(h), the vertical air velocity does not vary much near this height, so the sudden increase in $SL_{DR}$ and decrease in $\rho_{hv}$ may not be due to increased canting variance due to turbulence. Meanwhile, the spectral $Z_{DR}$

where the peak of $SL_{DR}$ and $\rho_{hv}$ is located shows multiple peaks (Fig. 26(d)). This could be due to a variety of hydrometeors with different axis ratios that are the seeds for forming conical graupel. Therefore, the high $SL_{DR}$ and low $\rho_{hv}$ in this case are likely due to co-existence of different types of particles.

The middle and highest peaks of $SL_{DR}$ and troughs of $\rho_{hv}$ are found at around 5900 m and 6400 m, such as in the example shown in Fig. 27. From Fig. 27(h), vertical air velocity changes sharply at these heights, which can produce strong turbulence.

Therefore, the sudden increase in $SL_{DR}$ and decrease in $\rho_{hv}$ may be due to increased canting variance under turbulence. With strong turbulence, the Doppler spectra is no longer ordered with small particles on the right and large particles on the left because particles with different sizes are mixed.





**Figure 26.** 18 June 2021 17:22:57 UTC where the lowest peak of $SL_{DR}$ and trough of $\rho_{hv}$ is observed. (a-b) Spectograms of reflectivity and differential reflectivity (c-e) spectra of reflectivity, $Z_{DR}$ and $\Psi_{DP}$ at 5060 m (f-i) profiles of $SL_{DR}$, $\rho_{hv}$, vertical air velocity and Doppler spectrum width.





**Figure 27.** 18 June 2021 17:23:47 UTC where the three peaks of $SL_{DR}$ and troughs of $\rho_{hv}$ are observed. (a-b) Spectograms of reflectivity and differential reflectivity (c-e) spectra of reflectivity, $Z_{DR}$ and $\Psi_{DP}$ at 5877 m (middle peak) and 6422 m (highest peak) (f-i) profiles of $SL_{DR}$, $\rho_{hv}$, vertical air velocity and Doppler spectrum width.

### 5.2.4 Possibility of chains

From Fig. 19(a), high $Z_{DR}$ is observed at the top of the cloud from 17:22 UTC onward. The Doppler spectra at 10003 m at 18
June 2021 17:22:57 UTC is shown in Fig. 28(d-f). The differential reflectivity of the Rayleigh plateau (Doppler velocity > 5 m s$^{-1}$) is around 0.2 dB, and the entire spectrum of $Z_{DR}$ is positive. One hypothesis is that the large particles with positive $sZ_{DR}$ are chain-like aggregates that formed under cloud electric field. According to Connolly et al. (2005), chains can be observed





when the temperature is lower than $-40$ °C. In the case being studied, temperature is lower than $-40$ °C above 9600 m, which backs up the hypothesis that chains might be present. Mie scattering occurs at around 5 m s$^{-1}$ as $sZ_{DR}$ increases and $s\Psi_{DP}$ 580 begins to fluctuate. Scattering simulation is carried out to estimate the size of the particles based on the trend of Mie scattering. In the simulation, chains are modelled as prolate particles. From the literature, chain-like aggregates can be up to several tens of particles long (Gayet et al., 2012) and individual crystals in the chains are aligned with their maximum dimension along the length of the chain (Connolly et al., 2005). The individual particles can be plates, with typical axis ratio of less than 0.3 (Spek et al., 2008), or quasi-spherical frozen droplets with diameters of 15-20 $\mu$m (Gayet et al., 2012). Based on these, the axis ratio 585 of chains (ratio of long axis to short axis) should be much larger than 1. However, the simulation code used cannot converge when particle size increases when the axis ratio is too large, thus the maximum possible axis ratio that can be used to allow Mie scattering regime to be reached is 7. Since chains are a type of aggregates, ice fraction of typical aggregates is used, which is 0.3. The chains are assumed to be oriented with their long axis parallel to the horizontal, thus the Euler angle $\beta = 90°$. The orientation follows a Gaussian distribution with a standard deviation of 0.1°.

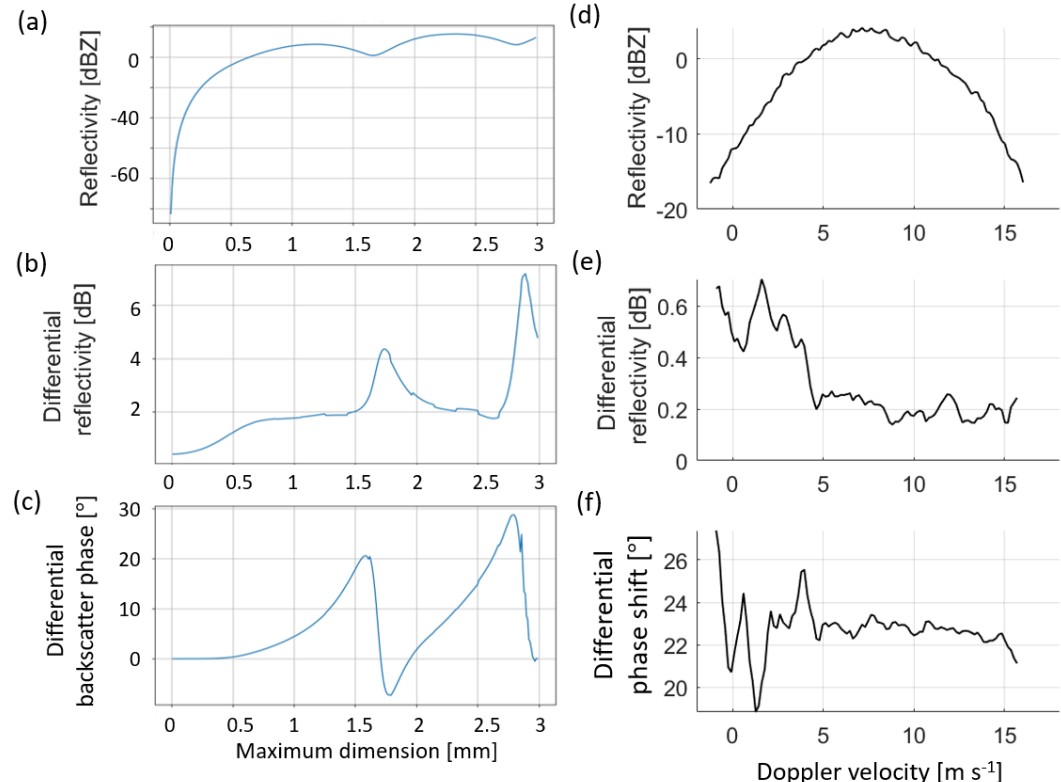

**Figure 28.** (a-c) Simulated reflectivity, differential reflectivity and differential backscatter phase of chains with axis ratio 7 and ice fraction 0.3 oriented with long axis parallel to the horizontal. (d-f) Spectral reflectivity, differential reflectivity and differential phase shift at 10003 m at 17:22:57 UTC. Note that the Doppler velocity decreases when the maximum dimension increases.





The simulated scattering behaviour of chains are shown in Fig. 28(a-c). The maximum dimension refers to the length of the long axis of the spheroid. Similar to the observed spectral differential phase shift in Fig. 28(f), the simulated differential backscatter phase Fig. 28(c) shows a peak followed by a trough, while spectral differential reflectivity shows a peak. This occurs when scatterers have a maximum dimension of around 1.5 mm, which suggests that the observed particles have sizes reaching around 1.5 mm. However, from the literature, chains collected in thunderstorm clouds have lengths of at most 0.8 mm

(Connolly et al., 2005). There are several reasons for the discrepancy between results based on simulation and the literature. For instance, the axis ratio used in the simulation may not be large enough for chains, and a spheroid model may not be sufficient to describe the shape of chains. According to Fig. 1, chains can be irregular in shape, and the effect of the irregularities could increase as their sizes become more comparable to the radar wavelength.

Other than using scattering simulations, the presence of chain-like aggregates could also be supported by checking whether

the region concerned fulfils the conditions described in the literature for the enhancement of aggregation by electric fields to occur, which are an electric field of 50-150 kV m$^{-1}$ and ice particle number concentration of at least 2 cm$^{-3}$. Unfortunately, these data are not available in this study.

## 6    Conclusions

In this study, two major thunderstorm clouds on 18 June 2021 from 16:10 to 17:45 UTC near Cabauw were studied in depth

to explore how bulk and spectral cloud radar data at 35 GHz can be used to help understand processes in thunderstorm clouds. Prior to the analysis of the spectral differential reflectivity, $sZ_{DR}$, the spectral differential phase, which indicates in which range of Doppler velocities Rayleigh and Mie scattering occur, was investigated. This prevents the misinterpretation of $sZ_{DR}$ extrema as characteristics of different ice particle populations in the Mie scattering regime.

Several types of hydrometeors are observed in clouds that produced lightning. In the first cloud, supercooled liquid water

is found at the edge of the cloud at around 6000 m, which is supported by the increased liquid water path and near zero differential reflectivity of a separate mode of particles on the right of the Doppler spectra. In the fourth cloud, comparison between scattering simulations and observations supports the presence of conical graupel with maximum dimension of 3-5.2 mm. The falling graupel coexist with ice particles that are brought upwards by updrafts, which could lead to non-inductive charging. The temperature at the corresponding heights could give rise to a tripolar structure of the thunderstorm cloud. In both

clouds discussed above, there is a possibility that chain-like aggregates of small ice particles are present in the top of the cloud, which is reflected by large magnitudes of $sZ_{DR}$ on the left side of the Doppler spectrum. Nonetheless, no realistic scattering simulation could be carried out to confirm the size and characteristics of the chains.

Vertical alignment of ice particles can be observed right before lightning up to 4 seconds before lightning and disappears within 3 seconds after the lightning reflected by negative $sZ_{DR}$ values as low as $-0.4$ dB at $45°$ elevation. When the lightning

is close to the line of sight of the radar, particles of all sizes are vertically aligned with $sZ_{DR}$ values all negative. At this juncture, the bulk variable $SL_{DR}$ tends to 0 because of the vertical alignment of all the particles in the radar resolution volume. It is worth noting that negative $sZ_{DR}$ can be observed if the lightning is close to the line of sight of the radar or is far (up to





5.5 km) but strong, but sudden decrease in $SL_{DR}$ can only be observed if the lightning is close but not when it is far. When the lightning is far away, only small particles on the right side of the Doppler spectra are vertically aligned and exhibit negative

$sZ_{DR}$ values, while the bulk variable $Z_{DR}$ has positive values. However, there are also some situations where negative $sZ_{DR}$ is observed that suggests vertical alignment of particles by the electric field, yet there are no lightning strokes measured nearby in space and time. This could be because the electric field is not strong enough to trigger lightning, or that some lightning strokes were not recorded.

Updrafts and downdrafts can be observed at different parts of the thunderstorm cloud. Near the edge of the first cloud,

downdrafts can be observed. At the top and near the core of the fourth cloud, strong updrafts of up to 30 m s$^{-1}$ can be observed. In general, vertical air velocity is not uniform in thunderstorm clouds, which suggests that there is strong turbulence. This is also supported by large Doppler spectrum width of up to 3-4 m s$^{-1}$. When strong turbulence is present, slanted linear depolarisation ratio increases and copolar correlation coefficient decreases, which suggest that canting angle variance of particles within a radar resolution volume increases.

In the case being studied, only measurements with constant elevation and azimuth and zenith observation were available, but their drawback is that only a small part of the thunderstorm cloud along the radar's line of sight could be measured, which leads to a low number of thunderstorm events recorded by the radar. In addition, it is not possible to look at the whole thunderstorm cloud at the same time to analyse the spatial variations within the cloud. Also, each part of the thunderstorm cloud only passes over the line of sight of the radar once, thus it is impossible to analyse the evolution of different parts of the cloud. A more

appropriate radar measurement mode for studying thunderstorm clouds would be azimuth scan with constant elevation (PPI). With PPI mode, thunderstorm clouds in all directions can be measured by the radar, so there can be more cases to choose from for in-depth study or statistical analysis. Moreover, it may become possible to analyse differences between different parts of the thunderstorm cloud with different levels of lightning activities, as well as how the cloud evolves with time.





## Appendix A: Weather radar images related to the study case



**Figure A1.** Radar images and location of lightning strokes (yellow asterisks) from 18 June 2021 16:10 to 16:40 UTC (© OpenStreetMap contributors 2023. Distributed under the Open Data Commons Open Database License (ODbL) v1.0.)(GmbH). Red triangle shows radar location, red ruler shows line of sight of radar with each mark equal to 1 km.





**Figure A2.** Radar images and location of lightning strokes from 18 June 2021 16:45 to 17:10 UTC (© OpenStreetMap contributors 2023. Distributed under the Open Data Commons Open Database License (ODbL) v1.0.)(GmbH). Legend same as Fig. A1.





**Figure A3.** Radar images and location of lightning strokes from 18 June 2021 17:15 to 17:40 UTC (© OpenStreetMap contributors 2023. Distributed under the Open Data Commons Open Database License (ODbL) v1.0.)(GmbH). Legend same as Fig. A1.




## Appendix B: Lightning maps

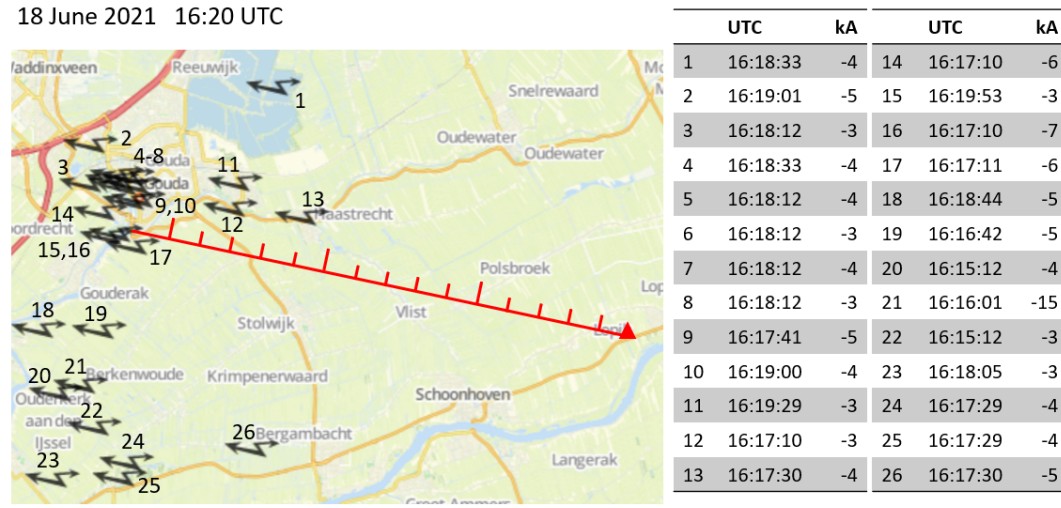

| | UTC | kA | | UTC | kA |
|---|---|---|---|---|---|
| 1 | 16:18:33 | -4 | 14 | 16:17:10 | -6 |
| 2 | 16:19:01 | -5 | 15 | 16:19:53 | -3 |
| 3 | 16:18:12 | -3 | 16 | 16:17:10 | -7 |
| 4 | 16:18:33 | -4 | 17 | 16:17:11 | -6 |
| 5 | 16:18:12 | -4 | 18 | 16:18:44 | -5 |
| 6 | 16:18:12 | -3 | 19 | 16:16:42 | -5 |
| 7 | 16:18:12 | -4 | 20 | 16:15:12 | -4 |
| 8 | 16:18:12 | -3 | 21 | 16:16:01 | -15 |
| 9 | 16:17:41 | -5 | 22 | 16:15:12 | -3 |
| 10 | 16:19:00 | -4 | 23 | 16:18:05 | -3 |
| 11 | 16:19:29 | -3 | 24 | 16:17:29 | -4 |
| 12 | 16:17:10 | -3 | 25 | 16:17:29 | -4 |
| 13 | 16:17:30 | -4 | 26 | 16:17:30 | -5 |

**1 to 3 kA Sissy mutterer**
**3 to 7 kA Medium roller**
**> 7 kA Strong slammer**

**Figure B1.** Location, time and power of lightning strokes from 18 June 2021 16:15 to 16:20 UTC (© OpenStreetMap contributors 2023. Distributed under the Open Data Commons Open Database License (ODbL) v1.0.)(GmbH). Red triangle shows radar location, red ruler shows line of sight of radar with each mark equal to 1 km.

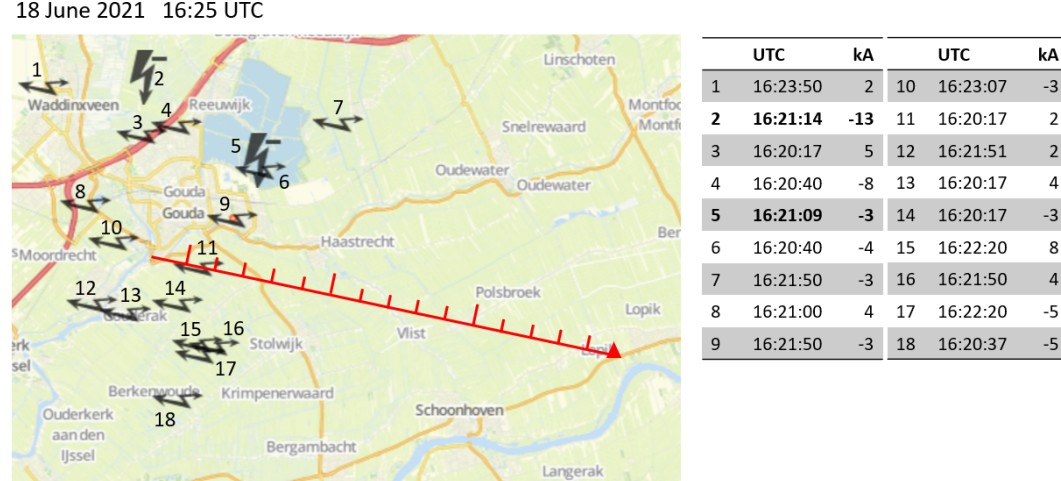

| | UTC | kA | | UTC | kA |
|---|---|---|---|---|---|
| 1 | 16:23:50 | 2 | 10 | 16:23:07 | -3 |
| **2** | **16:21:14** | **-13** | 11 | 16:20:17 | 2 |
| 3 | 16:20:17 | 5 | 12 | 16:21:51 | 2 |
| 4 | 16:20:40 | -8 | 13 | 16:20:17 | 4 |
| **5** | **16:21:09** | **-3** | 14 | 16:20:17 | -3 |
| 6 | 16:20:40 | -4 | 15 | 16:22:20 | 8 |
| 7 | 16:21:50 | -3 | 16 | 16:21:50 | 4 |
| 8 | 16:21:00 | 4 | 17 | 16:22:20 | -5 |
| 9 | 16:21:50 | -3 | 18 | 16:20:37 | -5 |

**Figure B2.** Location, time and power of lightning strokes from 18 June 2021 16:20 to 16:25 UTC (© OpenStreetMap contributors 2023. Distributed under the Open Data Commons Open Database License (ODbL) v1.0.)(GmbH). Legend same as Fig. B1. Cloud-to-ground lightning in bold.



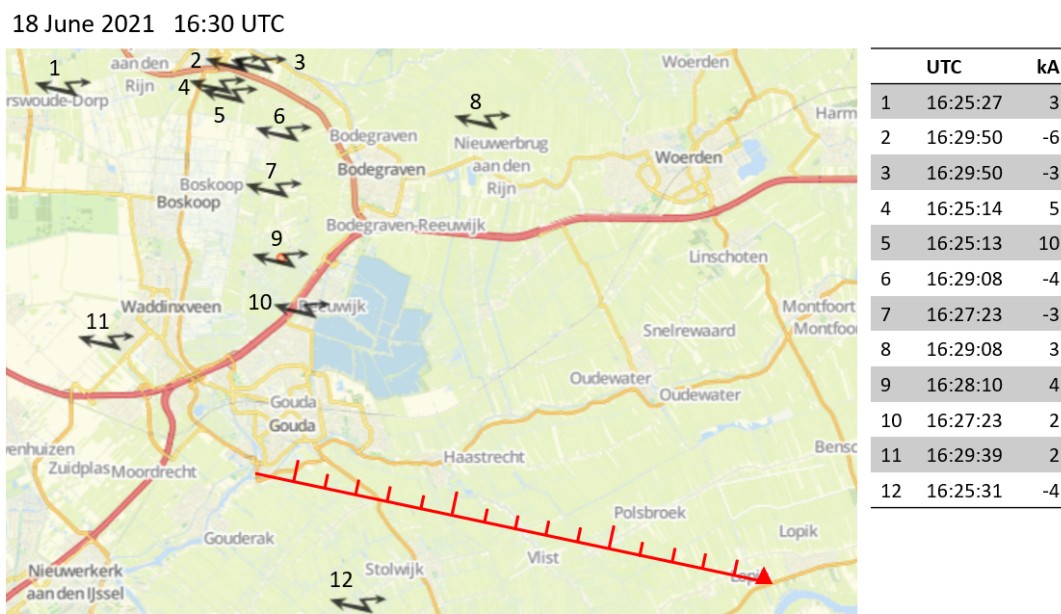

**Figure B3.** Location, time and power of lightning strokes from 18 June 2021 16:25 to 16:30 UTC (© OpenStreetMap contributors 2023. Distributed under the Open Data Commons Open Database License (ODbL) v1.0.)(GmbH). Legend same as Fig. B1.

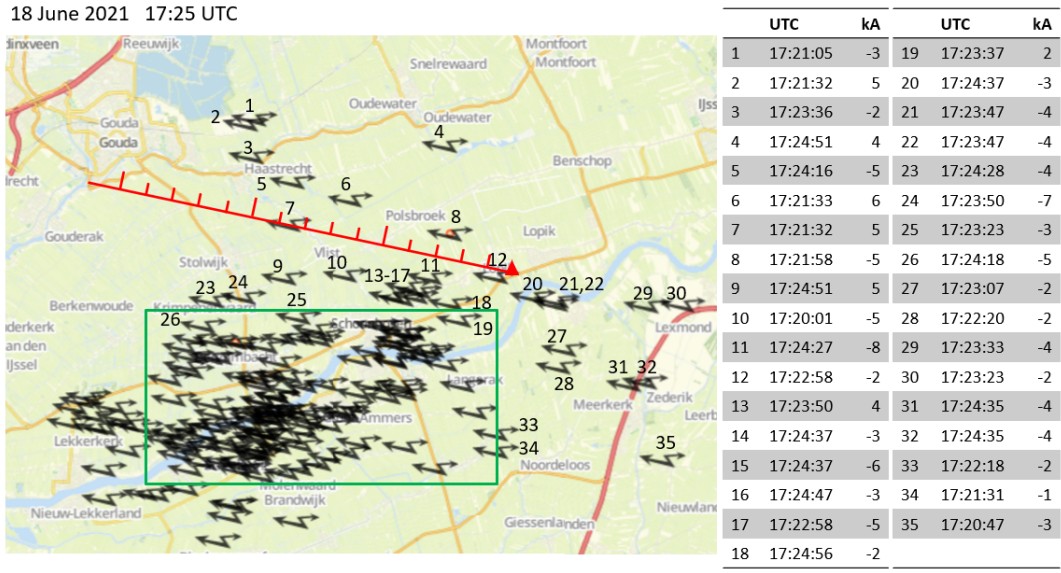

**Figure B4.** Location, time and power of lightning strokes from 18 June 2021 17:20 to 17:25 UTC (© OpenStreetMap contributors 2023. Distributed under the Open Data Commons Open Database License (ODbL) v1.0.)(GmbH). See Fig. B5 for lightning in green rectangle. Legend same as Fig. B1.





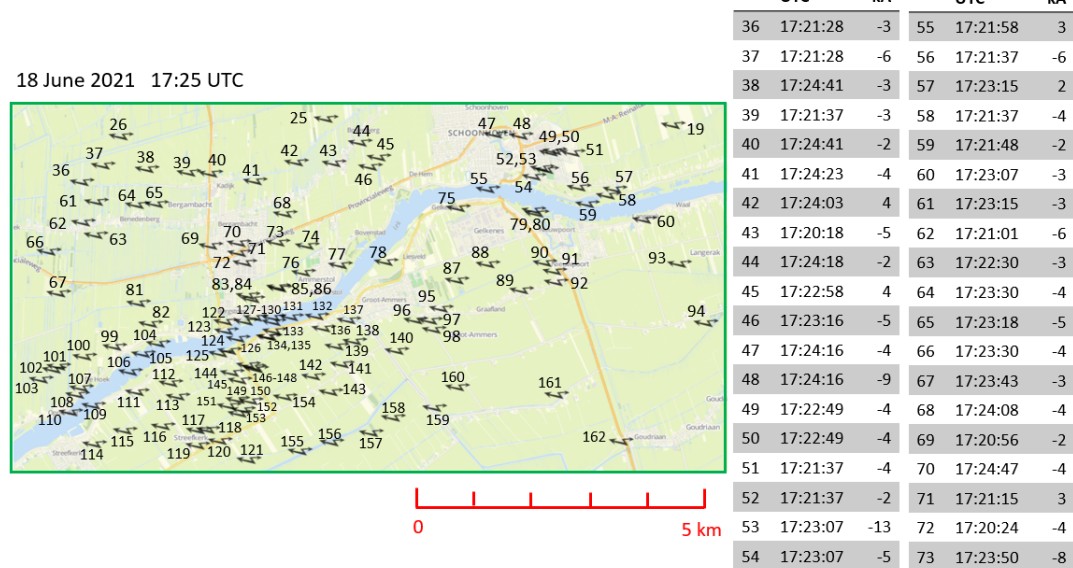

| | UTC | kA | | UTC | kA | | UTC | kA | | UTC | kA | | UTC | kA | | UTC | kA | | UTC | kA |
|---|---|---|---|---|---|---|---|---|---|---|---|---|---|---|---|---|---|---|---|---|
| 36 | 17:21:28 | -3 | 55 | 17:21:58 | 3 | 74 | 17:24:35 | -4 | 93 | 17:20:27 | -3 | 112 | 17:21:29 | -2 | 131 | 17:23:36 | -5 | 150 | 17:22:18 | -4 |
| 37 | 17:21:28 | -6 | 56 | 17:21:37 | -6 | 75 | 17:21:15 | -3 | 94 | 17:20:10 | -3 | 113 | 17:20:30 | -3 | 132 | 17:21:02 | -5 | 151 | 17:22:23 | -7 |
| 38 | 17:24:41 | -3 | 57 | 17:23:15 | 2 | 76 | 17:23:43 | -2 | 95 | 17:23:55 | -3 | 114 | 17:21:35 | -2 | 133 | 17:23:25 | -5 | 152 | 17:22:23 | -4 |
| 39 | 17:21:37 | -3 | 58 | 17:21:37 | -4 | 77 | 17:21:03 | -5 | 96 | 17:20:27 | -3 | 115 | 17:21:17 | -6 | 134 | 17:22:27 | -12 | 153 | 17:20:37 | -5 |
| 40 | 17:24:41 | -2 | 59 | 17:21:48 | -2 | 78 | 17:20:27 | -3 | 97 | 17:21:49 | -3 | 116 | 17:21:42 | -3 | 135 | 17:21:49 | -4 | 154 | 17:20:30 | -3 |
| 41 | 17:24:23 | -4 | 60 | 17:23:07 | -3 | 79 | 17:21:05 | -3 | 98 | 17:23:55 | -2 | 117 | 17:21:52 | -4 | 136 | 17:21:14 | -3 | 155 | 17:22:23 | -4 |
| 42 | 17:24:03 | 4 | 61 | 17:23:15 | -3 | 80 | 17:20:01 | -3 | 99 | 17:20:18 | -6 | 118 | 17:22:04 | -6 | 137 | 17:22:41 | -2 | 156 | 17:20:06 | -3 |
| 43 | 17:20:18 | -5 | 62 | 17:21:01 | -6 | 81 | 17:22:30 | -7 | 100 | 17:24:35 | -3 | 119 | 17:21:45 | -6 | 138 | 17:22:01 | -4 | 157 | 17:21:05 | -1 |
| 44 | 17:24:18 | -2 | 63 | 17:22:30 | -3 | 82 | 17:21:15 | -2 | 101 | 17:21:01 | -2 | 120 | 17:22:04 | -4 | 139 | 17:21:21 | -2 | 158 | 17:20:42 | -4 |
| 45 | 17:22:58 | 4 | 64 | 17:23:30 | -4 | 83 | 17:22:20 | -4 | 102 | 17:21:08 | -3 | 121 | 17:22:04 | -3 | 140 | 17:20:49 | -4 | 159 | 17:20:47 | -2 |
| 46 | 17:23:16 | -5 | 65 | 17:23:18 | -5 | 84 | 17:22:20 | 7 | 103 | 17:22:07 | -2 | 122 | 17:21:02 | -3 | 141 | 17:22:33 | -4 | 160 | 17:22:20 | -5 |
| 47 | 17:24:16 | -4 | 66 | 17:23:30 | -4 | 85 | 17:23:25 | -6 | 104 | 17:21:33 | -3 | 123 | 17:23:01 | -4 | 142 | 17:21:11 | 6 | 161 | 17:21:27 | -2 |
| 48 | 17:24:16 | -9 | 67 | 17:23:43 | -3 | 86 | 17:23:55 | -4 | 105 | 17:20:37 | -3 | 124 | 17:22:55 | -2 | 143 | 17:20:04 | -3 | 162 | 17:21:31 | -2 |
| 49 | 17:22:49 | -4 | 68 | 17:24:08 | -4 | 87 | 17:22:20 | -3 | 106 | 17:23:18 | -2 | 125 | 17:21:01 | -3 | 144 | 17:21:15 | -3 | | | |
| 50 | 17:22:49 | -4 | 69 | 17:20:56 | -2 | 88 | 17:22:22 | -3 | 107 | 17:20:25 | -2 | 126 | 17:21:26 | 7 | 145 | 17:21:29 | -4 | | | |
| 51 | 17:21:37 | -4 | 70 | 17:24:47 | -4 | 89 | 17:21:57 | -3 | 108 | 17:20:30 | -7 | 127 | 17:23:18 | -7 | 146 | 17:20:11 | -2 | | | |
| 52 | 17:21:37 | -2 | 71 | 17:21:15 | 3 | 90 | 17:21:05 | -7 | 109 | 17:21:08 | -2 | 128 | 17:23:50 | 3 | 147 | 17:21:49 | -8 | | | |
| 53 | 17:23:07 | -13 | 72 | 17:20:24 | -4 | 91 | 17:22:55 | -4 | 110 | 17:20:30 | -8 | 129 | 17:23:18 | -6 | 148 | 17:21:49 | -5 | | | |
| 54 | 17:23:07 | -5 | 73 | 17:23:50 | -8 | 92 | 17:20:27 | -18 | 111 | 17:22:49 | -3 | 130 | 17:23:25 | -8 | 149 | 17:24:35 | -4 | | | |

**Figure B5.** Location, time and power of lightning strokes in green rectangle in Fig. B4 (© OpenStreetMap contributors 2023. Distributed under the Open Data Commons Open Database License (ODbL) v1.0.)(GmbH). Legend same as Fig. B1.



*Data availability.* These cloud radar data are not published yet but are available in hourly NetCDF files from co-author Christine Unal (c.m.h.unal@tudelft.nl) upon request.

*Author contributions.* HYLM performed the simulations, created the plots, analysed the data, and wrote the manuscript. CU focused on the supervision of HYLM and provided feedback on the chosen methodology, the analysis and the manuscript.

*Competing interests.* The authors declare that they have no conflict of interest.

*Acknowledgements.* The authors would like to thank (Rob Mackenzie, Mahaut Sourzac) and (Saverio Guzzo, André Castro) for their support for maintaining the cloud radar operational and for data management, respectively. This work was supported by the Ruisdael Observatory, a scientific research infrastructure co-financed by the Netherlands Organisation for Scientific Research (NWO), under grant number 184.034.015.



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
