# Peer review of "Peering into the heart of thunderstorm clouds: Insights from cloud radar and spectral polarimetry"

_EGUsphere, 2024_

## Referee Comment (RC1)

Review for "Peering into the heart of theunderstorm clouds: Insights from cloud radar and spectral polarimetry" (10.5194/egusphere-2024-1232)

**General comments:**

This paper uses the technique of spectral data from a polarimetric cloud radar to study the passage of convective cells with lightning activity. The study is novel in that spectral polarimetry analysis in the Ka-band is used to study a thunderstorm cloud, so as to identify the different scattering regimes and the suitability of interpreting sZDR extrema. The inclusion of scattering simulations using the T-matrix method supplements the explanation of peaks and troughs in the spectral polarimetric data. In particular, the study makes convincing interpretations for the presence of conical graupel and vertical alignment of particles prior to a lightning strike, which are consistent with past literature. The phenomena of supercooled liquid water, chains and strong updrafts associated with in-cloud turbulence are also explored.

The manuscript is mostly well written, with a detailed set of results and scientific explanations that make it suitable for the scope of AMT. The novelty of using spectral polarimetry of a cloud radar on a thunderstorm case is well appreciated, especially when supplemented with sufficient results that ultimately lead to valid interpretations of the hydrometeor type and the sound conclusions reached regarding their presence in convective cells with lightning. There are several clarifications that are needed in the first two sections of the manuscript, alongside suggested reduction to the results section. Although the specific comments are mostly minor, they combine to push me to recommend a major revision.

**Specific comments:**

**Major**

1. Although the authors have made it clear that there could be differential attenuation due to attenuated reflectivity from the presence of rain drops for 17:24 to 17:30UTC, there still seems to be areas of enhanced calibrated ZDR just above 2000m of height in Fig. 19a. Could the authors explain whether they have taken into account differential attenuation when calibrating ZDR? If so, how? If not, what is the estimated magnitude of differential attenuation? How would this change the calibrated ZDR values and alter the interpretation of the results in Sections 5.2.1 and 5.2.4 regarding the evidence of conical graupel and possibility of chains.

2. There is a good variety of past literature included within the introduction section. The authors should consider an additional discussion section to the manuscript to discuss how results in this manuscript both qualitatively and

quantitatively compare with past studies if possible. The authors could consider the following questions including, but not limited to: How do the ZDR values of vertically aligned particles in this study comparable with those in Lund et al. (2009)? How does vertical alignment of ice up to 4 seconds before lightning comparable with past studies?

3. I appreciate the depth of the scientific deductions and explanations provided for the physical interpretation of the alignment of particles in Section 5.1.1. However, this has made the section rather long. To focus the discussion on the most probable reason for vertical alignment, the authors could consider shortening the paragraphs between lines 393 and 437 and potentially including some of the Doppler flipping figure and explanation as supplementary material.

**Minor**

Line 2: I agree that using a cloud radar to study thunderstorms mitigates the need to fly into one to take in-situ measurements. However, there is no mention of arrival time difference networks for lightning detection in the abstract. Yet, the authors do take advantage of the BLIDS system. Please mention this in the abstract.

Line 15: "reflected by the..." is confusing, especially since you mention spectrum flipping later on. I would just us "shown by the..."

Line 28: Include a diagram to supplement the explanations of the "numerous charging mechanisms".

Lines 32-36: Some explanation of the "inconsistencies" and "quantitatively unrealistic ineffective" mechanisms of charging would be appreciated here.

Line 44: Which part of the updraft column? Consider giving an indication of a height/temperature range, which would quantify what is meant by "where the temperature is low".

Line 46: What are the terminal velocities of the ice crystals that are "thrown upwards" in updrafts?

Lines 48-49: The authors mention that "all charging mechanisms above could contribute to certain extent at some time to cloud charging", but they also comment that some of these mechanisms are unrealistic earlier in the same paragraph. Please rewrite this paragraph so it is not self-contradicting.

Line 79: Provide temperature information if possible for the height range of 4-7 km in the Sokol et al. (2020) study, so as to be consistent with the descriptions of Mattos et al. (2016) earlier in the paragraph.

Line 81 and 558: A high Doppler spectrum width is not a sufficient condition to highlight the coexistence of different hydrometeor types. Such a measurement could

hint at strong shear and/or turbulence. See Feist et al. (2019) for details before clarifying the interpretation of results from Sokol et al. (2010).

Lines 107-109: Great to see the authors have considered past work in light of the novelty of using the spectral polarimetry technique in the Ka-band to study storms. However, I would like to see at least a brief literature review that mentions how polarimetric Doppler spectra have been used in other past studies. Were they for non-thunderstorm cases? Used other radar wavelengths? Did they also use the spectra to identify the transition between Rayleigh and Mie scattering regimes? Answering these questions will strengthen this paper on how novel the work is!

Line 111: Add a paragraph to introduce the remaining sections of the manuscript.

Line 116: Refer to Figure A1 immediately after constant azimuth of 282deg is mentioned.

Line 128: Consider adding a short explanation regarding how SLDR is an appropriate proxy for LDR.

Line 133: What is the initialisation time of the IFS forecast(s) used in this case study? Are outputs available hourly? Since the case spans longer than an hour, were outputs valid at different times used? Did the authors choose the closest valid time of the output to that of the radar observation and does the IFS forecast have to precede the radar observation time? Please include these details within the description of the IFS.

Line 136: Include a citation to the vertical coordinate system used by the IFS.

Lines 137-138: I don't think it is necessary to mention the act of "clicking on the lightning stroke".

Lines 148-149: It could be worth mentioning that although the bright band cannot be clearly seen in the Ze field, there are some signs of it in the dual-pol fields in Fig 3.

Lines 163-164: If vertical air velocity is obtained from the Doppler velocity bin with the smallest value, why do the colours in Fig. 5b hint at much larger vertical air velocity values than the mean vertical velocities in Fig 5a?

Line 169: Doppler spectra is not broadened purely by turbulence, but also by shear. See Chapter 5 of Doviak and Zrnić (1984) and Feist et al. (2019) for details.

Line 174: The use of "perpendicular distance" is potentially confusing here. Why not introduce this term earlier in line 173 when describing distance of the first cloud from the radar?

Line 187: Justify why integrated radar variables are computed from Level 0 data, rather than directly using Level 1 data.

Line 202: What were the original errors without the extra polarimetric calibration?

Lines 214-215: What is the definition of "North"? Grid North or True North? It is best to clarify this. From experience, there could be differences by up to a few degrees between grid north and true north, leading to significant differences in collocating radar echoes if the definitions are used interchangeably.

Line 227: Without having seen Fig 6 yet, it could be confusing as to what is meant by "rightmost valid bin". Worth clarifying that this means bin with the highest positive Doppler velocity value.

Lines 251-252: Convert the relation of psi_DP with phi_DP and delta_co into an equation.

Line 254: Clarify what is meant by "same set of particles in all previous ranges". I think the authors are referring particles within the Rayleigh scattering regime in all previous ranges, rather than the same particles existing in multiple ranges.

Lines 287-297: Consider summarising the details of your experimental configurations in a table.

Lines 290-291: Add relevant citations and justify how these axis ratio and ice fraction values were chosen.

Figure 7: Include details of the two frames of reference in the figure caption. The reader currently has no idea what the xyz and x'y'z' coordinate systems mean.

Lines 352-354: Consider rephrasing this sentence to "Second extrema are challenging to interpret and measure, especially at high altitudes where the SNR is low".

Figure 11: The height axis range is inconsistent in (c), compared to (a) and (b).

Line 368: Consider including the Ze field in Fig 11, since "reflectivity weighting" is mentioned in the interpretation of ZDR not showing significant negative values.

Line 380: How was the "lightest 10%" of the particles determined? Consider elaborating.

Line 431: Define the canting angle (gamma), or consider including it in Figure 7.

Line 434: Units of ms-1 are dimensionally inconsistent with the units of wind shear.

Line 434: Take care of the vertical resolution of the ECMWF IFS and thus the underestimation of vertical wind shear. Shear on the finer scales and turbulence associated with smaller eddies, could also enhance canting angle of a particle.

Line 439: Could there also be cloud charges that are not measured by the lightning sensor?

Lines 442-443: Provide the air temperature at 6000m measured by the radiometer, so readers could understand how cold the environment is with supercooled liquid water.

Line 447: What is the size of "small supercooled liquid water droplets"?

Line 644: Include possible radar scan strategies as ideas for future studies of similar convective storms involving the cloud radar.

Figure A1: Add details to the figure caption on the origin of these plan view radar images. Are these derived using a radar composite of operational radars? What defines the different classes of the colour bar? Is it precipitation rate or reflectivity?

**Technical corrections:**

Line 101: Add brackets around the citation.

Figure 2: Make the black rain rate lines thicker/darker.

Line 171: Citation should not be "GmbH".

Line 606: Add the symbol for spectral differential phase.

**References**

Doviak, R. J. and Zrnić, D. S.: Doppler Radar and Weather Observations, Academic press, London, UK, 1984.

Feist, M. M., Westbrook, C. D., Clark, P.A., Stein, T. H. M., Lean, H. W., and Stirling, A.J.: Statistics of convective cloud turbulence from a comprehensive turbulence retrieval method for radar observations, Quarterly Journal of the Royal Meteorological Society: A journal of the atmospheric sciences, applied meteorology and physical oceanography, 145, 727–744, 2019.

Lund, N. R., MacGorman, D. R., Schuur, T. J., Biggerstaff, M. I., and Rust, W. D.: Relationships between lightning location and polarimetric radar signatures in a small mesoscale convective system, Monthly Weather Review, 137, 4151–4170, 2009.

Mattos, E. V., Machado, L. A. T., Williams, E. R., and Albrecht, R. I.: Polarimetric radar characteristics of storms with and without lightning activity, Journal of Geophysical Research: Atmospheres, 121, 14–201, 2016.

Sokol, Z., Minářová, J., and Fišer, O.: Hydrometeor distribution and linear depolarization ratio in thunderstorms, Remote Sensing, 12, 2144, 2020.

---

## Referee Comment (RC2)

The authors study two lightning cases (the first and fourth cloud of a thunderstorm event) using a dual frequency radar (with unfortunately only one usable frequency of 35 GHz). The authors combine polarimetry and Doppler spectra to present an interesting and novel study which looks at the presence of different hydrometeor types and their alignment in thunderstorm clouds around the times when lightning is occurring. Other processes such as turbulence are also considered.

Although the manuscript is interesting and novel, I think it will benefit from the edits described in this document. In particular, I believe some of the analysis is based on scattering simulations which are not representative of the problem. I would like to see improvements related to that, or at least some clarification on why the authors believe conclusions can be drawn from these scattering simulations. Because of this, my recommendation is a major revision.

**General comments**

1- I feel like the section/subsection organisation in section 5 could be better. The current organisation is:

5 Case analysis
    5.1 First cloud
        5.1.1 Alignment of particles
        5.1.2 Supercooled liquid water
    5.2 Fourth cloud
        5.2.1 Evidence of conical graupel
        5.2.2 Alignment of particles
        5.2.3 Strong updraft and turbulence
        5.2.4 Possibility of chains

In terms of the subsections (5.1 and 5.2) I find it slightly confusing to refer to the first and fourth cloud. Without reading the paper in depth, one might expect to find sections for the second and third clouds too. Perhaps it's worth changing to something like cloud A and cloud B rather than first cloud and fourth cloud, and then just pointing out which clouds they are.
Moreover, I think it might be better if the subsubsections were consistent in subsections 5.1 and 5.2. For example, something along the lines of:

5 Case analysis
    5.1 Cloud A
        5.1.1   Alignment of particles
        5.1.2   Interesting microphysical properties
    5.2 Cloud B
        5.2.1   Alignment of particles
        5.2.2   Interesting microphysical properties

Or maybe 3 subsubsections like particle properties, particle alignment, other interesting findings…

2- In various places you refer to "Mie scattering" and the "Mie scattering regime". I understand that you are talking about non-Rayleigh scattering, and to me describing this as the "Mie scattering regime" is ok (although you could consider "resonance regime"). Referring to "Mie scattering" on its own is slightly questionable to me, as Mie theory is applicable only to spheres. I recommend that in places where you say "Mie scattering" you could change to resonance or non-Rayleigh scattering (unless you are talking about spheres in any of these places…).

**Specific comments**

Line 11: these thunderstorm clouds, or all thunderstorm clouds?

Consider adding Saunders and Wahab reference for lab measurements of chains forming due to electric fields at -12C and -8C:
*Saunders, C. P. R. and N. M. A. Wahab, 1975: The influence of electric fields on the aggregation of ice crystals. J. Meteorol. Soc. Jpn., 53, 121–126.*

Also Stith et al 2002 for chains observed in clouds:
Stith, J. L., J. E. Dye, A. Bansemer, A. J. Heymsfield, C. A. Grainger, W. A. Petersen, and R. Cifelli, 2002: Microphysical Observations of Tropical Clouds. *J. Appl. Meteor. Climatol.*, **41**, 97–117, https://doi.org/10.1175/1520-0450(2002)041<0097:MOOTC>2.0.CO;2.

54/55: Riming doesn't necessarily form large, dense, near spherical particles. For example, if you have a single dendrite monomer which experiences light riming, it won't become much bigger or "near spherical". Perhaps you could rephrase this in a less extreme way, such as "…generally resulting in increased particle size, density and sphericity".

60 and 63: You refer to aggregates as spherical. The appropriate shape approximation of aggregates is something that is still debated (e.g. see the introduction of this paper: https://doi.org/10.5194/amt-14-6851-2021), but usually a spheroidal or ellipsoidal shape would be assumed rather than a sphere. Anyway, in this instance it is probably sufficient to make very slight edits like:
- On line 60 you could maybe just make it more general, e.g. "…form larger particles that *tend to be* more spherical in shape". On line 63 you could just remove the word "spherical".

81: ", which also implies the existence of collisions of hydrometeors." I'm not sure what you mean here. Do you mean that collisions are more likely because there are mixtures of particles with different velocities?

118: Is the 94 GHz completely un-usable? The combination of 35 and 94 would be beneficial for determination of particle size. In fact, unless you are referring to something else, this is

precisely thanks to "complications due to Mie scattering". Can you not just correct for attenuation; I think you can correct for liquid water attenuation because you know the LWP?

112 (Section 2): In Figure 2 you have numbered clouds to identify which ones you are referring to. In the other figures (Figs. 3-5) it would be useful if you highlighted the regions of interest somehow, e.g. by drawing lines or a box around the specific times/heights you are looking at.

150: does the high SLDR occur slightly later than the negative ZDR? Hard to tell on this axis.

240-244: it would be helpful to include approximate values such as Doppler velocities close to 0m/s rather than just referring to the right and left part of the spectrum.

245/470: Your references Lu et al 2016 a and b are the same paper
Did you extract the conical graupel from the database, or does it have the information about negative ZDR and conical graupel in the reference you give?

Maybe the Aydin and Seliga (1984) paper could be useful for you:
https://journals.ametsoc.org/view/journals/atsc/41/11/1520-0469_1984_041_1887_rpbpoc_2_0_co_2.xml

255: could you add some references for "This part of the spectrum is often referred to as the Rayleigh plateau"?

285: as you are saying m_eff can be determined, would it be better for the equation to be m_eff=sqrt(eps_eff)?

291: "different types of particles" is a bit vague, please elaborate

295: Why did you choose these particular values of axis ratio and ice fraction? E.g. aggregates are often considered to have aspect ratio 0.6.

313: Can you provide an explanation of how a horizontally aligned prolate spheroidal model is different to an oblate spheroid?

319 (and elsewhere): Figure caption – change "scatterers" to spheroids

320: Figure 9 and Figure 10 - You don't seem to refer to the reflectivity plots here, either include a reference to them in the discussion, or remove them. This could even be something simple like pointing out that the Mie minimums can be seen in the reflectivity plots.

325: You are talking about ice fractions of 0.8 and 1, right? Thus, I think "For a radius of larger than 2.5 mm representing large aggregates such as graupel" is a bit misleading, as presumably such large ice fractions would *only* represent graupel / some heavily rimed particle and not any other unrimed aggregate type? If you agree, you could just rephrase e.g. "Particles of this size and ice fraction could represent graupel…".

366: why small but not large? Do you just mean that large particles usually aren't oriented by the field? Please elaborate.

368: What is the temperature here, can you get any information on the particles from that? Perhaps you only have temperature measurements in the vertical.

381: this is interesting, how did you obtain the "lightest 10%"?

387: Be consistent with units, either m or km

390: You mention Fig. 16 here but I think you have not yet discussed Figs 14 or 15, maybe the figures should be reordered.

406/408: make the vertical wind shear units the same.

417: here and in the rest of the subsubsection, you refer to "a lightning" or "lightnings". I believe these should be "a lightning stroke" and "lightning strokes".

450: What frequency? The caption of Fig 18 says 94 GHz but line 130 says LWP comes from 31.4GHz?

468: Can you explain what you mean by "Since small particles are more easily aligned by an electric field and they are not aligned in this case,"? Do you just mean that the part of the spectrum corresponding to small particles has sZDR close to zero, implying that the negative sZDR corresponding to the larger particles is not caused by vertical alignment?

There is also lower correlation coefficient in this region, implying mixture of particle shapes.

477: You hypothesize conical graupel, but you then model conical graupel with a spheroid. Please provide an argument for why you think you can get any information about conicality if you are using a spheroid. Perhaps you could just refer to results/simulations in the literature for conical graupel.

486: supported by low correlation coefficient

Figs 22 and 28: It's quite hard to compare the measurements and simulations here due to different y-axis scales and different variables on x-axis. Consider at least making the scale of the y-axis the same for observations and simulations. Can you calculate the fall speeds of your model particles and plot that on x-axis? Also, are your labels and units accurate? E.g. are you plotting reflectivity [dBZ] or spectral reflectivity [dBsZ]?

578: I think you should rephrase this or expand it a bit. At the minute it might be misinterpreted as if chains can only be found if the temperature is below -40. I think what you are trying to say is that although aggregation is usually associated with slightly warmer temperatures (maybe above -25), the presence of electric fields allows chains to form in colder temperature regions, which has been observed at temperatures below -40 (Stith et al

2002; Connolly et al 2005; Gayet et al 2012)? Maybe this could be mentioned around line 65 when you are discussing chains in cold temperatures.

581 onwards: Chains are modelled as prolate particles with axis ratio 7 and ice fraction 0.3. Can you put a reference here as to why you think 0.3 is suitable.

At 17:22, From high ZDR and positive ZDR spectrum you hypothesize that there may be chain-like aggregates. The low correlation coefficient suggests a mixture of particles. If small particles and chains co-exist here, could you comment on why sZDR in Fig 28e would be positive. Do you think if there was a high E-field forming the chains, the small particles would be oriented vertically by the field giving negative sZDR?

You say that you should use axis ratio much greater than 1, but that the maximum you can use is 7 otherwise the code cannot converge. Can you provide more specific details on sizes and axis ratios here please? For example, what are the maximum sizes that have been observed before? You have an example in Fig 1 which is 721 microns. You also say that the individual monomers could be 15-20 microns. Are you then calculating the axis ratio as, for example, 721/15=48? I don't think you can get much information about a chain with axis ratio 48 by modelling it as a prolate particle with axis ratio 7, which you also point out in the next paragraph. You show simulations in Fig. 28 for particles with a much larger maximum dimension of up to 3mm, and then point out that chains of this size have not been observed. Can the code converge if you use smaller sizes less than 1mm with a larger axis ratio?

I personally think the section is not very useful because the scattering simulations kind of go nowhere, then you say an alternative would be to look at data which isn't possible. I think you should either repeat simulations using more representative particles, or just remove the section and add a line saying that the high ZDR could be caused by chains.

640: Is there any scope to track thunderstorm evolution using steerable radars, like they did with weather radars during the WOEST campaign in the UK?

**Technical corrections**

Line 2: a cloud radar
Line 3: a thunderstorm case
Line 4/5: maybe "in the millimeter band", or something similar…kind of sounds like its exactly 1mm now
Line 5: studies *of* thunderstorm clouds
Line 18: phenomenon
Line 38: pellets
Line 70: when *the* ice particle number
        High concentration*s*
Line 73: Evidence
Line 101: need brackets around citations
Around line 276: sometimes use 1 sometimes one
Line 151: associated *with*

Line 216: dealias

Line 232: change *would* align to *can* align?

Line 240: spectra

329 Fig 9 caption: change (a-c) to Panels (a-c). Same for (d-f)

365:  are close to zero and do not show much variation

473: Scattering simulation*s are* (or *a* scattering simulation is)

578: In the case being studied, *the* temperature

580: Scattering simulation*s are*

592: *in* Fig 28c or (Fig 28c)

---

## Author Comment (AC1)

2024-10-14

**Referee comments**

October 14, 2024

**1 Referee 1**

**1.1 General comments**

This paper uses the technique of spectral data from a polarimetric cloud radar to study the passage of convective cells with lightning activity. The study is novel in that spectral polarimetry analysis in the Ka-band is used to study a thunderstorm cloud, so as to identify the different scattering regimes and the suitability of interpreting sZDR extrema. The inclusion of scattering simulations using the T-matrix method supplements the explanation of peaks and troughs in the spectral polarimetric data. In particular, the study makes convincing interpretations for the presence of conical graupel and vertical alignment of particles prior to a lightning strike, which are consistent with past literature. The phenomena of supercooled liquid water, chains and strong updrafts associated with in-cloud turbulence are also explored. The manuscript is mostly well written, with a detailed set of results and scientific explanations that make it suitable for the scope of AMT. The novelty of using spectral polarimetry of a cloud radar on a thunderstorm case is well appreciated, especially when supplemented with sufficient results that ultimately lead to valid interpretations of the hydrometeor type and the sound conclusions reached regarding their presence in convective cells with lightning. There are several clarifications that are needed in the first two sections of the manuscript, alongside suggested reduction to the results section. Although the specific comments are mostly minor, they combine to push me to recommend a major revision.

*The authors express their gratitude to Referee 1 for the thorough and meticulous review, which provided many valuable suggestions to enhance and improve the article. Referee 1 can find our responses in blue (italic) below.*

**1.2 Specific comments**

**Major**

1. Although the authors have made it clear that there could be differential attenuation due to attenuated reflectivity from the presence of rain drops for 17:24 to 17:30UTC, there still seems to be areas of enhanced calibrated ZDR just above 2000m of height in Fig. 19a. Could the authors explain whether they have taken into account differential attenuation when calibrating ZDR? If so, how? If not, what is the estimated magnitude of differential attenuation? How would this change the calibrated ZDR values and alter the interpretation of the results in Sections 5.2.1 and 5.2.4 regarding the evidence of conical graupel and possibility of chains.
*The ZDR values are not corrected for differential attenuation. The differential attenuation relating to the presence of oblate raindrops leads to a decrease of the measured ZDR values and can even lead to negative measured ZDR values, which is an issue, particularly if we want to detect the presence of conical graupels. Therefore, we added the following small paragraph for more information:*
*Lines 572-581: "The presence of liquid water introduces an additional challenge, namely differential attenuation, which influences the $sZ_{DR}$ values. While no direct measurements of the Rain Drop Size Distribution (RDSD) are available, a simulation can provide an estimate of the differential attenuation. For this purpose, the convective RDSD typical of the Netherlands, based on disdrometer data from Gatidis et al. 2024, is considered. The corresponding intercept parameter*

*$N_w$ equals 1300 $mm^{-1}$ $m^{-3}$ and the mass-weighted mean diameter $D_m$ is 2.2 mm. The shape parameter, derived using the $\mu$-$\lambda$ relationship from the same study, along with the shape-size relationship from Unal and van den Brule 2024, is applied. Consequently, in rainfall, the differential reflectivity is estimated at 0.15 dB, and the one-way differential attenuation at 0.06 dB $km^{-1}$. Except near the edges of the precipitation, $Z_{DR}$ measurements show an increase from 0 dB to 0.2 dB as height decreases from 3000 m to 2200 m. Thus, the two-way differential attenuation contribution is expected to be low, at less than 0.12 dB, and does not significantly affect the interpretation of the results discussed.*

2. There is a good variety of past literature included within the introduction section. The authors should consider an additional discussion section to the quantitatively compare with past studies if possible. The authors could consider the following questions including, but not limited to: How do the ZDR values of vertically aligned particles in this study comparable with those in Lund et al. (2009)? How does vertical alignment of ice up to 4 seconds before lightning comparable with past studies?
*ZDR values are not directly comparable because the frequency differs (Lund, S-band). Previous studies have not investigated the time that vertical alignment of ice occurred. Lund 2009 analysed ZDR at certain instants representative of particular stages of the thunderstorm, Sokol 2020 matched high LDR with a period with intense lightning, Mattos 2016 looked at mean profiles of ZDR / KDP at times with different lightning intensities. We have not seen any studies that analysed the vertical alignment of ice at the second level like we do.*

3. I appreciate the depth of the scientific deductions and explanations provided for the physical interpretation of the alignment of particles in Section 5.1.1. However, this has made the section rather long. To focus the discussion on the most probable reason for vertical alignment, the authors could consider shortening the paragraphs between lines 393 and 437 and potentially including some of the Doppler flipping figure and explanation as supplementary material.
*The authors consider important to discuss all the possible causes of particle alignment in the first study case (thunderstorm cloud 1). However, the paragraphs between former lines 393 and 437 (45 lines) have been shortened to 32 lines.*

**Minor**

Line 2: I agree that using a cloud radar to study thunderstorms mitigates the need to fly into one to take in-situ measurements. However, there is no mention of arrival time difference networks for lightning detection in the abstract. Yet, the authors do take advantage of the BLIDS system. Please mention this in the abstract.
*Added in Lines 4-5: "The time and location of individual lightning strikes are determined using the BLIDS system, operated by SIEMENS, which is based on the Time-Of-Arrival principle."*

Line 15: "reflected by the..." is confusing, especially since you mention spectrum flipping later on. I would just us "shown by the..." *Corrected.*

Line 28: Include a diagram to supplement the explanations of the "numerous charging mechanisms".
*The authors are generally in favor of illustration or diagram. However, we think that there is already a good summary of the three major categories of charging mechanisms, with references.*

Lines 32-36: Some explanation of the "inconsistencies" and "quantitatively unrealistic ineffective" mechanisms of charging would be appreciated here.
*Expanations are added in Lines 34-35 and Lines 37-38.*
*Lines 34-35: "However, numerous investigators such as Chiu and Klett (1976) have found inconsistencies between this mechanism and observations, such as opposite cloud polarity if the cloud forms close to the ground."*
*Lines 37-38: "However, many studies have shown that these mechanisms are quantitatively unrealistic or ineffective since it is only effective when the electric field is below typical values for the initiation of*

*lightning in thunderstorms (Pruppacher and Klett, 1980; Wang, 2013)."*

Line 44: Which part of the updraft column? Consider giving an indication of a height/temperature range, which would quantify what is meant by "where the temperature is low".
*Lines 46-47: "where the temperature is low" is changed to "where temperature is below $T_R$".*

Line 46: What are the terminal velocities of the ice crystals that are "thrown upwards" in updrafts?
*Lines 47-49. The sentence is revised to:*
*"The negatively charged graupel will fall at the periphery of the column where the updraft is weak, while the positively charged ice crystals with negligible fall velocity will be thrown upwards."*

Lines 48-49: The authors mention that "all charging mechanisms above could contribute to certain extent at some time to cloud charging", but they also comment that some of these mechanisms are unrealistic earlier in the same paragraph. Please rewrite this paragraph so it is not self-contradicting.
*Lines 50-53. The sentence is revised to:*
*"Although non-inductive charging due to the collision of graupel and ice crystals best explains tripolar cloud structure, it should be noted that all charging mechanisms above could contribute to certain extent at some time to cloud charging even though certain mechanisms alone would produce inadequate or reversed charges (Pruppacher and Klett, 1980)."*

Line 79: Provide temperature information if possible for the height range of 4-7 km in the Sokol et al. (2020) study, so as to be consistent with the descriptions of Mattos et al. (2016) earlier in the paragraph.
*The temperature information is added.*
*Lines 86-88: "With Ka-band cloud radar, Sokol et al. (2020) identified a mixture of hydrometeors at an elevation of 4–7 km (from 6.6 to 27°C) with a predominance of ice and snow particles and graupel based on the terminal velocities of different hydrometeors."*

Line 81 and 558: A high Doppler spectrum width is not a sufficient condition to highlight the co-existence of different hydrometeor types. Such a measurement could hint at strong shear and/or turbulence. See Feist et al. (2019) for details before clarifying the interpretation of results from Sokol et al. (2010).
*Lines 88-89. The sentence is revised to:*
*"The coexistence of different types of hydrometeors is supported by the measured high Doppler spectrum width."*

Lines 107-109: Great to see the authors have considered past work in light of the novelty of using the spectral polarimetry technique in the Ka-band to study storms. However, I would like to see at least a brief literature review that mentions how polarimetric Doppler spectra have been used in other past studies. Were they for non-thunderstorm cases? Used other radar wavelengths? Did they also use the spectra to identify the transition between Rayleigh and Mie scattering regimes? Answering these questions will strengthen this paper on how novel the work is!
*The authors believe this is not the ideal section for a brief literature review on the use of polarimetric Doppler spectra. Our focus remains on thunderstorms (the medium) and polarimetric Doppler spectra (the methodology). Nonetheless, we have improved this section and included an additional reference.*

Line 111: Add a paragraph to introduce the remaining sections of the manuscript.
*Added.*

Line 116: Refer to Figure A1 immediately after constant azimuth of 282deg is mentioned.
*Done.*

Line 128: Consider adding a short explanation regarding how SLDR is an appropriate proxy for LDR.
*SLDR explanation added.*
*Lines 148-149: "Compared to $L_{DR}$, $SL_{DR}$ in the STSR mode loses the direct mean canting angle information due to the inability to acquire cross-polar measurements, but retains information on the*

*variance of the canting angles and axis ratios."*
*This statement is also added in the conclusion.*

Line 133: What is the initialisation time of the IFS forecast(s) used in this case study? Are outputs available hourly? Since the case spans longer than an hour, were outputs valid at different times used? Did the authors choose the closest valid time of the output to that of the radar observation and does the IFS forecast have to precede the radar observation time? Please include these details within the description of the IFS.
*The outputs are available hourly. The output at 16h for 1600-1700 and the output at 17h for 1700-1800 are used. Details are added in lines 180-181 and lines 586-587.*
*Lines 180-181: "The mean vertical velocity in Fig. 5(a) eliminates from the measured mean Doppler velocity the contribution of horizontal wind in the same hour obtained from ECMWF model forecast initialised at 17 June 2021 12:00 UTC."*
*Lines 586-587: "The vertical velocity is estimated by assuming uniform horizontal wind predicted by the ECMWF model in the same hour."*

Line 136: Include a citation to the vertical coordinate system used by the IFS.
*Added.*
*Lines 155-156: "The model uses an eta-coordinate system, with vertical resolution of the first 10000 m ranging from around 20 m near the surface to around 300 m at the top."*

Lines 137-138: I don't think it is necessary to mention the act of "clicking on the lightning stroke".
*This is removed.*

Lines 148-149: It could be worth mentioning that although the bright band cannot be clearly seen in the Ze field, there are some signs of it in the dual-pol fields in Fig 3.
*Lines 169-170 are added:"However, after 17:15, a brief indication of a melting layer can be observed using the radar variables, $Z_{DR}$, $SL_{DR}$ and $\rho_{hv}$."*

Lines 163-164: If vertical air velocity is obtained from the Doppler velocity bin with the smallest value, why do the colours in Fig. 5b hint at much larger vertical air velocity values than the mean vertical velocities in Fig 5a?
*The authors think that this visual effect is due to the different velocity scaling in Fig. 5a and Fig. 5b.*

Line 169: Doppler spectra is not broadened purely by turbulence, but also by shear. See Chapter 5 of Doviak and Zrnić (1984) and Feist et al. (2019) for details.
*Lines 188-190: "This could mean that there is a wide variety of particles within the radar resolution volume or the Doppler spectrum is broadened by turbulence or shear (Doviak and Zrnic, 2006; Feist et al. 2019)."*

Line 174: The use of "perpendicular distance" is potentially confusing here. Why not introduce this term earlier in line 173 when describing distance of the first cloud from the radar?
*The text is revised.*
*Lines 194-198: "For the first cloud, lightning occurred near the line of sight at more than 10 km away from the radar. For the second cloud, lightning occurred at the ranges 3 to 8 km with a cross-range varying from 1 to 10 km. The third cloud only produced two lightning strikes after passing through the line of sight of the radar. The fourth cloud produced a large number of lightning strikes near the radar line of sight from less than 1 km to more than 15 km along-range."*

Line 187: Justify why integrated radar variables are computed from Level 0 data, rather than directly using Level 1 data.
*The paragraph is rewritten.*
*Lines 207-210: "This research utilized spectral polarimetric radar variables derived directly from the Level 0 data. Consequently, the majority of the integrated radar variables were also computed from Level 0 data. This approach facilitates consistency checks between Level 0 and Level 1 data, enables spectral domain filtering when necessary, and allows for the dealiasing of Doppler spectra prior to the*

*calculation of Doppler moments."*

Line 202: What were the original errors without the extra polarimetric calibration?
*The original errors are now indicated.*
*Lines 224-225: "This procedure resulted in reducing the expected error associated with $Z_{DR}$ and $\Psi_{DP}$ from 0.18 dB to 0.05 dB and from 1.6° to 0.6° respectively."*

Lines 214-215: What is the definition of "North"? Grid North or True North? It is best to clarify this. From experience, there could be differences by up to a few degrees between grid north and true north, leading to significant differences in collocating radar echoes if the definitions are used interchangeably.
*True North. That is explicitly mentioned in the text.*
*Lines 237-238: "D is the wind direction and $\phi$ is the azimuth angle of the radar beam, both being relative to True North."*

Line 227: Without having seen Fig 6 yet, it could be confusing as to what is meant by "rightmost valid bin". Worth clarifying that this means bin with the highest positive Doppler velocity value.
*The authors leave the sentence unchanged. This Doppler bin does not always correspond to the highest positive Doppler velocity value. It can correspond as well to a negative Doppler velocity.*

Lines 251-252: Convert the relation of psi_DP with phi_DP and delta_co into an equation.
*Done in Line 277.*

Line 254: Clarify what is meant by "same set of particles in all previous ranges". I think the authors are referring particles within the Rayleigh scattering regime in all previous ranges, rather than the same particles existing in multiple ranges.
*The sentence was improved.*
*Lines 278-279: "In the Rayleigh scattering regime, where $\delta_{co}$ is zero, the spectral differential phase shift at a fixed range remains constant because the electromagnetic wave at both polarizations has passed through the same particles in all preceding ranges."*

Lines 287-297: Consider summarising the details of your experimental configurations in a table.
*The authors think that the simulation configurations are clearly mentioned in the text and the figure captions.*

Lines 290-291: Add relevant citations and justify how these axis ratio and ice fraction values were chosen.
*Citation added in the second sentence of the paragraph.*
*Lines 313-315: "The axis ratio and ice fraction of the particles in the simulation experiments were chosen according to the data given in Spek and al. (2008)."*

Figure 7: Include details of the two frames of reference in the figure caption. The reader currently has no idea what the xyz and x'y'z' coordinate systems mean.
*The caption is rephrased with details of the two frames:*
*"Definition of Euler angles $\alpha$ and $\beta$. The $xyz$ coordinate frame has the $z$ axis aligned with the radar's zenith direction. The rotated frame is denoted as $x'y'z'$, corresponding to the particle's orientation. Starting from the $xyz$ frame, a rotation by angle $\alpha$ around the $z$ axis results in the intermediate frame $x'y_1z$. This is followed by a rotation by angle $\beta$ around the $x'$ axis to achieve the final $x'y'z'$ frame."*

Lines 352-354: Consider rephrasing this sentence to "Second extrema are challenging to interpret and measure, especially at high altitudes where the SNR is low".
*Thank you for rephrasing the sentence. Done in Lines 382-383.*

Figure 11: The height axis range is inconsistent in (c), compared to (a) and (b).
*Figure modified.*

Line 368: Consider including the Ze field in Fig 11, since "reflectivity weighting" is mentioned in the

interpretation of ZDR not showing significant negative values.

*The reflectivity is plotted in Figure 2 and for further interpretation of ZDR, it is necessary to investigate sZDR (spectral ZDR). Therefore the authors think that it is not necessary to include an extra figure related to the reflectivity profiles.*

*We rephrased the first paragraph (Lines 396-402) for improved clarity:*

*"From Fig. 11(a) and (b), intriguing polarimetric signatures can be observed within the cloud. Figure 11(a) illustrates that $Z_{DR}$ values are near zero with minimal variation. Conversely, Fig. 11(b) reveals a cluster of negative $K_{DP}$ values between 7600 m and 9300 m, suggesting the alignment of non-spherical small ice particles. If these small ice particles are present in sufficient concentration, $K_{DP}$ would become negative. The large ice particles, on the other hand, are expected to be slightly non-spherical, which leads to a small contribution to $K_{DP}$, and may not align with an electric field unless it is sufficiently strong. Because $Z_{DR}$ is reflectivity-weighted, large ice particles significantly influence $Z_{DR}$, which likely explains why $Z_{DR}$ does not exhibit significant negative values."*

Line 380: How was the "lightest 10%" of the particles determined? Consider elaborating.

*We replaced "lightest" by "smallest" in the text and Fig. 13.*

*Lines 413-415: Figure 13 shows the mean $sZ_{DR}$ of the smallest 10% of the particles in each radar resolution volume at the three time instants. This is achieved by averaging $sZ_{DR}$ over the rightmost 10% of the Doppler bins.*

Line 431: Define the canting angle (gamma), or consider including it in Figure 7.

*Eq. 13: $\gamma$ is replaced by $\beta$, which is defined in Fig.7.*

Line 434: Units of ms-1 are dimensionally inconsistent with the units of wind shear.

*This is corrected.*

*Lines 454-455: "Using the vertical shear $s = \frac{dv_H}{dz} = 4 \ m \ s^{-1} \ km^{-1} = 0.004 \ s^{-1}$ and terminal velocity of 2 $m \ s^{-1}$, the canting angle is estimated at 0.05°, which is negligible."*

Line 434: Take care of the vertical resolution of the ECMWF IFS and thus the underestimation of vertical wind shear. Shear on the finer scales and turbulence associated with smaller eddies, could also enhance canting angle of a particle.

*We added the following the sentences:*

*Lines 455-457: "Even considering underestimation due to model resolution, achieving significant canting would require a much higher shear of 4.9 $s^{-1}$, making wind shear an unlikely cause of the observed negative $sZ_{DR}$ values."*

Line 439: Could there also be cloud charges that are not measured by the lightning sensor?

*The lightning sensor only record lightning strikes, not cloud charges. We already mentioned the case that there could be charges (i.e. electric field) that may not be strong enough to trigger lightning, and the case that some lightning strikes may not be measured by the sensor, which cover the presence of charges that does not give measurable as well as the limitation of the sensor.*

Lines 442-443: Provide the air temperature at 6000m measured by the radiometer, so readers could understand how cold the environment is with supercooled liquid water.

*The air temperature at 6000 m is added.*

*Lines 463-465:"Another interesting feature observed in this cloud is the possible presence of supercooled liquid water. From 16:20:21 to 16:21:15 UTC, spectrograms of reflectivity show a separate mode of particles on the right side of the spectrum at around 6000 m (see Fig. 16(a)), where air temperature measured by the microwave radiometer is around $-12.5$ °C."*

Line 447: What is the size of "small supercooled liquid water droplets"?

*The authors removed "small".*

Line 644: Include possible radar scan strategies as ideas for future studies of similar convective storms involving the cloud radar.

*The authors already included one cloud radar scan strategy in the conclusion.*

*Lines 683-684 : "A more appropriate radar measurement mode for studying thunderstorm clouds would be azimuth scan (PPI) with the constant elevation of 45°."*

Figure A1: Add details to the figure caption on the origin of these plan view radar images. Are these derived using a radar composite of operational radars? What defines the different classes of the colour bar? Is it precipitation rate or reflectivity?

*Unfortunately, no information about this is given on the webpage `https://meteologix.com/nl/stormradar/utrecht/20210618-1650z.html`. However, by looking through other products provided on the website, we find that the Radar SD Netherlands (2.5km) looks quite similar `https://meteologix.com/nl/radar-hd/utrecht/20210618-1650z.html`. Following this, marginal refers to 0.1-0.4 mm/h, light is 0.4-2.3 mm/h, moderate is 2.3-12 mm/h, heavy is 12-44 mm/h, very heavy is 44-146 mm/h and extreme/hail is 146-491 mm/h. However, there is no explicit mentioning that the source of the radar images we used is Radar SD Netherlands (2.5km), and there is also no information about the origin of the data of Radar SD Netherlands (2.5km) (we suspect it is a composite of the radars at Den Helder and Herwijnen `https://meteologix.com/nl/radar-hd/utrecht/20210618-1649z_srcnl-zsweep-1km-hw.html`. There is a similar dataset in KNMI `https://dataplatform.knmi.nl/dataset/radar-reflectivity-composites-2-0`, but we did not make the comparison). We chose the currently used radar images over the Radar SD Netherlands ones simply because they show the locations of lightning strikes on the same images.*
*Summarizing we indicate now that the different classes of the colour bar relate to precipitation rate.*

**1.3 Technical corrections**

Line 101: Add brackets around the citation. *Brackets added.*

Figure 2: Make the black rain rate lines thicker/darker. *Figure modified.*

Line 171: Citation should not be "GmbH". *Citation modified.*

Line 606: Add the symbol for spectral differential phase. *Symbol added.*

---

## Author Comment (AC2)

**Referee comments**

October 14, 2024

**1 Referee 2**

The authors study two lightning cases (the first and fourth cloud of a thunderstorm event) using a dual frequency radar (with unfortunately only one usable frequency of 35 GHz). The authors combine polarimetry and Doppler spectra to present an interesting and novel study which looks at the presence of different hydrometeor types and their alignment in thunderstorm clouds around the times when lightning is occurring. Other processes such as turbulence are also considered.

Although the manuscript is interesting and novel, I think it will benefit from the edits described in this document. In particular, I believe some of the analysis is based on scattering simulations which are not representative of the problem. I would like to see improvements related to that, or at least some clarification on why the authors believe conclusions can be drawn from these scattering simulations. Because of this, my recommendation is a major revision.

*The authors express their gratitude to Referee 2 for the thorough and meticulous review, which provided many valuable suggestions to enhance and improve the article. Referee 2 can find our responses in blue (italic) below.*
*Regarding the scattering simulation, for this initial study of thunderclouds based on radar observations at mm-wavelengths and spectral polarimetry, the authors aimed to use existing scattering codes or databases. The use of the T-matrix code was intended to aid in interpreting the spectral polarimetric measurements, and we are aware of its limitations. Specifically, we sought to gain a preliminary understanding of the variability and sensitivity of $sZ_{DR}$ and $s\delta_{co}$ in relation to axis ratio, ice fraction, and canting angle.*

**1.1 General comments**

1. I feel like the section/subsection organisation in section 5 could be better. The current organisation is:

   5 Case analysis
         5.1 First cloud
               5.1.1 Alignment of particles
               5.1.2 Supercooled liquid water
         5.2 Fourth cloud
               5.2.1 Evidence of conical graupel
               5.2.2 Alignment of particles
               5.2.3 Strong updraft and turbulence
               5.2.4 Possibility of chains

   In terms of the subsections (5.1 and 5.2) I find it slightly confusing to refer to the first and fourth cloud. Without reading the paper in depth, one might expect to find sections for the second and third clouds too. Perhaps it's worth changing to something like cloud A and cloud B rather than first cloud and fourth cloud, and then just pointing out which clouds they are.

   *Considering the figures showing the four thunderstorm clouds (Figs 2-5), which relate to the weather radar images in Appendix A, the authors would like to keep the terminology first cloud and fourth cloud for clarity.*

   Moreover, I think it might be better if the subsubsections were consistent in subsections 5.1 and 5.2. For example, something along the lines of:

5 Case analysis
    5.1 Cloud A
        5.1.1 Alignment of particles
        5.1.2 Interesting microphysical properties
    5.2 Cloud B
        5.2.1 Alignment of particles
        5.2.2 Interesting microphysical properties

*The authors agree with the proposed restructure of the subsections, which are now:*

*5 Case analysis*
    *5.1 First cloud*
        *5.1.1 Alignment of particles*
        *5.1.2 Interesting microphysical properties*
            *5.1.2.1 Supercooled liquid water*
    *5.2 Fourth cloud*
        *5.2.1 Alignment of particles*
        *5.2.2 Interesting microphysical properties*
            *5.2.2.1 Evidence of conical graupel*
            *5.2.2.2 Strong updraft and turbulence*
            *5.2.2.3 Possibility of chains*

2. In various places you refer to "Mie scattering" and the "Mie scattering regime". I understand that you are talking about non-Rayleigh scattering, and to me describing this as the "Mie scattering regime" is ok (although you could consider "resonance regime"). Referring to "Mie scattering" on its own is slightly questionable to me, as Mie theory is applicable only to spheres. I recommend that in places where you say "Mie scattering" you could change to resonance or non-Rayleigh scattering (unless you are talking about spheres in any of these places...).

   *The authors kept "Mie scattering regime" and replaced "Mie scattering" by "non-Rayleigh scattering" or "resonance" in function of the context.*

**1.2 Specific comments**

Line 11: these thunderstorm clouds, or all thunderstorm clouds?
*The sentence was rewritten as: "From the results, there is a high chance that supercooled liquid water and conical graupel are present in the investigated thunderstorm clouds."*

Consider adding Saunders and Wahab reference for lab measurements of chains forming due to electric fields at -12C and -8C: Saunders, C. P. R. and N. M. A. Wahab, 1975: The influence of electric fields on the aggregation of ice crystals. J. Meteorol. Soc. Jpn., 53, 121–126.

Also Stith et al 2002 for chains observed in clouds: Stith, J. L., J. E. Dye, A. Bansemer, A. J. Heymsfield, C. A. Grainger, W. A. Petersen, and R. Cifelli, 2002: Microphysical Observations of Tropical Clouds. J. Appl. Meteor. Climatol., 41, 97–117, https://doi.org/10.1175/1520-0450(2002) 041<0097:MOOTC>2.0.CO;2.
*The authors would like to thank Referee 2 for providing complementary references. Lines 65-70 are rewritten to include the results of these studies.*

54/55: Riming doesn't necessarily form large, dense, near spherical particles. For example, if you have a single dendrite monomer which experiences light riming, it won't become much bigger or "near spherical". Perhaps you could rephrase this in a less extreme way, such as "...generally resulting in increased particle size, density and sphericity".
*Rephrased. Lines 58-59.*

60 and 63: You refer to aggregates as spherical. The appropriate shape approximation of aggregates is something that is still debated (e.g. see the introduction of this paper: https://doi.org/10.5194/ amt-14-6851-2021), but usually a spheroidal or ellipsoidal shape would be assumed rather than a sphere. Anyway, in this instance it is probably sufficient to make very slight edits like:

- On line 60 you could maybe just make it more general, e.g. "...form larger particles that tend to be more spherical in shape". On line 63 you could just remove the word "spherical".
*Thank you for the proposed edits.*
*Lines 64-68: "Aggregation occurs when ice crystals collide with each other and form larger particles that tend to be more spherical in shape. Various lab measurements have demonstrated that when an electric field of more than around 50 kV $m^{-1}$ is present, aggregation of ice crystals may be enhanced due to attractive electrical forces induced between neighbouring conducting crystals, forming elongated chains rather than almost spherical clusters (Conolly et al. 2005)."*

81: ", which also implies the existence of collisions of hydrometeors." I'm not sure what you mean here. Do you mean that collisions are more likely because there are mixtures of particles with different velocities?
*We suppressed this part of the sentence.*

118: Is the 94 GHz completely un-usable? The combination of 35 and 94 would be beneficial for determination of particle size. In fact, unless you are referring to something else, this is precisely thanks to "complications due to Mie scattering". Can you not just correct for attenuation; I think you can correct for liquid water attenuation because you know the LWP?
*The authors agree with Referee 2 and may revisit the combination of 35 and 94 GHz using Dual-Wavelength Ratio. The main issue encountered was a significant decrease of sensitivity for the Doppler spectra measurements at 94 GHz at large heights.*
*Lines 135-137: "In this study of thunderstorm clouds, only the 35 GHz data is used since there are numerous issues associated with the 94 GHz data including significant attenuation, less sensitivity at large heights, Doppler aliasing and complications due to resonance."*

112 (Section 2): In Figure 2 you have numbered clouds to identify which ones you are referring to. In the other figures (Figs. 3-5) it would be useful if you highlighted the regions of interest somehow, e.g. by drawing lines or a box around the specific times/heights you are looking at.
*The authors think that based on Fig. 2, the clouds 1, 2, 3 and 4 can be clearly identified in Figs. 3-5.*

150: does the high SLDR occur slightly later than the negative ZDR? Hard to tell on this axis.
*This case was discarded and not studied in details. The authors focused on selecting appropriate times with no precipitation reaching the ground to avoid misinterpretation.*

240-244: it would be helpful to include approximate values such as Doppler velocities close to 0m/s rather than just referring to the right and left part of the spectrum.
*The authors did not change the formulation. The Doppler velocity measurement contains two contributions, one caused by the fall velocities and one due to the radial wind. Therefore, it can occur that Doppler velocities related to the smallest particles have significant positive values, while the Doppler velocities related to the largest particles have values near 0 m/s.*

245/470: Your references Lu et al 2016 a and b are the same paper
Did you extract the conical graupel from the database, or does it have the information about negative ZDR and conical graupel in the reference you give?
*Duplicated reference is fixed. The information about negative ZDR and conical graupel is not given in the paper Lu et al. 2016. Therefore, we obtained Zdr of conical graupel from the database. This is made clear in the following lines:*
*Lines 61-63: "Scattering simulations carried out by Oue et al. (2015) and data from the scattering database created by Lu et al. (2016) indicate that conical graupel can produce negative differential reflectivity ($Z_{DR}$) values at X-, Ka- and W-band."*
*Lines 270-271: "On the other hand, based on the database described by Lu et al. (2016), negative $Z_{DR}$ for large particles only may indicate the presence of conical graupel."*

Maybe the Aydin and Seliga (1984) paper could be useful for you: https://journals.ametsoc.org/view/journals/atsc/41/11/1520-0469_1984_041_1887_rpbpoc_2_0_co_2.xml
*Thank you for the reference. The paper provides zdr of conical graupel at 10 cm wavelength.*

255: could you add some references for "This part of the spectrum is often referred to as the Rayleigh plateau"?

*A reference was added.*

285: as you are saying m_eff can be determined, would it be better for the equation to be m_eff=sqrt(eps_eff)?

*The equation was modified to better relate to the text.*

291: "different types of particles" is a bit vague, please elaborate

*The text was modified.*
*Lines 315-317: "In the first experiment, the axis ratio of spheroids with a zero mean canting angle was varied from 0.1 to 1.2. This range encompasses the axis ratios of plates, dendrites, aggregates, and graupel. The ice fraction was held constant at 0.6, representing the average ice fraction for the aforementioned ice particles."*

295: Why did you choose these particular values of axis ratio and ice fraction? E.g. aggregates are often considered to have aspect ratio 0.6.

*Citation added in the second sentence of the paragraph.*
*Lines 313-315: "The axis ratio and ice fraction of the particles in the simulation experiments were chosen according to the data given in Spek and al. (2008)."*
*Further, the authors replaced "aggregates" by "slightly oblate aggregates" in Line 321 and caption of Figure 10.*

313: Can you provide an explanation of how a horizontally aligned prolate spheroidal model is different to an oblate spheroid?

*Thank you for spotting this inconsistency. In section 4, "horizontally aligned spheroids" has been replaced by "spheroids with zero mean canting angle" in the text, the figures caption and Table 2.*
*We added the following sentence in the subsection 4.3 (canting angles):*
*Lines 358-360: "A zero mean canting angle corresponds to oblate spheroids being horizontally aligned and prolate spheroids being vertically aligned. To represent prolate particles as horizontally aligned, they are modeled with a mean canting angle of 90 degrees."*

319 (and elsewhere): Figure caption – change "scatterers" to spheroids

*Done.*

320: Figure 9 and Figure 10 - You don't seem to refer to the reflectivity plots here, either include a reference to them in the discussion, or remove them. This could even be something simple like pointing out that the Mie minimums can be seen in the reflectivity plots.

*There is now reference to Figures 9-10 first row (reflectivity) in Lines 347 and 357.*

325: You are talking about ice fractions of 0.8 and 1, right? Thus, I think "For a radius of larger than 2.5 mm representing large aggregates such as graupel" is a bit misleading, as presumably such large ice fractions would only represent graupel / some heavily rimed particle and not any other unrimed aggregate type? If you agree, you could just rephrase e.g. "Particles of this size and ice fraction could represent graupel...".

*Rephrased.*
*Lines 350-353: "When ice fraction is large (0.8 and 1), the sign of $Z_{DR}$ flips soon after reaching the first extremum, and the trend is rather unpredictable. For particles of this ice fraction with radius larger than 2.5 mm, which could represent graupel, significant negative (positive) values could be obtained, which increases the interpretation challenge."*

366: why small but not large? Do you just mean that large particles usually aren't oriented by the field? Please elaborate.

*We rephrased the paragraph to improve its clarity.*
*Lines 396-402 : "From Fig. 11(a) and (b), intriguing polarimetric signatures can be observed within the cloud. Fig. 11(a) illustrates that $Z_{DR}$ values are near zero with minimal variation. Conversely, Fig.*

*11(b) reveals a cluster of negative $K_{DP}$ values between 7600 m and 9300 m, suggesting the alignment of non-spherical small ice particles. If these small ice particles are present in sufficient concentration, $K_{DP}$ would become negative. The large ice particles, on the other hand, are expected to be slightly non-spherical, which leads to a small contribution to $K_{DP}$, and may not align with an electric field unless it is sufficiently strong. Because $Z_{DR}$ is reflectivity-weighted, large ice particles significantly influence $Z_{DR}$, which likely explains why $Z_{DR}$ does not exhibit significant negative values."*

368: What is the temperature here, can you get any information on the particles from that? Perhaps you only have temperature measurements in the vertical.
*The temperature from 7600 to 9300 m measured by the microwave radiometer is around -24 to -41 °C. However, it is challenging to get information on the particle types from the temperature since particles formed at certain temperature ranges are moved around by updrafts and downdrafts, and the temperature is measured vertically, so it is not the actual temperature in the measured cloud at 45 deg. elevation.*

381: this is interesting, how did you obtain the "lightest 10%"?
*We replaced "lightest" by "smallest" in the text and Fig. 13.*
*Lines 413-415: "Figure 13 shows the mean $sZ_{DR}$ of the smallest 10% of the particles in each radar resolution volume at the three time instants. This is achieved by averaging $sZ_{DR}$ over the rightmost 10% of the Doppler bins."*

387: Be consistent with units, either m or km
*Corrected.*
*Lines 420-422: "Negative $K_{DP}$ appears at a distance of 7600 m to 9300 m away from the radar, but the lightning strikes occurred at least 13000 m away from the radar."*

390: You mention Fig. 16 here but I think you have not yet discussed Figs 14 or 15, maybe the figures should be reordered.
*Fig. 16 is dedicated to the section 5.1.2.1 Supercooled liquid water. Therefore, we must have it after Figs. 14-15.*

406/408: make the vertical wind shear units the same.
*Done.*
*Lines 436-439: "For a 10 m s$^{-1}$ spectrum width and $V_t = 2$ m s$^{-1}$ corresponding to the upper bound of the terminal velocity of plates (Spek et al. 2008), a shear of approximately 25000 m s$^{-1}$ km$^{-1}$ would be required to flip the spectrum, much larger than the 4 m s$^{-1}$ km$^{-1}$ observed from 7500 m to 10000 m in ECMWF data shown in Fig. 15(c)."*

417: here and in the rest of the subsubsection, you refer to "a lightning" or "lightnings". I believe these should be "a lightning stroke" and "lightning strokes".
*"stroke" was changed into " lightning strike".*

450: What frequency? The caption of Fig 18 says 94 GHz but line 130 says LWP comes from 31.4GHz?
*Thank you for noticing the discrepancy. That is now corrected in the caption of Fig. 18 and the text (Line 150). We used the embedded 89 GHz passive channel.*

468: Can you explain what you mean by "Since small particles are more easily aligned by an electric field and they are not aligned in this case,"? Do you just mean that the part of the spectrum corresponding to small particles has sZDR close to zero, implying that the negative sZDR corresponding to the larger particles is not caused by vertical alignment?
*We rephrased the sentence:*
*Lines 542-546: "When negative $sZ_{DR}$ appears on the left part of the spectrum, the $sZ_{DR}$ on the right part of the spectrum is close to zero. The observed negative $sZ_{DR}$ values on the left part of the spectrum may suggest the presence of conical graupel (Lu et al. 2016), as smaller particles, which are typically more easily aligned by an electric field, do not appear to be aligned in this case, as indicated by the absence of slightly negative $sZ_{DR}$ values."*

There is also lower correlation coefficient in this region, implying mixture of particle shapes.
*That is true. We have therefore mentioned this observation.*

477: You hypothesize conical graupel, but you then model conical graupel with a spheroid. Please provide an argument for why you think you can get any information about conicality if you are using a spheroid. Perhaps you could just refer to results/simulations in the literature for conical graupel.
*To strengthen the discussion related to the simulation, we analysed as well Lu et al. 2016 database results for conical graupels. We used 8 points corresponding to the radius, 0.2 mm, 0.3 mm, 0.4 mm, 0.5 mm, 1 mm, 1.5 mm, 2 mm and 2.5 mm and an approximate density corresponding to ice fraction 0.6. For the cone angles we selected 40 and 50 deg. In that case, simulations (T matrix prolate particles), (conical graupels Lu database) and observations provide comparable results in terms of trends for $Z_{DR}$ and $\delta_{co}$. Fig. 22 includes now Lu results about conical graupels.*

486: supported by low correlation coefficient
*added in the text.*
*Lines 574-576: "This suggests that the two peaks in spectral reflectivity represent two particle populations, the left peak corresponding to conical graupel and the right peak relating to nearly spherical smaller ice particles. This hypothesis is supported by a lower co-polar correlation coefficient."*

Figs 22 and 28: It's quite hard to compare the measurements and simulations here due to different y-axis scales and different variables on x-axis. Consider at least making the scale of the y-axis the same for observations and simulations. Can you calculate the fall speeds of your model particles and plot that on x-axis? Also, are your labels and units accurate? E.g. are you plotting reflectivity [dBZ] or spectral reflectivity [dBsZ]?
*Fig. 22: we reduced the y-scale of the modelled reflectivity and add different axis ratio for the T matrix simulation. The goal here is to obtain comparable trends between simulations and the three observations to identify the possibility of conical graupels. The analysis emphasizes spectral polarimetric observations, where polarimetric measurements provide greater insight compared to Doppler velocity data. The Doppler velocity does not express yet the fall velocity of the particles.*
*The linear spectral reflectivity (sZ) is expressed in $mm^6 m^{-3}$. We consider 10\*log10(sZ), which is then expressed in dBZ.*
*Fig. 28: The simulations are removed.*

578: I think you should rephrase this or expand it a bit. At the minute it might be misinterpreted as if chains can only be found if the temperature is below -40. I think what you are trying to say is that although aggregation is usually associated with slightly warmer temperatures (maybe above -25), the presence of electric fields allows chains to form in colder temperature regions, which has been observed at temperatures below -40 (Stith et al 2002; Connolly et al 2005; Gayet et al 2012)? Maybe this could be mentioned around line 65 when you are discussing chains in cold temperatures.
*Rephrased.*

581 onwards: Chains are modelled as prolate particles with axis ratio 7 and ice fraction 0.3. Can you put a reference here as to why you think 0.3 is suitable.
*The simulation is removed.*

At 17:22, From high ZDR and positive ZDR spectrum you hypothesize that there may be chain-like aggregates. The low correlation coefficient suggests a mixture of particles. If small particles and chains co-exist here, could you comment on why sZDR in Fig 28e would be positive. Do you think if there was a high E-field forming the chains, the small particles would be oriented vertically by the field giving negative sZDR?
*Based on this comment and comment line 578, the text is rephrased:*
*Lines 629-634 : "The differential reflectivity of the Rayleigh plateau (Doppler velocity > 5 m s$^{-1}$) is around 0.2 dB, and the entire $Z_{DR}$ spectrum is positive. One hypothesis is that the large particles with positive $sZ_{DR}$ are chain-like aggregates that formed earlier under a strong E-field. The lower copolar correlation coefficient in Fig. 26(g) suggests a mixture of particles (small ice crystals and chains), but currently, there is no high E-field to vertically align the small particles. At that moment, the temper-*

*ature above 9600 m is lower than −40 °C, and it is indeed possible for chains to be present at such temperatures, according to Conolly et al. (2005)."*

You say that you should use axis ratio much greater than 1, but that the maximum you can use is 7 otherwise the code cannot converge. Can you provide more specific details on sizes and axis ratios here please? For example, what are the maximum sizes that have been observed before? You have an example in Fig 1 which is 721 microns. You also say that the individual monomers could be 15-20 microns. Are you then calculating the axis ratio as, for example, 721/15=48? I don't think you can get much information about a chain with axis ratio 48 by modelling it as a prolate particle with axis ratio 7, which you also point out in the next paragraph. You show simulations in Fig. 28 for particles with a much larger maximum dimension of up to 3mm, and then point out that chains of this size have not been observed. Can the code converge if you use smaller sizes less than 1mm with a larger axis ratio?

*The authors agree that simulation result of prolate particle with axis ratio 7 does not give much information about a chain with axis ratio 48. However, the simulation does not converge even if we only use sizes less than 1 mm. (For axis ratio 15, we can reach at most 670 microns, but still it is far from axis ratio 48) So, here, we encounter a clear limitation of the Tmatrix method using spheroids. It cannot approximately simulate chains. Therefore, the simulation is removed.*

I personally think the section is not very useful because the scattering simulations kind of go nowhere, then you say an alternative would be to look at data which isn't possible. I think you should either repeat simulations using more representative particles, or just remove the section and add a line saying that the high ZDR could be caused by chains.

*We focus on the observation. The simulation is removed and the section is shortened.*

640: Is there any scope to track thunderstorm evolution using steerable radars, like they did with weather radars during the WOEST campaign in the UK?

*Delft University of Technology has definitively interest in this topic. There is work in progress regarding tracking using steerable radars (PhD position) and development of a fast-scanning phased-array radar at Ku-band with polarization diversity (PHARA).*

**1.3 Technical corrections**

Line 2: a cloud radar. *Corrected.*

Line 3: a thunderstorm case. *Corrected.*

Line 4/5: maybe "in the millimeter band", or something similar...kind of sounds like its exactly 1mm now. *Corrected.*

Line 5: studies of thunderstorm clouds. *Corrected.*

Line 18: phenomenon. *Corrected.*

Line 38: pellets. *Corrected.*

Line 70: when the ice particle number, High concentrations. *Both corrected.*

Line 73: Evidence. *Corrected.*

Line 101: need brackets around citations. *Brackets added.*

Around line 276: sometimes use 1 sometimes one. *Corrected. We chose "1"*

Line 151: associated with. *Corrected.*

Line 216: dealias. *Corrected.*

Line 232: change would align to can align? *Corrected.*

Line 240: spectra. *Corrected.*

329 Fig 9 caption: change (a-c) to Panels (a-c). Same for (d-f). *Modified.*

365: is are close to zero and do not show much variations. *Corrected.*

473: Scattering simulations are (or a scattering simulation is). *Corrected.*

578: In the case being studied, the temperature. *Corrected.*

580: Scattering simulations are. *Corrected.*

592: in Fig 28c or (Fig 28c). *Corrected.*

---

## Editor Decision (ED1)

The authors study 2 clouds within a thunderstorm case, using a 35 GHz cloud radar with 45 degree elevation. They use spectral polarimetry to study particle properties and other processes. Valuable improvements have been made to the original manuscript.

I recommend the following **minor corrections** before publication.
* * *
I'm not sure if you mention the radar you are using, is it the CLARA radar?

Abstract line 2: change "thunderstorm clouds" to "them"

Sub-sections: I appreciate you following my suggestion to re-structure, however I believe the AMT guidelines state that three sectioning levels are allowed, i.e. 5, 5.1,5.1.1, thus the new sectioning you have chosen might not be allowed.

Line 170: link to the relevant figures (Figs. 3 and 4?)

Line 251: Can you comment on the accuracy of the vertical air velocity estimation?

Line 394: "The centre of the cloud that contained lightning activities was more than 10 km away from the radar, thus the radar could only see the edge of the cloud."
Could you elaborate on this slightly, e.g. What is the maximum distance that can be seen? Are you saying it's 10km, or is it the value of 14969.9m given in Table 1?

Line 398: Also provide the time here (I think you include it later - 16:20:11 to 16:21:37 UTC?)

Line 403: "mainly shows downdrafts from 16:15-16:18"
I think you mean that within the time period examined, these are the times when downdrafts occur. However, saying the cloud "mainly shows downdrafts" could be misinterpreted as you saying there are mainly downdrafts, as opposed to updrafts. At 16:15-16:18 there are updrafts and downdrafts (at different heights)

Line 404: In the second paragraph of section 5.1.1, there is a discussion about updrafts and downdrafts on the edge of the thunderstorm compared to the core, saying that you would expect updrafts in the core. You say that at 16:18-16:22 the core is in sight, while before and after that period (at 16:15-16:18 and after 16:22) the radar sees the edge of the thunderstorm. In Fig A1 it is obvious that at 16:15-16:18 the radar is looking at edge, but is 16:22 – 16:25 not looking at the core?

Line 409: The first two panels are before the negative KDP is observed, right? If so, it would be helpful to the reader to mention that explicitly here.

Fig 12 and Fig 16 seem like they could be combined somehow as they show overlapping time periods. (Though the times chosen in both figures seem a bit random, the time difference between consecutive panels is not the same)

Line 423: In the 4th paragraph of section 5.1.1, you question whether the vertical alignment of particles (as seen by the negative sZDR values in the right part of spectrum at 16:21:05 ) is associated with cloud electrification before lightning. This is done by comparing the times and distances when negative KDP is observed (Fig 11b) to the times and distances of the lightning strikes in Fig B2.
I find some of the text confusing here. You say "negative KDP appears at a distance of 7600 m to 9300 m away from the radar", and "one would expect to observe negative KDP also for ranges beyond 9000 m". Do you mean *height* of 7600 m to 9300 m rather than distance from the radar? The range isn't shown on the KDP plot, right?

Line 428: The Wang et al 2019 paper you have cited doesn't have the vertical wind term you have included in Equation 12

Line. 432: For *a* radar

Line 434: "Vt increases with particle size" – this statement is only broadly true

Line 438: How accurate is the ECMWF forecast of vertical wind shear, and how do you think wind shear over Cabauw would compare to the actual values within the thunderstorm?

Line 440: I'm not sure what you mean here, it sounds like you are saying that large ice chains are more likely than other particles to be aligned by an electric field? Please consider rephrasing this paragraph.

Line 460: Have you already discussed somewhere the possibility that lightning may not be measured by the sensor? How likely it that?

Line 464: You could highlight here that the mode is particularly obvious in the 4th panel otherwise some readers may immediately be drawn to the 2 peaks between 6000-7000m that can be seen in the first panel.

Line 465: think it's worth pointing out here that the temperature is measured vertically and not within the cloud.

Line 486: *on* the line of sight -> *in* the line of sight

Line 488: why is sZDR more negative at the edges and closer to 0 for intermediate velocities?

Line 497: Don't you say elsewhere that *large* SLdr and small rho hv could be caused by low SNR?

Line 501: 5.5 km is a large distance - You say in the intro that the magnitude of the electric field decreases to 3 kV m−1 within 5 km away from the cloud edge, and you only

discuss alignment potentially occurring for fields of 10 kV m-1 and larger. So from those numbers it seems unlikely that there would be alignment here.

Line 510: other -> a different

Line 512: don't separate these paragraphs

Line 515/516: upper part of the spectrum -> I think you mean the spectrogram?

Line 517: I think you are saying that at 17:19:09 there could still be the underlying vertical particles that would cause negative sZDR, but there are also other particles introduced into the same velocity bin which are not aligned, and these dominate the signal.
However, if the particles are in the same velocity bin wouldn't they be aligned too?

Line 519: is SLdr shown here?

Line 523: would be good to be more specific with your summary here, e.g. did the alignment occur before/during/after the lightning?

Line 526: spectrum -> spectrogram or spectra

Line 532: lightnings -> lightning strikes

Line 538: enhanced slanted linear depolarisation ratio and reduced copolar correlation not due to low SNR?

Line 539, 595, and Figs 26 and 27 captions: spectogram -> spectrogram

Line 550:  Evidence of differential attenuation …

Line 551: I haven't read that paper in great detail, but noticed a reference to mean axial ratios of 0.75-0.9 for sizes bigger than 1mm. I guess your definition of axial ratio is just the inverse of theirs? I think you define axis ratio but not axial ratio in your manuscript.

Fig. 25: I like the improvements made to this figure.
Consider labelling the panels left to right instead of top to bottom, i.e. a, b, c in the top row rather than a, d, g. Also, you label some y-axes but not others, please label all y-axes. You could also consider flipping the x-axes on panels g, h, i, which would make it easier to compare to the simulations.

Line 561: In this paragraph please clarify which panels you are referring to, e.g. differential reflectivity of the simulated conical graupel is mostly negative (Fig. 25b?). When you say "delta_co reaches a local maximum and decreases slightly before increasing again. ZDR increases and continues to oscillate." – which panels are you looking at here? Differential backscatter phase (delta_co) is plotted in panel c, but panels f and i show the differential phase shift (Phi_DP), is that right? The behaviour you

describe is not obvious in panel c, I think you have changed the axis limits since the last version so maybe that's why.

Line 561: earlier -> at smaller sizes?

Line 570: suggests -> highlights?

Line 573: could this difference be due to air motion?

Line 585: data *from* Gatidis et al

Line 590: Is this differential attenuation (in which case change dB -> dB/km)? It might be clearer to integrate it and give an estimated value of two-way path integrated attenuation in dB.

Line 592: Could you expand on this slightly please? What distance is the storm from the radar?

Line 595: Might be worth mentioning that in the Fig 24 figure caption (vertical velocity rather than Doppler velocity)

Line 628: is shown -> are shown

Line 630: At 17:22:57 you hypothesize that there could be chains at 10003m. This is because the entire ZDR spectrum is positive with values of about 0.2-0.6 dB. You also say that since there is a low copolar correlation coefficient, there could be a mixture of small ice and chains. Can you comment on why chains are the likely particles here rather than having for example plates or columns with varying aspect ratios?

---

## Author Response (AR2)

**Referee comments**

December 16, 2024

The authors study 2 clouds within a thunderstorm case, using a 35 GHz cloud radar with 45 degree elevation. They use spectral polarimetry to study particle properties and other processes. Valuable improvements have been made to the original manuscript.

I recommend the following **minor corrections** before publication.

*The authors express their gratitude to the Referee for this second thorough review. The Referee can find our responses in blue (italic) below.*
* * *
I'm not sure if you mention the radar you are using, is it the CLARA radar?
*The radar name is now mentioned.*
*Lines 132-136: "The cloud radar used in this study is a dual-frequency scanning polarimetric frequency-modulated continuous-wave (FMCW) radar produced by Radiometer Physics GmbH located at Cabauw, the Netherlands (51.968° N 4.929° E). It is named CLoud Atmospheric RAdar (CLARA) and operates at 35 GHz (Ka-band) and 94 GHz (W-band) in Simultaneous Transmission Simultaneous Reception (STSR) mode and measures at a constant elevation of 45° and constant azimuth of 282° (see Fig. A1) at some selected periods."*

Abstract line 2: change "thunderstorm clouds" to "them"
*Done.*

Sub-sections: I appreciate you following my suggestion to re-structure, however I believe the AMT guidelines state that three sectioning levels are allowed, i.e. 5, 5.1,5.1.1, thus the new sectioning you have chosen might not be allowed.
*Thank you for this information. For the moment, we leave the sub-sections as they are.*

Line 170: link to the relevant figures (Figs. 3 and 4?)
*Done.*
*Lines 175-176: "However, after 17:15, a brief indication of a melting layer can be observed using the radar variables, $Z_{DR}$, $SL_{DR}$ and $\rho_{hv}$ in Fig. 3(a) and Fig. 4.*

Line 251: Can you comment on the accuracy of the vertical air velocity estimation?
*Lines 257-260: "Then, the vertical air velocity $w$ can be estimated by solving Eq. (6) with $V_t = 0$ and $v_H$ and $D$ estimated from the ECMWF model data. The latter estimation may influence the accuracy of vertical air velocity measurements. The ECMWF model supplies an average horizontal wind profile, whereas the cloud radar observations are associated with thunderstorm clouds, where local dynamic variability is anticipated."*

Line 394: "The centre of the cloud that contained lightning activities was more than 10 km away from the radar, thus the radar could only see the edge of the cloud." Could you elaborate on this slightly, e.g. What is the maximum distance that can be seen? Are you saying it's 10km, or is it the value of 14969.9m given in Table 1?
*The sentence is removed and a better introduction of the first cloud is provided.*
*Lines 406-408: "The weather radar images presented in Fig. A1 indicate heavy precipitation occurring at a distance of 10–15 km from the radar between 16:20 and 16:25. Correspondingly, owing to its 45°*

*elevation angle, the cloud radar observes the thundercloud at altitudes between 6 and 10 km, and not the precipitation below."*

Line 398: Also provide the time here (I think you include it later - 16:20:11 to 16:21:37 UTC?)
*Done.*
*Lines 412-413: "Conversely, Fig. 11(b) reveals a cluster of negative $K_{DP}$ values between 7600 m and 9300 m and within the time period 16:20:11 to 16:21:37 UTC, suggesting the alignment of non-spherical small ice particles.*

Line 403: "mainly shows downdrafts from 16:15-16:18" I think you mean that within the time period examined, these are the times when downdrafts occur. However, saying the cloud "mainly shows downdrafts" could be misinterpreted as you saying there are mainly downdrafts, as opposed to updrafts. At 16:15-16:18 there are updrafts and downdrafts (at different heights)
*Rephrased.*
*Line 417: "From Fig. 11(c), downdrafts occur in the first cloud from 16:15 to 16:18 UTC and after 16:22 UTC.*

Line 404: In the second paragraph of section 5.1.1, there is a discussion about updrafts and downdrafts on the edge of the thunderstorm compared to the core, saying that you would expect updrafts in the core. You say that at 16:18-16:22 the core is in sight, while before and after that period (at 16:15-16:18 and after 16:22) the radar sees the edge of the thunderstorm. In Fig A1 it is obvious that at 16:15-16:18 the radar is looking at edge, but is 16:22 – 16:25 not looking at the core?
*We discussed the cloud radar observations in the second paragraph of section 5.1.1. We added the following sentences for the comparison cloud radar data - weather radar data:*
*Lines 422-424: "Cloud observations between 6 and 10 km height generally show good agreement with the precipitation patterns and intensity measured by the weather radar at lower heights in Fig. A1. However, timing differences on the order of 1 minute may arise due to the differing temporal resolutions of the two radars."*

Line 409: The first two panels are before the negative KDP is observed, right? If so, it would be helpful to the reader to mention that explicitly here.
*Added.*
*Line 425: "Figure 12 shows the spectral $Z_{DR}$ across the period when negative $K_{DP}$ is observed (panels 3-4).*

Fig 12 and Fig 16 seem like they could be combined somehow as they show overlapping time periods. (Though the times chosen in both figures seem a bit random, the time difference between consecutive panels is not the same)
*These figures have different purposes. Fig. 12 displays the possibility of particle alignment and shows a time period of more than 3 minutes. For this time period, Fig. 16 provides a time zoom of 1 minute where the presence of liquid water is identified.*

Line 423: In the 4th paragraph of section 5.1.1, you question whether the vertical alignment of particles (as seen by the negative sZDR values in the right part of spectrum at 16:21:05 ) is associated with cloud electrification before lightning. This is done by comparing the times and distances when negative KDP is observed (Fig 11b) to the times and distances of the lightning strikes in Fig B2.
I find some of the text confusing here. You say "negative KDP appears at a distance of 7600 m to 9300 m away from the radar", and "one would expect to observe negative KDP also for ranges beyond 9000 m". Do you mean height of 7600 m to 9300 m rather than distance from the radar? The range isn't shown on the KDP plot, right?
*The authors thank the Referee for spotting this issue. Corrections are done.*
*Lines 437-439: "Negative $K_{DP}$ are observed within the height range of 7600 m to 9300 m, whereas the lightning strikes occurred at least 13000 m away from the radar. If the electric field that caused these lightning strikes is responsible for the alignment of particles observed, one would expect to observe negative $K_{DP}$ also for heights beyond 9000 m."*
*In the whole manuscript, for the profiles and spectrograms, the y-axis represents the height (and not*

*the range).*

Line 428: The Wang et al 2019 paper you have cited doesn't have the vertical wind term you have included in Equation 12
*Rephrased.*
*Lines 444-445: "By modifying the formulation of Wang et al. (2019) to incorporate vertical wind velocity, the horizontal and vertical particle velocities can be expressed as:"*

Line. 432: For a radar
*Corrected in Line 449.*

Line 434: "Vt increases with particle size" – this statement is only broadly true
*Line 451: "When $V_t$ increases, a negative shear s causes the spectrum to widen as the left side shifts more than the right......"*

Line 438: How accurate is the ECMWF forecast of vertical wind shear, and how do you think wind shear over Cabauw would compare to the actual values within the thunderstorm?
*Rephrased.*
*Lines 454-459: "For a spectrum width of 10 m s$^{-1}$ and a terminal velocity ($V_t$) of 2 m s$^{-1}$, corresponding to the upper bound for plate-like particles (Spek et al. 2008), a shear of approximately 25000 m s$^{-1}$ km$^{-1}$ would be required to invert the spectrum. This value is substantially higher than the observed shear of 4 m s$^{-1}$ km$^{-1}$ between 7500 m and 10000 m in ECMWF data, as shown in Fig. 15(c). While recognizing the limitations of ECMWF wind shear data in the context of thunderclouds, a wind shear of 25000 m s$^{-1}$ km$^{-1}$ is highly improbable. Therefore, wind shear is unlikely to account for the negative $sZ_{DR}$ observed on the left side of the spectrum.*

Line 440: I'm not sure what you mean here, it sounds like you are saying that large ice chains are more likely than other particles to be aligned by an electric field? Please consider rephrasing this paragraph.
*Rephrased.*
*Lines 460-461: "Alternatively, the hypothesis is that the axis ratios of small particles are close to one and that the electric fields could align larger particles vertically, leading to negative $sZ_{DR}$ on the left side of the Doppler spectrum.*

Line 460: Have you already discussed somewhere the possibility that lightning may not be measured by the sensor? How likely it that?
*We mentioned this in the text but we did not investigate the detection limitations of BLIDS for cloud-to-cloud lightning. In the future, we would like to use a more accurate information of lightnings.*

Line 464: You could highlight here that the mode is particularly obvious in the 4th panel otherwise some readers may immediately be drawn to the 2 peaks between 6000-7000m that can be seen in the first panel.
*Thank you. Added.*
*Line 486: "This separate mode is most clearly discernible in the fourth panel."*

Line 465: think it's worth pointing out here that the temperature is measured vertically and not within the cloud.
*The authors think it is clear that the microwave radiometer provides vertical profiles of air temperature. Later in the manuscript at the end of section 5.2.2.1, the following sentence is present: "Nonetheless, the temperature profile inside the thunderstorm cloud may be different from the temperature profile measured by the microwave radiometer looking towards the zenith, .....(Lines 637-639).*

Line 486: on the line of sight → in the line of sight
*Corrected in Line 507.*

Line 488: why is sZDR more negative at the edges and closer to 0 for intermediate velocities?
*We added the following sentences to explain this:*

*Lines 510-515 : "The $sZ_{DR}$ values are predominantly negative across the entire spectrum, with an average value of $-0.12$ dB. Analysis of the spectrum at 8018 m, in comparison with the simulations presented in Fig. 10, indicates that $sZ_{DR}$ aligns with the behavior expected for slightly oblate particles with a canting angle of $\beta = 90$ deg. Specifically, negative values are observed on the right side of the measured spectrum, increasing with particle size before decreasing toward the left side coinciding with the first Mie minimum. At heights exceeding 8000 m, the spectra become broader and exhibit diminished resonance features due to enhanced turbulence."*

Line 497: Don't you say elsewhere that large SLdr and small rho hv could be caused by low SNR?
*Yes. The Referee probably refers to Lines 175-178 where the impact of important attenuation on the polarimetric variables is shortly discussed. In that case, we have another situation where the reflectivity is large, SLdr is small, rho hv has large values approaching 1, therefore we think of a possible alignment of the ice particles that decreases canting angle variance.*
*Line 524: "During this period, $\rho_{hv}$ does not change significantly and is high (Fig. 21(c))."*
*We removed "which means that the decrease in $SL_{DR}$ is not due to low SNR."*

Line 501: 5.5 km is a large distance - You say in the intro that the magnitude of the electric field decreases to 3 kV m$^{-1}$ within 5 km away from the cloud edge, and you only discuss alignment potentially occurring for fields of 10 kV m$^{-1}$ and larger. So from those numbers it seems unlikely that there would be alignment here.
*That was the largest peak current measured among 162 cloud-to-cloud lightning discharge. It corresponds well in time to the apparition of the sZDR negative values on the right side of the Doppler spectrum while apparently there is no change of particle populations. We still think that there is a correlation between these negative values of sZDR and this lightning. However, we would have liked to have quantitative values of the electric field in the line-of-sight of the cloud radar.*

Line 510: other $\rightarrow$ a different
*Corrected in Line 537.*

Line 512: don't separate these paragraphs
*The second paragraph is now removed.*

Line 515/516: upper part of the spectrum $\rightarrow$ I think you mean the spectrogram?
*We removed this paragraph.*

Line 517: I think you are saying that at 17:19:09 there could still be the underlying vertical particles that would cause negative sZDR, but there are also other particles introduced into the same velocity bin which are not aligned, and these dominate the signal.
However, if the particles are in the same velocity bin wouldn't they be aligned too?
*Yes, they should be aligned too. We agree with the Referee's comment. That is corrected by removing the paragraph and figure related to this discussion.*

Line 519: is SLdr shown here?
*No SLdr is not shown because at the heights of interest, its value does not really change versus time.*

Line 523: would be good to be more specific with your summary here, e.g. did the alignment occur before/during/after the lightning?
*We indicated time estimates for the two cases, and for the summary we added the following sentences: Lines 544-545: "The vertical alignment of ice particles was observed 2 to 5 seconds prior to the lightning strike and dissipated 5 to 6 seconds afterward. These temporal estimates account for the measurement timing of chirp 3."*
*We modified the conclusion accordingly in Lines 715-716.*

Line 526: spectrum $\rightarrow$ spectrogram or spectra
*Corrected in Line 546.*

Line 532: lightnings → lightning strikes
*Corrected in Line 552.*

Line 538: enhanced slanted linear depolarisation ratio and reduced copolar correlation not due to low SNR?
*We added the following:*
*Lines 558-561: From Fig. 19 (c) and (d), this region has enhanced slanted linear depolarisation ratio and reduced copolar correlation coefficient. In fact, the SNR from 17:22 to 17:24 UTC at 4000 m to 6000 m ranges from 14.0 dB to 40.8 dB with a mean of 31.1 dB, suggesting that the enhanced slanted linear depolarisation ratio and reduced copolar correlation values are not due to low SNR.*

Line 539, 595, and Figs 26 and 27 captions: spectogram → spectrogram
*Corrected.*

Line 550: An Evidence of differential attenuation . . .
*Corrected in Line 550.*

Line 551: I haven't read that paper in great detail, but noticed a reference to mean axial ratios of 0.75-0.9 for sizes bigger than 1mm. I guess your definition of axial ratio is just the inverse of theirs? I think you define axis ratio but not axial ratio in your manuscript.
*Corrected. We use the parameter axis ratio only in the whole article. It is defined in Lines 308-309.*
*Line 572-574 : "From the literature, the theoretical axis ratio of conical graupel is 1.05, while measurements of mean axis ratios of conical graupels show values ranging from 1.1 to 1.3 for sizes in excess of 1 mm (Spek et al. 2008).*

Fig. 25: I like the improvements made to this figure.
 *We also appreciate the improvements made possible by the Referee's suggestions.*
Consider labelling the panels left to right instead of top to bottom, i.e. a, b, c in the top row rather than a, d, g.
 *For consistency that should be done for the whole article and text. Therefore, we left it as it is.*
Also, you label some y-axes but not others, please label all yaxes.
 *All the y-axes are now labelled.*
You could also consider flipping the x-axes on panels g, h, i, which would make it easier to compare to the simulations.
*Again for consistency with all the other figures showing Doppler spectra, we don't flip the x-axis.*

Line 561: In this paragraph please clarify which panels you are referring to, e.g. differential reflectivity of the simulated conical graupel is mostly negative (Fig. 25b?).
*We added references to the different panels for clarity.*
When you say "$\delta_{co}$ reaches a local maximum and decreases slightly before increasing again. ZDR increases and continues to oscillate." – which panels are you looking at here?
*Thank you for spotting this. Some parts of these sentences relate to the figure of the previous version where the equal-volume sphere radius range was larger. This is corrected in Lines 586-587.*
Differential backscatter phase ($\delta_{co}$) is plotted in panel c *(right)*, but panels f and i show the differential phase shift (PhiDP), is that right? *panel f also shows ($\delta_{co}$) while panel i shows the differential phase shift (PhiDP). Both simulations (panels c and f) provide $\delta_{co}$ and the measurement (i) supplies ($s\Psi_{DP}$). We wrote the following sentence for clarification: "Since the constant spectral differential propagation phase ($s\Phi_{DP}$) is nearly 0 deg. (Doppler velocities from 2 to -1 $ms^{-1}$ in Fig. 25(i)), the spectral differential phase shift ($s\Psi_{DP}$) corresponds to the spectral differential backscatter phase $s\delta_{co}$." in Lines 588-590.*
The behaviour you describe is not obvious in panel c, I think you have changed the axis limits since the last version so maybe that's why.
*That is indeed the reason. Corrected.*

Line 561: earlier → at smaller sizes?
*Modified in Line 585.*

Line 570: suggests → highlights?
*The sentence has been changed.*

Line 573: could this difference be due to air motion?
*The air motion affects the location of the first Mie minimum for the measurements $sZ_{hh}$, $sZ_{DR}$, and $s\Psi_{DP}$ in the same way. Here, the medium is complex, there are probably two types of ice particles and the first Mie minimum location difference may indicate the possibility of an extra dependency: shape. Lines 603-607: "Furthermore, the location of the measured first Mie minimum is influenced by both the equi-volume sphere radius and air velocity. However, a comparison of polarimetric spectra related to the same radar resolution volume reveals variations in the Mie minimum location, indicating an additional dependence on particle shape. Consequently, simultaneous consideration of the three parameters—$sZ_{hh}$, $sZ_{DR}$, and $s\Psi_{DP}$—at the same time and height is essential for a comprehensive analysis."*

Line 585: data from Gatidis et al
*Corrected in Line 616.*

Line 590: Is this differential attenuation (in which case change dB → dB/km)? It might be clearer to integrate it and give an estimated value of two-way path integrated attenuation in dB.
*Corrected.*
*Lines 621-622: "Thus, the two-way path integrated differential attenuation contribution is expected to be low, at less than 0.12 dB, and does not significantly affect the interpretation of the results discussed."*

Line 592: Could you expand on this slightly please? What distance is the storm from the radar?
*Corrected and expanded.*
*Lines 623-626: "It is worth noting from Fig. 24(a) and (b) that the population of graupel ends at around 4000 m height, which means the region with graupel is localised in the thunderstorm cloud. Since the radar is looking at an elevation angle of 45°, this suggests that graupel is not present closer than 5700 m from the radar. At this range, measurements cannot be obtained at lower altitudes due to the 45° elevation angle. Below 4000 m, graupel begins to melt."*

Line 595: Might be worth mentioning that in the Fig 24 figure caption (vertical velocity rather than Doppler velocity)
*Thank you. That is done now.*

Line 628: is shown → are shown
*Corrected in Line 662.*

Line 630: At 17:22:57 you hypothesize that there could be chains at 10003m. This is because the entire ZDR spectrum is positive with values of about 0.2-0.6 dB. You also say that since there is a low copolar correlation coefficient, there could be a mixture of small ice and chains. Can you comment on why chains are the likely particles here rather than having for example plates or columns with varying aspect ratios?
*The authors modified the section and propose an hypothesis with more consistency than the previous one. The trigger was the large increase of the differential propagation phase. The hypothesis is a mixture of two types of ice particles: oblate ice particles coated with liquid water (larger particles) and chains (smaller particles). We think that the entire ZDR spectrum is attenuated and expect higher values of ZDR. The same for the spectral reflectivity. Therefore, we think that chains are more probable than plates.*